# RNA-triggered Cas12a3 cleaves tRNA tails to execute bacterial immunity

Oleg Dmytrenko[1,11], Biao Yuan[2,11], Kadin T. Crosby[3], Max Krebel[1], Xiye Chen[1], Jakub S. Nowak[4], Andrzej Chramiec-Głąbik[4], Bamidele Filani[3], Anne-Sophie Gribling-Burrer[5], Wiep van der Toorn[6,7], Max von Kleist[6,7], Tatjana Achmedov[1], Redmond P. Smyth[5], Sebastian Glatt[4,8], Jack P. K. Bravo[9], Dirk W. Heinz[2 ✉], Ryan N. Jackson[3 ✉] & Chase L. Beisel[1,10 ✉]

In all domains of life, tRNAs mediate the transfer of genetic information from mRNAs to proteins. As their depletion suppresses translation and, consequently, viral replication, tRNAs represent long-standing and increasingly recognized targets of innate immunity[1–5]. Here we report Cas12a3 effector nucleases from type V CRISPR–Cas adaptive immune systems in bacteria that preferentially cleave tRNAs after recognition of target RNA. Cas12a3 orthologues belong to one of two previously unreported nuclease clades that exhibit RNA-mediated cleavage of non-target RNA, and are distinct from all other known type V systems. Through cell-based and biochemical assays and direct RNA sequencing, we demonstrate that recognition of a complementary target RNA by the CRISPR RNA triggers Cas12a3 to cleave the conserved 5′-CCA-3′ tail of diverse tRNAs to drive growth arrest and anti-phage defence. Cryogenic electron microscopy structures further revealed a distinct tRNA-loading domain that positions the tRNA tail in the RuvC active site of the nuclease. By designing synthetic reporters that mimic the tRNA acceptor stem and tail, we expanded the capacity of current CRISPR-based diagnostics for multiplexed RNA detection. Overall, these findings reveal widespread tRNA inactivation as a previously unrecognized CRISPR-based immune strategy that broadens the application space of the existing CRISPR toolbox.

Immune defences across all domains of life counteract viral infections by clearing the invader or disabling host processes that are essential for viral replication. One growing theme associated with innate immune systems is the inactivation of tRNAs[1–5]. tRNAs have a critical role in translation, serving as the bridge between mRNAs and nascent proteins. Accordingly, inactivating a portion of the tRNA pool can impair the synthesis of viral proteins or drive systematic cellular shutdown to block viral replication[1,2,4,6,7]. Representative bacterial defences such as PrrC, VapC, colicin E5 and PARIS cleave the anticodon loop of specific tRNAs[5,8–11]. In animals, the type I interferons SLFN11 and SAMD9 cleave the acceptor stem and anticodon loop, respectively, of tRNAs to suppress codon-specific production of viral particles[4,7,12,13].

Conspicuously absent from the set of immune defences that specifically use tRNA inactivation are CRISPR–Cas systems, the only known source of adaptive immunity in bacteria and archaea[14]. These widespread systems immunize against future infection by acquiring snippets of viral sequences expressed as CRISPR RNAs (crRNAs) that pair with CRISPR-associated (Cas) effector nucleases. The complex then searches for complementary target RNA or DNA that match the originating virus and, after target recognition, cleaves the bound nucleic acid targets to clear viral genomes or transcripts[15–17]. Some activated Cas nucleases also collaterally cleave non-target RNA or DNA with little sequence preference to halt cellular processes that are essential for viral replication and to drive growth arrest[18–21]. One such effector, the RNA-triggered Cas13 nuclease from *Leptotrichia shahii* (*Lsh*Cas13a), was recently shown to cleave U-rich anticodon loops associated with a subset of tRNAs when activated in *Escherichia coli*[22]. However, *Lsh*Cas13a also efficiently cleaves its target RNA at U-rich sequences to drive targeted silencing[23–26]. Thus, it remains unknown whether any CRISPR–Cas systems have evolved to preferentially cleave tRNAs over other RNA species, including their target RNA, as part of an immune response.

Here we report a previously uncharacterized clade of Cas nucleases, which we term Cas12a3. After target RNA recognition, these nucleases preferentially cleave the conserved 3′ CCA tails of tRNAs to drive growth arrest and block phage dissemination. Activated Cas12a3 engages the tRNA tail, acceptor stem and T-arm to load tRNA substrates into its RuvC nuclease domain for cleavage. Harnessing the distinct properties of Cas12a3, we expanded the multiplexing capacity of current CRISPR-based RNA detection approaches to illustrate one of the many applications of programmable RNA-mediated tRNA cleavage.

[1]Helmholtz Institute for RNA-based Infection Research (HIRI), Helmholtz Centre for Infection Research (HZI), Würzburg, Germany. [2]Helmholtz Centre for Infection Research (HZI), Braunschweig, Germany. [3]Department of Chemistry and Biochemistry, Utah State University, Logan, UT, USA. [4]Malopolska Center of Biotechnology, Jagiellonian University, Krakow, Poland. [5]Architecture et Réactivitié de l'ARN, Université de Strasbourg, CNRS, Institute of Molecular and Cellular Biology (IBMC), University of Strasbourg, Strasbourg, France. [6]Department of Mathematics and Computer Science, Freie Universität Berlin, Berlin, Germany. [7]Project Groups, Robert Koch Institute, Berlin, Germany. [8]University of Veterinary Medicine, Vienna, Austria. [9]Institute of Science and Technology Austria (ISTA), Klosterneuburg, Austria. [10]Medical Faculty, University of Würzburg, Würzburg, Germany. [11]These authors contributed equally: Oleg Dmytrenko, Biao Yuan. ✉e-mail: dirk.heinz@helmholtz-hzi.de; ryan.jackson@usu.edu; chase.beisel@helmholtz-hiri.de

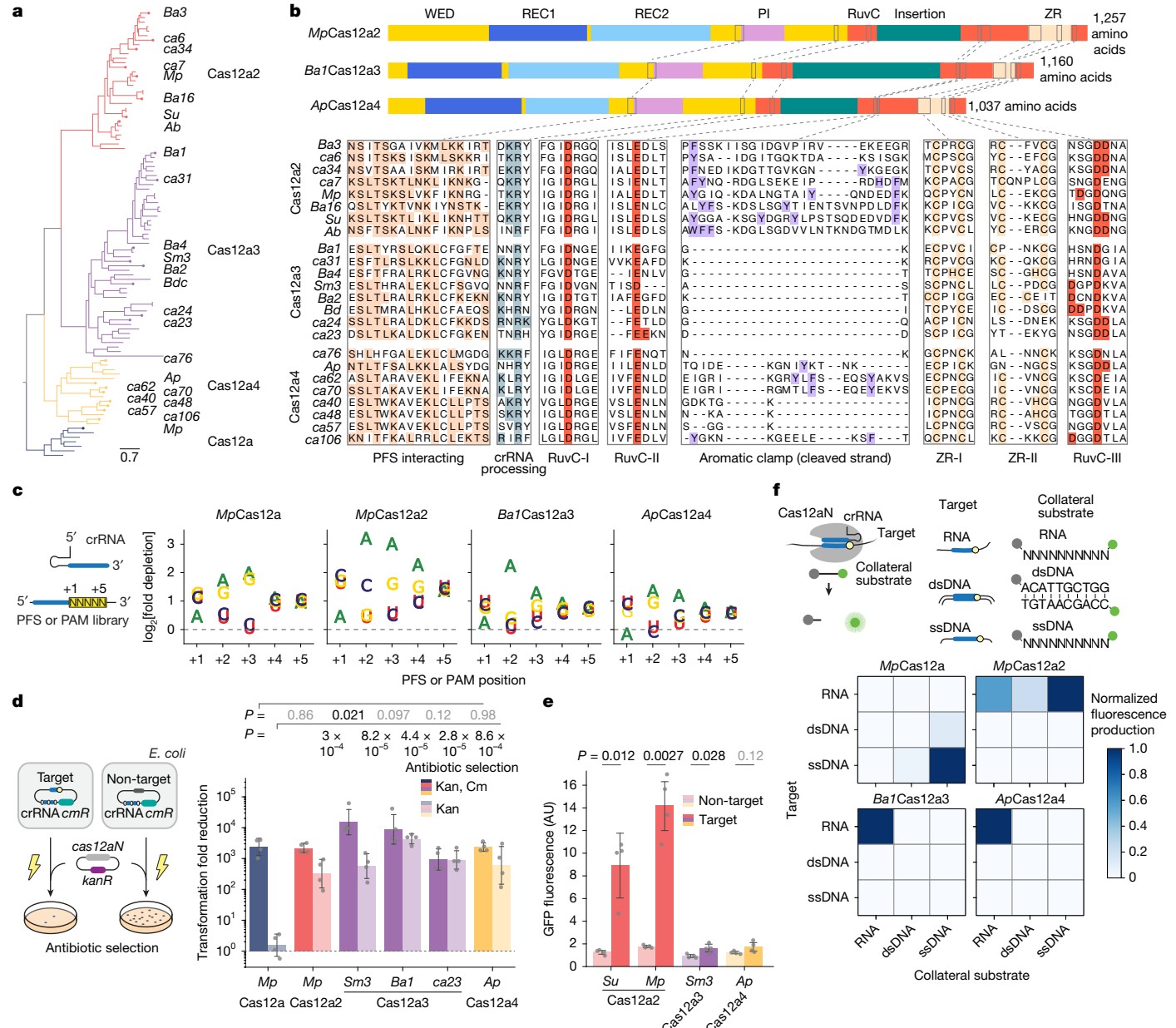

**Fig. 1 | Two distinct clades related to type V Cas12a2 nucleases exhibit RNA-activated cleavage of RNA but not DNA. a**, Phylogenetic analysis depicting Cas12a and the related Cas12a2, Cas12a3 and Cas12a4 clades. **b**, Domain and targeted sequence alignment across representative Cas12a2, Cas12a3 and Cas12a4 nucleases. **c**, Schematic (left) and quantification (right) of the nucleotide-depletion screen as part of the PFS (Cas12a2, Cas12a3 and Cas12a4) or protospacer-adjacent motif (PAM; Cas12a) determination for representative nucleases. Additional PFS screens are presented in Supplementary Fig. 2a. Results are the averages of independent experiments ($n = 2$). Note that the PAM for $Mp$Cas12a is the reverse complement of the sequence commonly reported for Cas12a nucleases (YYV). **d**, Schematic (left) and quantification (right) of the assessment of plasmid clearance versus growth arrest in *E. coli* based on variations of a plasmid interference assay. Plasmid clearance and growth arrest were differentiated on the basis of antibiotic selection for the target plasmid or

the target plasmid and the nuclease plasmid. *cmR*, chloramphenicol-resistance cassette; *kanR*, kanamycin resistance cassette; Kan, kanamycin; Cm, chloramphenicol. **e**, Quantification of induction of the SOS DNA damage response in *E. coli* based on a transcriptional fluorescent reporter. Bars and error bars in **d** and **e** represent the geometric mean ± geometric s.d. and the mean ± s.d., respectively, of independent experiments starting from separate colonies ($n = 4$), with grey dots representing each measurement. **f**, Schematic (top) and measurement (bottom) of the assessment of different targets and cleavage substrates in vitro. Values represent the mean of independent experiments ($n = 3$ or $4$). Complete time courses are shown in Fig. 2a and Supplementary Fig. 2e. Statistical analyses were performed using two-tailed Welch's *t*-tests with all biological replicates. *P* values that are not significant ($P \geq 0.05$) are shown in grey. AU, arbitrary units. Illustrations of thunderbolts and Petri dishes in **d** reproduced from ref. 27, CC BY 4.0.

## Cas12a3 and Cas12a4 halt growth without DNA damage

The phylogenetic proximity of the functionally diverse Cas12a nucleases (which target and cleave complementary DNA)[17] and Cas12a2 nucleases (which target RNA and then indiscriminately shred RNA and DNA)[27,28] indicated that other functionalities might exist in the adjacent

phylogenetic space. To explore this possibility, we searched public databases for sequences closely related to Cas12a2. We identified 61 orthologues primarily from environmental metagenomic assemblies that resolved into two clades distinct from Cas12a2, Cas12a and each other (Fig. 1a,b, Extended Data Fig. 1 and Supplementary Data 1). Most of these were associated with CRISPR arrays and the spacer-acquisition

genes *cas1*, *cas2* and *cas4* (Supplementary Table 1). We tentatively designate members of these clades Cas12a3 and Cas12a4, as further variants of Cas12a. These two clades retain the three motifs that form the canonical RuvC endonuclease domain and the domains and conserved residues involved in the processing of crRNA, the recognition of the protospacer-flanking sequence (PFS) and the zinc ribbon (ZR) (Fig. 1b). However, orthologues from the identified clades showed limited conservation of the aromatic clamp residues in Cas12a2, which are crucial for broad DNA collateral cleavage[28], with Cas12a3 lacking these regions entirely (Fig. 1b and Extended Data Fig. 1).

To investigate the possibility that Cas12a3 and Cas12a4 have distinct activities compared with Cas12a2, we first characterized representative Cas12a3 and Cas12a4 nucleases using our previously established plasmid interference assays in *E. coli*[27]. The nucleases were treated with a library of potential PFS sequences encoded in a target plasmid under antibiotic selection (Fig. 1c and Supplementary Fig. 2a). This assay identified a preference for a purine-rich PFS, a finding in line with the known preferences of Cas12a2 nucleases and the high amino-acid similarity in the PFS-interacting region across the orthologues (Fig. 1b). Using a consensus PFS, the Cas12a3 and Cas12a4 nucleases reduced the number of transformants even without antibiotic selection for the target plasmid, similar to a representative Cas12a2 from the microbial community of *Microcerotermes parvus* (*Mp*Cas12a2) (Fig. 1d). We obtained comparable results for different target sequences (Supplementary Fig. 2b) and observed impaired growth in liquid culture without antibiotic selection (Supplementary Fig. 2c). Cas12a2 and Cas12a3 nucleases further provided defence against T4 phage infection (Supplementary Fig. 2d). However, unlike Cas12a2, the Cas12a3 and Cas12a4 nucleases did not induce a measurable SOS DNA damage response, as indicated by a transcriptional reporter driven from the *recA* promoter[27] (Fig. 1e). Thus, although Cas12a3 and Cas12a4 nucleases arrest growth after activation in a manner similar to Cas12a2, both nucleases induce a distinct mechanism of immunity.

## Active Cas12a3 and Cas12a4 cleave RNA but not DNA

We next examined whether Cas12a3 and Cas12a4 orthologues accept DNA and RNA targets and cleavage substrates (Fig. 1f). Given that the Cas nucleases could have sequence preferences for specific substrates[27,29,30], we used a randomized library of RNA and single-stranded DNA (ssDNA) substrates and a mixed-nucleotide sequence for the double-stranded DNA (dsDNA) substrate. As expected, *Mp*Cas12a (as a representative Cas12a nuclease) primarily cleaved the ssDNA substrate library and, to a limited extent, the dsDNA substrate in response to the ssDNA and dsDNA targets, respectively[29,31]. Moreover, *Mp*Cas12a2 cleaved all three substrates in response to the RNA target[27,28] (Fig. 1f and Supplementary Fig. 2e). Similar to *Mp*Cas12a2, Cas12a3 from an unknown *Bacteroidetes* bacterium (*Ba1*Cas12a3) and Cas12a4 from a microbial community of the *Alvinella pompejana* hydrothermal vents worm (*Ap*Cas12a4) were also activated by the RNA target. However, they exclusively cleaved the RNA substrate library. When presented with additional individual ssRNA, ssDNA or dsDNA substrates in vitro, similar trends were observed for these nucleases and for Cas12a3 from *Smithella* sp. M82 (*Sm3*Cas12a3) (Extended Data Fig. 2). Notably, *Ba1*Cas12a3 and *Sm3*Cas12a3 only partially cleaved their target RNA even after prolonged incubation, which was in contrast to complete cleavage by *Ap*Cas12a4 and *Mp*Cas12a2. This result suggests that the RNA target is not the preferred cleavage substrate of Cas12a3. RNA-mediated RNA cleavage by Cas12a3 and Cas12a4 is consistent with the lack of a measurable SOS response in *E. coli* (Fig. 1d) and the absence of the aromatic clamp amino-acid residues in the nucleases of these clades (Fig. 1b and Extended Data Fig. 1). These observations reveal that exclusive RNA-activated RNA cleavage also occurs in the highly diverse family of Cas12 nucleases[32], similar to the cleavage activities of the phylogenetically distinct Cas13 nucleases[33].

## Cas12a3 prefers specific RNA substrates

Building on the limited cleavage of target RNA by these nucleases, we noted that *Ba1*Cas12a3 also exhibited 3.8-fold less cleavage of the RNA substrate library compared with *Mp*Cas12a2 (Fig. 2a). By contrast, *Ap*Cas12a4 exhibited a continual increase in fluorescence over the course of the reaction without plateauing and efficiently cleaved its target RNA in vitro (Extended Data Fig. 2). We speculated that reduced cleavage of the RNA library by *Ba1*Cas12a3 resulted from a selective preference for specific RNA substrates.

To identify RNA substrates relevant to an immune response, we tested *Ba1*Cas12a3 in a cell-free transcription–translation (TXTL) system that closely mimics the bacterial cellular environment[34,35]. As expected, adding an activated *Mp*Cas12a2 or *Ba1*Cas12a3 ribonucleoprotein complex together with a GFP reporter plasmid reduced fluorescence relative to a non-target control (Fig. 2b). To distinguish whether silencing of the reporter resulted from cleavage of the *gfp* transcript or other essential RNA components (for example, tRNAs or rRNAs), we added activated nucleases 4 h before introducing a GFP-expressing reporter plasmid. Notably, GFP fluorescence was abolished when activated *Ba1*Cas12a3 was added before the reporter plasmid, whereas *Mp*Cas12a2, even when activated, had no effect relative to the non-target control (Fig. 2b). The enhanced silencing obtained only by activated *Ba1*Cas12a3 suggests that this nuclease selectively degrades RNA components essential for gene expression.

## Cas12a3 cleaves tRNA tails

We aimed to identify the RNA substrates of activated *Ba1*Cas12a3 that underlie silencing of the reporter in TXTL assays. To that end, we directly sequenced RNA from 4-h reactions using nanopore direct RNA sequencing, which enables complete sequencing of chemically modified RNAs[36] (Supplementary Fig. 3). Comparisons of read coverage between the non-target and target conditions showed that no notable cleavage of 5S, 16S or 23S rRNAs or the target RNA occurred. By contrast, many of the reads mapped to tRNAs (27 out of 47) were significantly ($z$ score $\geq 2$) truncated roughly 2–4 nucleotides upstream of their 3′ aminoacylated ends up to the discriminator base and the CCA tail conserved across all tRNAs (Fig. 2c and Supplementary Fig. 3c). Cleavage of selected tRNAs was confirmed by northern blot analysis (Supplementary Fig. 4). Similar cleavage patterns were observed for *Sm3*Cas12a3 in TXTL assays (25 out of 47) (Supplementary Figs. 3b,d and 4). This specific cleavage of tRNAs accounts for the target-dependent silencing of the reporter observed in TXTL assays, as it would disrupt the translation machinery before the reporter plasmid is added.

To further characterize the cleavage patterns in tRNA by Cas12a3 orthologues, we established a more controlled in vitro assay using purified bulk tRNAs from *E. coli* incubated with activated nucleases (Fig. 2d). Activated *Ba1*Cas12a3 led to significant 3′ cleavage ($z$ score $\geq 2$) of all but 3 out of the 49 mapped tRNAs, even though the remaining tRNAs (tRNA$^{\text{Asp(GUC)}}$, tRNA$^{\text{Val(GAC)1}}$ and tRNA$^{\text{Val(GAC)2}}$) still underwent measurable cleavage (Supplementary Fig. 5), with cleavage principally occurring three to five nucleotides upstream of each tRNA 3′ end. Similar cleavage patterns were observed with *Sm3*Cas12a3 and *ca23*Cas12a3 (identified in a wastewater microbial metagenome). However, the cleavage sites for *Sm3*Cas12a3 were shifted slightly upstream (Fig. 2d and Supplementary Fig. 5), which indicated possible mechanistic differences within the Cas12a3 clade. Consistent with the direct RNA sequencing results, *Ba1*Cas12a3 trimmed the 3′ end of the entire pool of *E. coli* tRNAs labelled with a 5′ fluorophore (Fig. 2e, top), including specific tRNAs detected by northern blot analysis (Supplementary Fig. 6). Although these tRNAs were charged with amino acids and contained extensive chemical modifications associated with *E. coli*[37], *Ba1*Cas12a3 similarly cleaved the same pool of chemically modified tRNAs with the amino

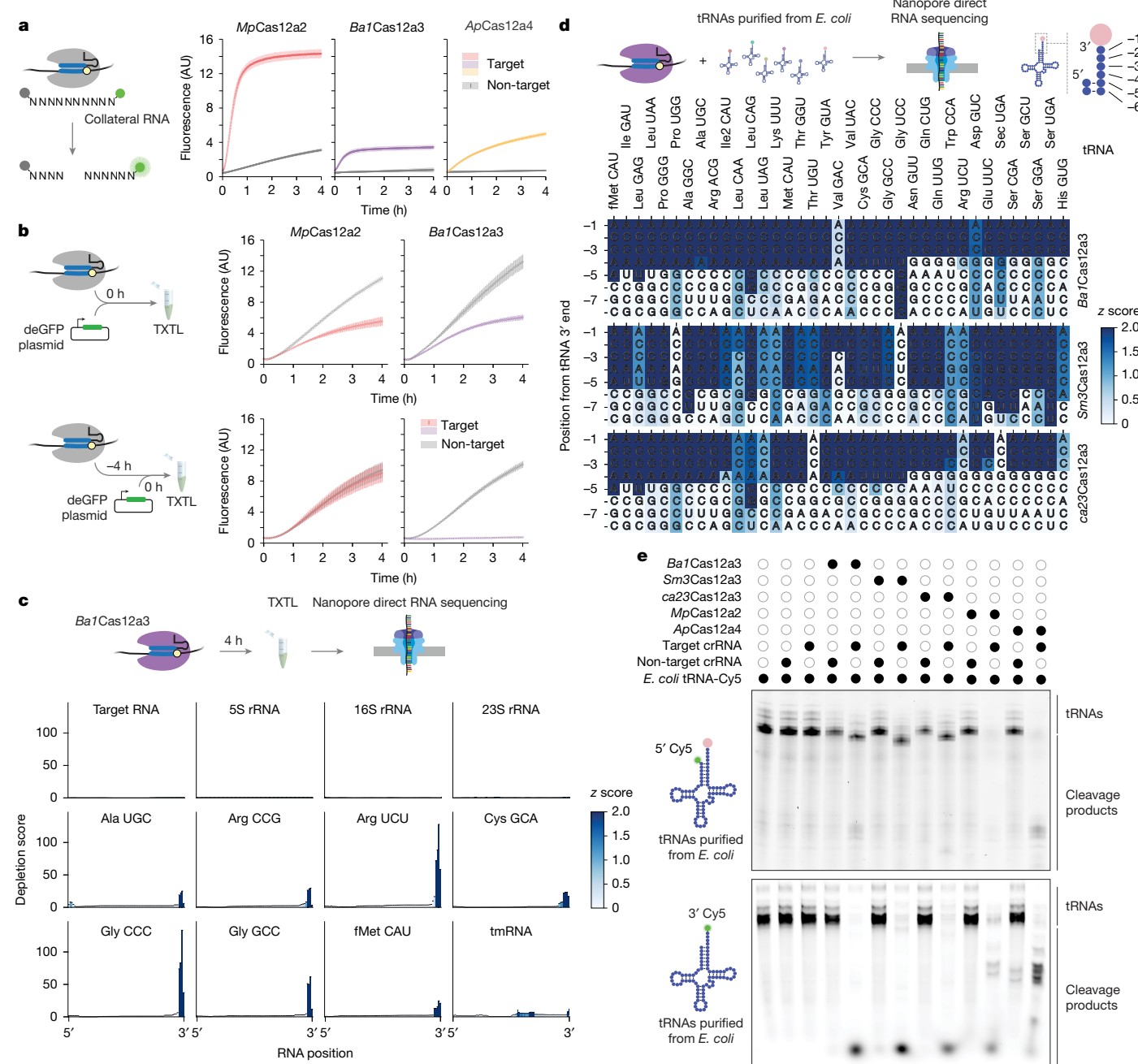

**Fig. 2 | Cas12a3 members preferentially cleave the conserved tail of tRNAs.**
**a**, Schematic (left) and quantification (right) of the time course of cleavage of the RNA substrate library from Fig. 1f. Dots and bars represent the mean ± s.d. from independently mixed in vitro reactions ($n = 4$). AU, arbitrary units.
**b**, Schematic (left) and quantification (right) of the fluorescence time courses of TXTL assays assessing nuclease activity based on silencing of deGFP expression. Dots and bars represent the mean and s.d. from independently mixed TXTL reactions ($n = 3$ or 4). **c**, Schematic (top) and quantification (bottom) of RNA sequencing of total RNA ≤ 200 nucleotides by Nanopore from TXTL reactions with *Ba1*Cas12a3 and a target or non-target RNA after 4 h. Values represent independent TXTL reactions ($n = 3$). The colour map indicates the $z$ score of depletion scores at each position. **d**, Schematic (top) and quantification (bottom) of Nanopore sequencing of purified *E. coli* MRE600 tRNAs incubated with *Ba1*Cas12a3, *Sm3*Cas12a3 or *ca23*Cas12a3 under targeting versus non-targeting

conditions. Values represent independent reactions ($n = 3$), with depletion $z$ scores for each nucleotide as a colour map. Shown tRNAs (labelled with the corresponding anticodons) were detected in every nuclease reaction across the three independent experiments. Apparent cleavage sites ending in A could be shorter by one or (in the case of tRNA$^{fMet(CAU)}$) two nucleotides towards the 3′ end owing to the use of a poly(A) extension to the cleavage product. See Supplementary Figs. 3 and 5 for full tRNA sequences with single-nucleotide depletion scores. **e**, Cleavage patterns of *E. coli* MRE600 tRNAs incubated with activated nucleases. The tRNA pool was 5′ labelled with a fluorophore, leaving the amino acyl group attached (top), or by replacing the 3′ amino acyl group with a fluorophore (bottom). Gel images are representative of independent cleavage reactions ($n = 3$). Open and closed circles represent the absence or presence, respectively, of the indicated component. For gel source data, see Supplementary Fig. 1.

acid removed and replaced with a fluorophore (Fig. 2e, bottom). *Ba1*Cas12a3 also cleaved in vitro-transcribed tRNAs that lack both chemical modifications and charged amino acids (Supplementary

Fig. 7a) and bulk tRNA isolated from yeast (Supplementary Fig. 7b). Notably, *Ap*Cas12a4 produced a different cleavage pattern of the tRNA pool (Fig. 2e), which suggests that the 3′ tRNA tail is not the preferred

substrate of this nuclease. Together, these findings indicate that activated Cas12a3 nucleases preferentially cleave the 3′ tail of tRNAs while sparing the bound RNA target, thereby further differentiating Cas12a3 from Cas12a2 and Cas12a4.

Among the diverse set of Cas nucleases, *Lsh*Cas13a is the only other nuclease reported to cleave tRNAs[22]. As *Lsh*Cas13a primarily cleaves its targeted transcript[30], and Cas12a3 exhibits minimal target RNA cleavage (Fig. 2c and Extended Data Fig. 2a), we sought to directly compare their activities. When targeting the same site in an expressed GFP transcript in TXTL assays, both *Lsh*Cas13a and *Ba1*Cas12a3 efficiently reduced GFP fluorescence (Extended Data Fig. 3a). However, quantitative PCR with reverse transcription (RT–qPCR) revealed that the *gfp* transcript underwent cleavage only by *Lsh*Cas13a (Extended Data Fig. 3b). These results show that *Lsh*Cas13a and *Ba1*Cas12a3 have different modes of action after target recognition, with *Lsh*Cas13a but not Cas12a3 substantially cleaving the target RNA.

Trimming tRNA tails by Cas12a3 would prevent tRNAs from participating in translation, thereby potentially driving global translational shutdown and arresting cell growth. Alternatively, growth arrest could be mediated by tRNA cleavage products that induce systemic stress responses, such as the stringent response, which is activated through the detection of deacetylated tRNAs bound to the ribosome by the RelA protein[38–41]. However, deleting *relA* did not impair plasmid interference by any of the tested Cas12a3 orthologues in *E. coli* (Extended Data Fig. 4). Therefore, Cas12a3-mediated immune defence operates independently of the stringent response mediated by RelA, which indicates that Cas12a3-mediated growth arrest results from either a different stress response or the direct disruption of translation.

## tRNAs are positioned by a loading domain

We confirmed that purified *Ba1*Cas12a3 strongly binds crRNA (dissociation constant ($K_d$) of about 0.2 nM), target RNA ($K_d$ of about 5 nM) and in vitro-transcribed tRNA$^{Ala(UGC)}$ when activated (*Ba1*Cas12a3, $K_d$ of about 20 nM; catalytically dead *Ba1*Cas12a3 (d*Ba1*Cas12a3), $K_d$ of about 6 nM) (Supplementary Figs. 8 and 9). Therefore, we used single-particle cryogenic electron microscopy (cryo-EM) to determine the 3.1 Å quaternary structure of *Ba1*Cas12a3 in complex with these three RNAs (Extended Data Table 1 and Supplementary Fig. 10). To limit tRNA cleavage, the complex was reconstituted on ice with a reduced Mg$^{2+}$ concentration. The resulting quaternary structure revealed that, similar to Cas12a2 from *Sulfuricurvum* sp. PC08-66 (*Su*Cas12a2)[28], *Ba1*Cas12a3 contains a REC lobe comprising REC1 (unresolved) and REC2 domains and a NUC lobe comprising a wedge (WED), PFS-interacting (PI), RuvC, ZR and an insertion domain (Fig. 3a,b). However, a portion of the insertion domain in *Ba1*Cas12a3 (residues 855–960) did not have discernible homology with *Su*Cas12a2 and had no structural homologues in the Protein Data Bank (PDB) database (as determined using Foldseek and Dali)[42,43]. We designate this fold the tRNA-loading domain (tRLD), as it interacts directly with the tRNA in the quaternary structure and probably facilitates tRNA loading into the RuvC nuclease domain (described below).

*Ba1*Cas12a3 interacts with crRNA in a similar way to Cas12a and Cas12a2, with the 5′ pseudoknot of the crRNA anchored between the RuvC and WED domains[28,44]. The target RNA forms an A-form duplex with the crRNA guide that traverses through the centre of the protein, and the 3′ PFS sequence is gripped by the PI domain, albeit through interactions distinct from those associated with *Su*Cas12a2 (Extended Data Fig. 5). The most notable feature of the *Ba1*Cas12a3 quaternary complex is the specific interaction with tRNA$^{Ala(UGC)}$ at two contact points: (1) the phosphate backbone of the T-arm, which is recognized through hydrogen-bonding interactions by a small loop in the REC2 domain (residues 251–257) (Fig. 3c and Extended Data Fig. 6a); and (2) the acceptor stem plus the 3′ CCA tail, which are clamped between the tRLD and RuvC domain (Fig. 3d and Extended Data Fig. 6b).

The tRLD binds the 3′ CCA tail through interactions that position the two cytosine bases adjacent to Y922 and stack the terminal adenosine between R902 and N924. Moreover, several electrostatic interactions between basic side chains (for example, K881 and K885) and peptide backbone amines with the tRNA phosphodiester backbone help to position the scissile phosphate in the RuvC endonuclease active site (Fig. 3d and Extended Data Fig. 6c). Notably, the 3′ hydroxyl group of the terminal adenosine points towards a vacant cavity, which would accommodate the respective charged amino acid (Extended Data Fig. 6d). Finally, the region of the tRNA bound by *Ba1*Cas12a3 is the same as that bound by the elongation factor Tu (EF-Tu) alone or in complex with the ribosome (Supplementary Fig. 11). This result suggests that *Ba1*Cas12a3 cleaves free tRNAs that are not actively engaged in protein synthesis.

To determine the role of the identified interactions in tRNA cleavage, we assessed the impact of mutations in both the tRNA and *Ba1*Cas12a3. Stepwise elimination of the tRNA loops did not impair cleavage of the acceptor stem or 3′ CCA tail in vitro (Fig. 3e). A minimal substrate comprising the anticodon loop, acceptor stem and 3′ CCA (h1) was also cleaved, although it bound d*Ba1*Cas12a3 with 62-fold lower affinity than the full-length tRNA (Extended Data Fig. 7). Using this truncated tRNA substrate, we next examined the impact of mutating the acceptor stem and the conserved 3′ CCA tail on its activity. *Ba1*Cas12a3 tolerated stem alterations provided the 3′ CCA was present (Fig. 3f and Extended Data Fig. 8a). Nonetheless, the catalytic efficiency was higher with a truncated tRNA than with linear RNAs, even when the CCA tail was located at the 3′ end (Extended Data Fig. 8a). In the CCA tail, *Ba1*Cas12a3 was sensitive to cytosine mutations, with transversions (C-to-G) impairing cleavage more strongly than transitions (C-to-U) (Fig. 3f and Extended Data Fig. 8b). By contrast, *Sm3*Cas12a3 did not cleave any substrates after any cytosine substitution (Fig. 3f), which suggests that these nucleases have divergent strategies for substrate recognition.

We next introduced mutations in *Ba1*Cas12a3 in the tRLD and the REC2 loop that interacts with the T-arm of the tRNA (Fig. 3g,h). Removing the tRLD (ΔtRLD) did not impair the overall secondary structure of the nuclease or binding of the crRNA and target RNA (Supplementary Fig. 12). However, removing this domain or mutating the residues that stack with the terminal adenosine (R902 or N924) substantially reduced both reporter silencing in TXTL assays and in vitro cleavage of different RNA substrates (Fig. 3g–i and Supplementary Fig. 13). Notably, the Y922A mutant increased cleavage activity in TXTL assays and in vitro with the truncated tRNA substrate (Fig. 3g,h), but not with non-tRNA substrates (Fig. 3i and Supplementary Fig. 13). This result suggests that Y922 has a more complex mechanistic role in *Ba1*Cas12a3. Mutation of the REC2 loop residues that interact with the T-arm impaired targeting in TXTL assays, whereby full-length tRNAs were cleaved, but only partially impeded cleavage of the truncated tRNA substrate lacking a T-arm (Fig. 3g,h and Supplementary Fig. 14). This finding indicates that T-arm interactions help position full-length tRNAs into the active site. Collectively, these data demonstrate that *Ba1*Cas12a3 preferentially cleaves free tRNAs by positioning the acceptor stem and the 3′ CCA tail of tRNAs into the RuvC active site through shape-specific and charge-specific interactions of tRNAs with the REC2 loop and tRLD.

## Structural changes enable tRNA binding

To gain deeper insights into the dynamics of *Ba1*Cas12a3 nuclease activation and tRNA cleavage, we determined the following cryo-EM structures: *Ba1*Cas12a3 bound to a crRNA (binary complex, 3.8 Å; Supplementary Fig. 15a,c–e); *Ba1*Cas12a3 bound to crRNA and target RNA (ternary complex, 3.9 Å; Supplementary Fig. 15b,f–h); and *Ba1*Cas12a3 bound to the cleaved 3′ ACCA tail from tRNA$^{Ala(UGC)}$ (post-cleavage complex, 3.3 Å; Supplementary Fig. 10c,g–i) (Fig. 4a–d). Together with the

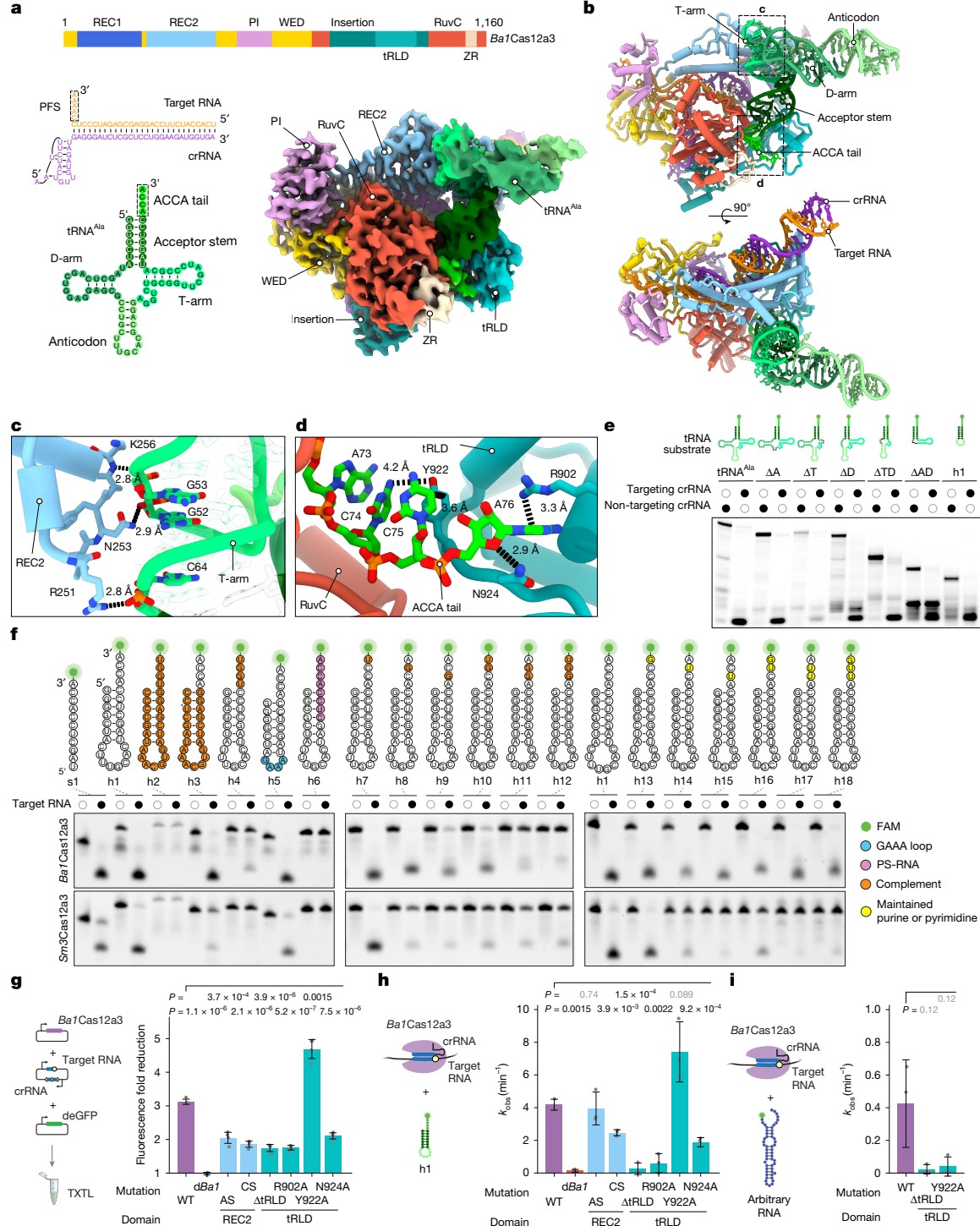

**Fig. 3 | Structural basis for tRNA capture by activated *Ba1*Cas12a3. a,** Overview of *Ba1*Cas12a3 bound to a crRNA, target RNA and tRNA[Ala(UGC)]. The REC1 domain in the cryo-EM structure was not resolved. **b,** Atomic model of the *Ba1*Cas12a3 quaternary complex. The anticodon region of the model was predicted using AlphaFold3 owing to the flexibility of the region. **c,** Interactions between the REC2 domain and the T-arm. **d,** Interactions between the tRLD domain and the 3′ CCA tail. **e,** Impact of truncating tRNA[Ala(UGC)] on in vitro cleavage by *Ba1*Cas12a3. A, anticodon; D, D-arm; T, T-arm. Open and closed circles represent the absence or presence, respectively, of the indicated component. **f,** Mutational analysis of the truncated tRNA substrate h1 as part of in vitro cleavage by *Ba1*Cas12a3 and *Sm3*Cas12a3. **g,** Schematic (left) and quantification (right) of the mutational analysis of tRNA recognition domains in *Ba1*Cas12a3 in TXTL reactions. The fold reduction in GFP fluorescence was calculated in comparison to a non-target control. The three residues shown in **d** plus a poorly resolved neighbouring

residue (K257) were swapped with alanine residues (AS; R251A, N253A, K256A and K257A) or with residues with an altered charge (CS; R251E, N253D, K256E and K257E). d*Ba1*, *Ba1*Cas12a3 with one RuvC active site mutation (E1032A); WT, wild type. **h,** Schematic (left) and quantification (right) of the mutational analysis of tRNA recognition domains in *Ba1*Cas12a3 as part of in vitro cleavage of the truncated tRNA substrate h1. d*Ba1*, *Ba1*Cas12a3 with three RuvC active-site mutations (D712A, E1032A and D1137A). **i,** Schematic (left) and quantification (right) of in vitro cleavage of an arbitrary RNA sequence by selected *Ba1*Cas12a3 mutants. Images in **e** and **f** are representative of independently prepared cleavage reactions (*n* = 3). Bars and error bars represent the mean ± s.d. of independently mixed TXTL or in vitro reactions. Statistical analysis was performed using two-tailed Welch's *t*-tests with all biological replicates (*n* = 4 for **g**, *n* = 3 for **h** and **i**). *P* values that are not significant (*P* ≥ 0.05) are shown in grey. For gel source data, see Supplementary Fig. 1.

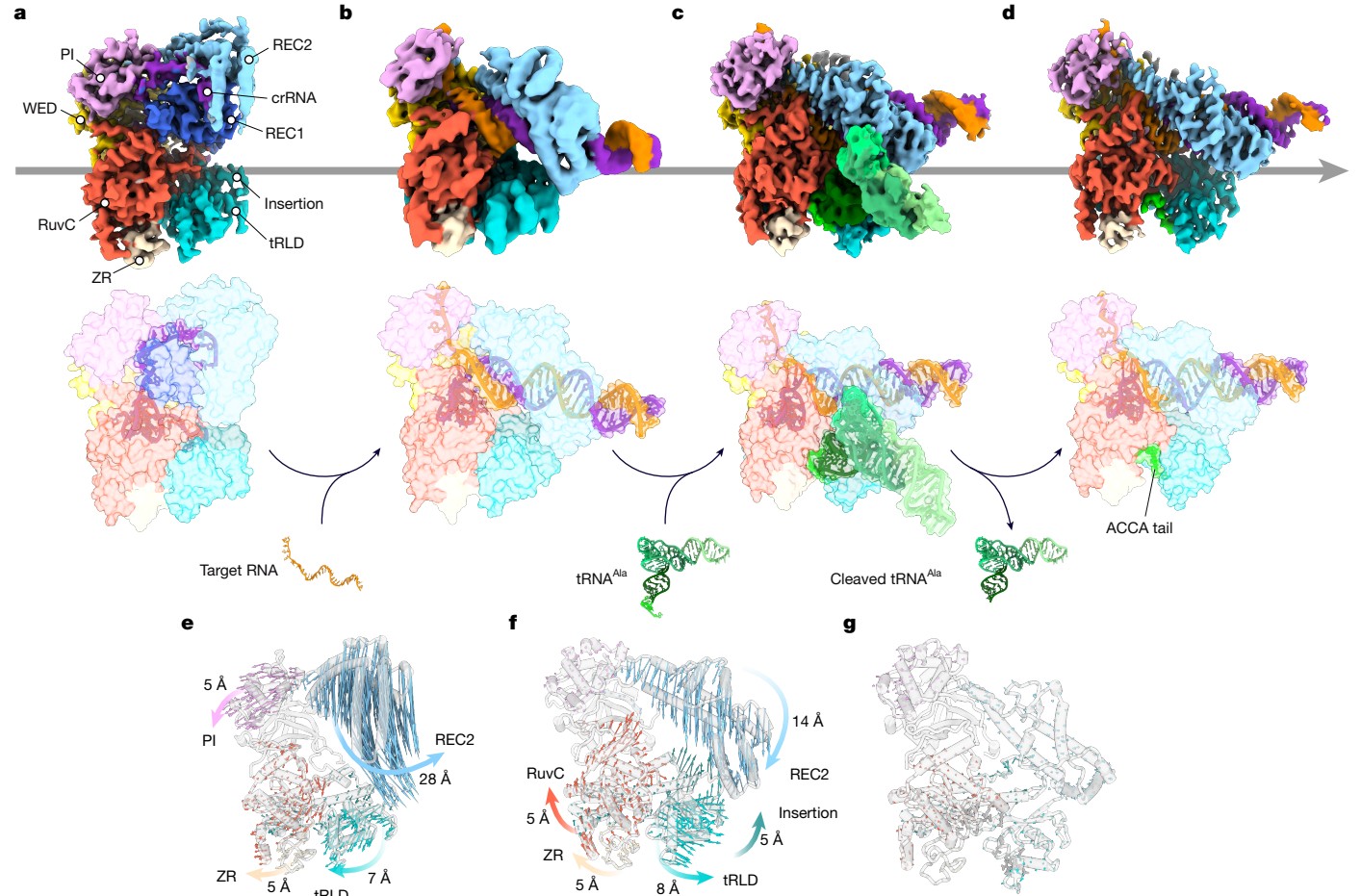

**Fig. 4 | Mechanism of tRNA binding and cleavage by *Ba1*Cas12a3. a**, Cryo-EM structure of the *Ba1*Cas12a3–crRNA binary complex. **b**, Cryo-EM structure of *Ba1*Cas12a3–crRNA–target RNA ternary complex. **c**. Cryo-EM structure of the *Ba1*Cas12a3–crRNA–target RNA–tRNA[Ala(UGC)] quaternary complex. **d**, Cryo-EM structure of the *Ba1*Cas12a3–crRNA–target RNA–tRNA tail quaternary complex in the post-cleaved state. **e**, *Ba1*Cas12a3 domain motion trajectories from the binary to the ternary complex. **f**, *Ba1*Cas12a3 domain motion trajectories from the ternary to the quaternary complex. **g**, *Ba1*Cas12a3 domain motion trajectories followed by release of the cleaved tRNA.

quaternary complex, which contains an uncleaved tRNA (Fig. 3b,c), these structures revealed a series of conformational changes that *Ba1*Cas12a3 undergoes to recognize a target RNA, selectively bind tRNAs and catalytically cleave the 3′ CCA tail of each tRNA (Fig. 4e–g and Supplementary Video 1).

Starting from the binary structure, target RNA binding drives multiple conformational changes in *Ba1*Cas12a3 and the bound crRNA, with the REC2 domain shifting by 28 Å to accommodate the guide–target duplex (Fig. 4e) and the 3′ end of the crRNA shifting 22 Å from its initial position bound to the PI domain (Supplementary Fig. 16). Although similar conformational changes are sufficient to fully activate *Su*Cas12a2 after target RNA recognition[28], in *Ba1*Cas12a3, the tRLD remains positioned near the RuvC active site. However, it has increased flexibility, as evidenced by the decrease in local resolution (Supplementary Fig. 15b). After cleavage of the tRNA tail by the RuvC domain (Fig. 4f, Supplementary Fig. 17 and Supplementary Video 1), the tRNA dissociates with the exception of four nucleotides comprising the tRNA tail (5′-ACCA-3′), which remain wedged between the RuvC and tRLD domains through contacts with R902, N924 and Y922 (Fig. 4d), consistent with the mapped cleavage sites across tRNAs (Fig. 2d and Supplementary Fig. 5). With the cleavage product bound, *Ba1*Cas12a3 retains its activated conformation (Fig. 4g), which indicates that another tRNA could be subsequently captured and cleaved without necessitating conformational resetting. Taken together, these structural snapshots reveal an activation pathway in which, after target

RNA recognition, *Ba1*Cas12a3 relies on its unique tRLD to direct cleavage of tRNA 3′ CCA tails.

## Cas12a3 expands multiplex RNA detection

An immediate application of the specific cleavage activity of Cas12a3 is multiplexed RNA detection[45,46]. Multiplexed RNA detection has been principally achieved by pairing Cas13 nucleases with orthogonal RNA substrates[23,47]. However, further expanding multiplexing requires new nucleases with orthogonal substrate preferences. To explore whether *Ba1*Cas12a3 can offer such an expansion, we adapted the minimal tRNA substrate as a reporter with a conjugated fluorophore and quencher[45]. A hairpin with a 3′ CCA tail and 3′ fluorophore produced the strongest signal (Fig. 5a,b). Notably, *Sm3*Cas12a3 and *ca23*Cas12a3 displayed distinct substrate preferences; however, *Ba1*Cas12a3 exhibited 226-fold greater activity than either orthologue (Fig. 5c and Supplementary Fig. 18a–c).

Using the optimal reporter for *Ba1*Cas12a3 (h25), we assessed multiplexing with Cas13a from *Leptotrichia wadei* (*Lwa*Cas13a) and Cas13b from *Prevotella* sp. MA2016 (*Psm*Cas13b), which primarily cleave short RNAs composed only of U or A nucleotides, respectively[23,48]. As these linear substrates do not contain the CCA sequence and Cas13 nucleases do not efficiently cleave structured RNAs, each nuclease preferentially cleaved its cognate substrate (Fig. 5c). Leveraging this specificity, we combined the three nucleases and their cognate probes, each labelled

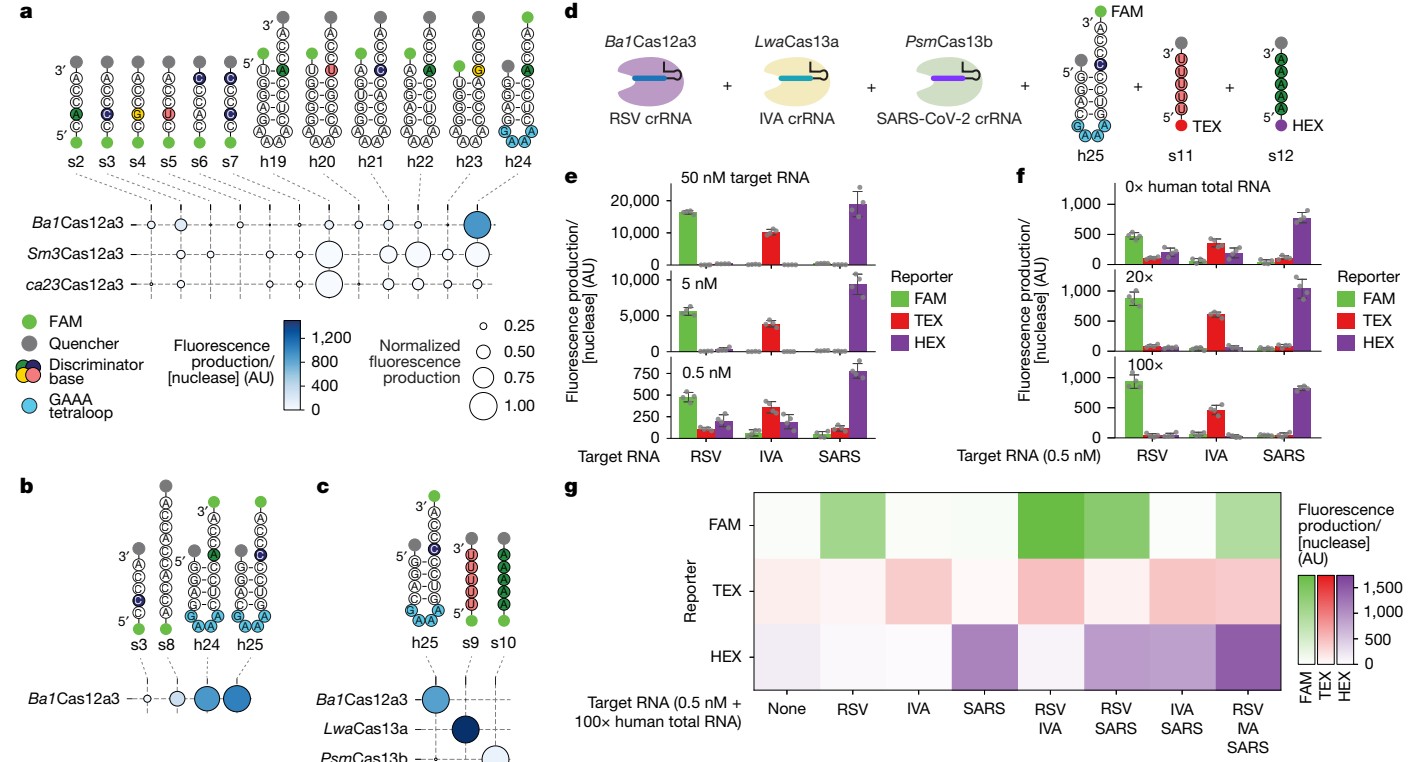

**Fig. 5 | Specific substrate recognition by *Ba1*Cas12a3 expands the scale of multiplexed RNA detection. a**, Fluorescence production after cleavage of FAM-labelled or quencher-tagged RNA substrates by *Ba1*Cas12a3, *Sm3*Cas12a3 and *ca23*Cas12a3. The area of each circle indicates fluorescence normalized to the highest value for each nuclease, whereas the colour of each circle represents absolute fluorescence production divided by the concentration of nuclease in the reaction. **b**, Fluorescence production after cleavage of additional FAM-labelled or quencher-tagged RNA substrates by *Ba1*Cas12a3. **c**, Fluorescence production after cleavage of FAM-labelled or quencher-tagged RNA substrates specific to *Ba1*Cas12a3, *Lwa*Cas13a or *Psm*Cas13b. **d**, Schematic of the combined components for multiplexed one-pot detection. In this setup, each fluorophore

(FAM, TEX and HEX) can be independently quantified. **e**, Impact of varying the applied concentration of detected RNAs derived from respiratory syncytial virus (RSV), influenza virus A (IVA) and SARS-CoV-2 (SARS) as part of the one-pot setup. **f**, Impact of adding human total RNA on one-pot RNA detection. Ratios are based on molarity, assuming an average RNA length of 2 kb in the human total RNA. **g**, Multiplexed, one-pot RNA detection in the presence of an excess of human total RNA. Dots depict individual measurements of independently prepared reactions, whereas bars and error bars represent the mean ± s.d. of independent measurements (*n* = 4). Fluorescence measurements depicted as circles or heatmaps represent the average of independent cleavage reactions (*n* = 3 or 4 in **a** and **b**, *n* = 4 in **e** and **f**, *n* = 6 in **g**).

with a distinct fluorophore, into a one-pot reaction for multiplexed RNA detection. This one-pot setup enabled the separate and combinatorial detection of RNA transcripts derived from the respiratory viruses SARS-CoV-2, respiratory syncytial virus (RSV) and influenza A (Fig. 5d–g and Supplementary Fig. 18d–f). Notably, the presence of a large excess of human total RNA did not interfere with, but even enhanced, detection (Fig. 5f and Supplementary Fig. 18e). These findings show that Cas12a3 nucleases can be readily incorporated into multiplexed detection assays alongside widely used Cas13 platforms[45].

## Discussion

In this work, we have reported the discovery of crRNA-guided Cas12a3 nucleases that recognize complementary RNA targets and, in response, cleave the conserved 3′ tails of tRNAs to induce growth arrest and block phage dissemination. Integrating genetic, biochemical, sequencing and structural studies, we propose a model (Extended Data Fig. 9) in which the nuclease undergoes a large conformational change that can then bind free tRNAs through multiple sequence-specific and shape-specific contacts. The unique tRLD in the nuclease positions the tRNA tail next to the RuvC active site to cleave off the tail, with extensive cleavage of cellular tRNAs leading to growth arrest. This immune response does not depend on a traditional stringent response but instead probably arises directly from translational inhibition or from a RelA-independent stress response activated by the tRNA cleavage products. Despite the

high binding preference for tRNAs in vitro, it also remains possible that the nuclease targets additional RNAs not present in our TXTL system, which may further contribute to immune defence.

Cas12a3 adds to a growing set of immune defences and cellular processes that inactivate tRNAs[1–12,22,49–52]. The anticodon loop is a common target of bacterial defences, with sequence-specific recognition of the loop beyond the anticodon required for translation. In response, some phages encode variant tRNAs that escape cleavage, thereby replenishing the tRNA pool and sustaining phage propagation[53]. Cleavage of the universally conserved 3′ CCA tail of tRNAs by Cas12a3 represents a distinct strategy that cannot be readily circumvented by viral tRNAs. To overcome this defence mechanism, viruses would require other, yet unknown, means of resistance. tRNA tail cleavage has been implicated with other cellular processes[51,54], but so far, has not been directly linked to phage defence. Cas12a3 therefore could represent a previously unrecognized yet widely used phage defence mechanism based on preferential tRNA tail cleavage.

The discovery of RNA-mediated tRNA cleavage in CRISPR–Cas systems reflects the rich functional diversity of antiviral defences. Although numerous defence strategies remain uncharacterized[55,56], our work reveals that even well-known families can have previously unknown functions. In particular, Cas12a2, Cas12a3 and Cas12a4 are phylogenetically related to DNA-targeting Cas12a nucleases[27] but exhibit functions that more closely resemble those of RNA-targeting Cas13 nucleases[57]. Cas12a3 is particularly notable as a Cas nuclease that

can specifically direct its cleavage activity towards a distinct substrate while sparing the bound target, whereas Cas12a4 also exhibits RNA-activated RNA cleavage but functionally deviates from Cas12a3. These findings warrant further investigation. Together, Cas12a2, Cas12a3 and Cas12a4 seem to be hyper-evolvable, with domain alterations dictating whether the RuvC site cleaves broadly across multiple substrates (for example, Cas12a2) or preferentially against specific targets such as tRNA tails (for example, Cas12a3). Deviations are also possible in these clades, as illustrated by the different tRNA cleavage sites and substrate requirements of *Ba1*Cas12a3 and *Sm3*Cas12a3. These findings underscore the broader diversity of Cas12 nucleases and indicate the existence of additional, yet to-be-discovered functions.

Exploring the hidden functional space of Cas nucleases has the potential to further expand the CRISPR toolbox. Cas12a3 in particular is an important addition for multiplexed RNA detection. The targeted cleavage of tRNA tails by Cas12a3 led us to the generation of a structured reporter specifically targeted by Cas12a3 but ignored by Cas13 nucleases, which enabled us to independently detect three distinct RNA biomarkers in a one-pot setup. Given the availability of other Cas12a3 and Cas13 orthologues that recognize distinct substrates[58] (Fig. 5b), we anticipate that even more RNA biomarkers can be independently detected in a single reaction. Complementary advances in signal post-amplification and rendering CRISPR-based tests compatible with point-of-care settings could further enhance the diagnostic utility of Cas12a3. Beyond molecular diagnostics, the ability to inactivate most tRNAs through cleavage of their universally conserved 3′ CCA tail suggests applications in other areas, such as transcript-dependent cell arrest, viral suppression or selective cell elimination without direct DNA damage. Finally, our structural insights into PFS recognition and tRNA engagement could provide a foundation for engineering Cas12a3 with expanded target access and orthogonal substrates. Together, these advances position Cas12a3 as a versatile addition to CRISPR technologies and a prime candidate for enhancement through protein and guide RNA engineering.

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

## Methods

### Orthologue identification, phylogenetic analysis and sequence alignment

We used previously identified Cas12a2 protein sequences[27] as queries for tBLASTn and BLASTp searches in the NCBI databases (https://www.ncbi.nlm.nih.gov) and the JGI Integrated Microbial Genomes and Microbiomes database (https://img.jgi.doe.gov) to identify closely related orthologues. The resulting amino acid sequences, along with Cas12a orthologues used as an outgroup, were aligned using Clustal Omega[59]. The trimmed alignment generated using ClipKIT[60] was then used to reconstruct a phylogeny using IQ-TREE (v.2.3.6) (-m MFP -T 8 -B 1000)[60,61] with a maximum-likelihood approach. Assignment of putative domains and conserved amino acid residues, as shown in Fig. 1a,b and Extended Data Fig. 1, was performed with reference to *Su*Cas12a2 (refs. 27,28). The amino acid sequences of the nucleases are provided in Supplementary Data 1. Information on each nuclease, including contig accession numbers, source organisms and the presence of spacer acquisition genes (*cas1*, *cas2* and *cas4*), is provided in Supplementary Table 1. The presence of these genes was determined using DefenseFinder (v.2.0.1) with defense-finder-models (v.2.0.2)[62]. CRISPR arrays were identified using CRISPRCasFinder (v.4.2.21)[63].

### Strains, plasmids and oligonucleotides

Strains, plasmids and oligonucleotides used in this study are listed in Supplementary Table 2. Nuclease-encoding sequences were codon-optimized for expression in *E. coli* and synthesized by Twist Bioscience, unless stated otherwise. DNA oligonucleotides and FAM-labelled reporters were synthesized by Integrated DNA Technologies.

### PFS screen in *E. coli*

A PFS-containing plasmid library (CBS-6873) was constructed by incorporating a target sequence (CAO1: 5′-CAUCAAGCCUUCCUUCAGGU GUUGCUCCA-3′) followed by 1,024 combinations of five randomized nucleotides (NNNNN). Thus, the target, placed under the PJ23119 promoter (https://parts.igem.org/Promoters/Catalog/Anderson) was cloned into a low-copy sc101 plasmid (around 5 copies per cell), which was then amplified using the primers ODpr23 and ODpr24 (Supplementary Table 2), with the forward primer including a 5-nucleotide randomized overhang. After DpnI treatment to remove template DNA, the resulting PCR product was ligated and electroporated into *E. coli* TOP10, which produced >2 million transformants (about 2,000-fold library coverage). The PFS preferences of Cas12a, Cas12a2, Cas12a3 and Cas12a4 orthologues was assessed by targeting the CBS-6873 library with a CAO1-targeting crRNA plasmid (CBS-6875), using a non-targeting crRNA plasmid (CBS-6876) as a control. The nucleotide-encoding sequences were codon-optimized for *E. coli* and expressed under a T7 promoter, whereas crRNAs were driven by the PJ23119 promoter. *E. coli* BL21(AI) cells with the nuclease and crRNA plasmids were electroporated in three separate reactions, each using around 500 ng of library DNA in 50 μl competent cells recovered in LB with 0.1 mM isopropyl β-D-1-thiogalactopyranoside (IPTG) and 0.2% L-arabinose and grown overnight to produce about 2 million transformants (>2,000-fold library coverage). Plasmids were then purified using a ZymoPURE II Plasmid Midiprep kit (D4201). Reactions for each experimental condition were carried out in duplicate.

Purified plasmids from both target and non-target conditions were first PCR-amplified using the primers ODpr55 and ODpr56 (Supplementary Table 2) with KAPA HIFI HotStart polymerase (KK2601) for 20 cycles at 64.5 °C following the manufacturer's protocol. After amplification, these PCR products were purified using AMPureXP beads (Beckman Coulter, A63880) and subsequently indexed for Illumina sequencing using standard indexing primers with KAPA HIFI HotStart polymerase (KK2601) for 8 cycles at 61.5 °C with 2 μM forward and reverse primers and 5 ng μl⁻¹ DNA. The resulting indexed PCR products were sequenced on an Illumina NovaSeq 6000 (paired-end, 150 bp reads), which ensured that at least 2 million reads per sample were sequenced. Raw FASTQ files were processed with Trimmomatic (v.0.39)[64] using the following parameters: ILLUMINACLIP:TruSeq3-PE.fa:2:30:10, LEADING:3, TRAILING:3, and SLIDINGWINDOW:4:15. Paired-end reads were merged using BBMerge (qtrim=t, trimq=10, minlength=20)[64,65]. Sequences containing motifs matching "TTCCTTCAGGTGTTGCTCCA (.....) GGTGAGTTCT", corresponding to the 20-nucleotide target-encoding sequence and the 10-nucleotide downstream sequence, were extracted, excluding any sequences with ambiguous bases (N) or Phred scores below 20. Depletion scores were then calculated using the formula: depletion = (sum(non-target)/sum(target)) × (count(target)/count(non-target)). The $\log_2$ fold change values for these scores were computed for the nucleotides at PFS positions (+1 to +5), and scatterplots visualizing the PFS preferences were generated using Matplotlib in Python.

### Plasmid clearance in *E. coli*

To test plasmid clearance, we expressed *E. coli* codon-optimized orthologues (as in the PFS screen) from plasmids (Supplementary Table 2). Target RNA and crRNA were co-expressed from a single plasmid under separate PJ23119 promoters (plasmids CBS-6177–6182 in Supplementary Table 2 include CAO1, GAPDH and GFP targets with both target and non-target crRNAs). *E. coli* BL21(AI) cells with these target–guide constructs were electroporated with 1 μl of high-purity 50 ng μl⁻¹ nuclease plasmid. Cells were then recovered in LB with 0.2% L-arabinose and 1 mM IPTG at 37 °C for 1 h with shaking without antibiotics. Serial tenfold dilutions (up to 10⁻¹ in 1× PBS) were then prepared, and 5 μl of each dilution was spotted onto LB agar plates containing 0.2% L-arabinose, 0.1 mM IPTG and the appropriate antibiotics, including 50 μg ml⁻¹ kanamycin for selection of the nuclease plasmid alone (assessing growth arrest) or 50 μg ml⁻¹ kanamycin plus 25 μg ml⁻¹ chloramphenicol for co-selection (assessing plasmid clearance). Plates were incubated overnight at 37 °C. Experiments were performed in four biological replicates.

### *relA* deletion in *E. coli*

Deletion of the *relA* gene in *E. coli* BL21(AI) was performed using λ Red recombineering as previously described[66]. To initiate recombineering, wild-type *E. coli* BL21(AI) cells were transformed with plasmid pKD46 (Supplementary Table 2), recovered and incubated overnight at 28 °C on LB agar containing ampicillin. The kanamycin resistance (*kanR*) cassette was PCR-amplified from plasmid pKD13 using the primers ODpr1652 and ODpr1653, which included 50 bp homology arms targeting sequences immediately upstream and downstream of the *relA* locus (positions 2,715,617–2,717,851 on GenBank accession CP047231), which encodes GTP pyrophosphokinase (NP_311671.1). The resulting PCR product was electroporated into *E. coli* cells with pKD46, which had been induced with 0.2% L-arabinose to express the λ-RED recombination machinery. Cells were plated on LB agar containing kanamycin and incubated at 37 °C to select for successful recombinants. To remove the inserted *kanR* cassette, plasmid pCP20 (Supplementary Table 2), which encodes FLP recombinase, was transformed into the strain. The cassette was excised via recombination at the flanking FRT sites, and pCP20 was subsequently cured by incubation at 37 °C. Successful deletion of *relA* was verified by colony PCR using the primers ODpr1655 and ODpr1654 (Extended Data Fig. 4), and by confirming the absence of growth on kanamycin-containing plates. Four independent mutants were generated and used to perform plasmid clearance (Extended Data Fig. 4), with the GAPDH T17 target under targeting and non-targeting conditions, using the plasmids CBS-6179 and CBS-6180 (Supplementary Table 2), respectively.

### SOS response induction in *E. coli*

To quantify the *recA*-dependent SOS response, *E. coli* BL21 (AI) cells were co-transformed with the carbenicillin-resistant reporter P_{recA}-GFP

(CBS-3611) or a noGFP control (CBS-3616) along with the nuclease and target–guide plasmids, performed as previously described[27]. The resulting overnight cultures were grown in LB containing 100 μg ml$^{-1}$ carbenicillin, 50 μg ml$^{-1}$ kanamycin, 25 μg ml$^{-1}$ chloramphenicol and 0.2% glucose to repress nuclease expression. The next day, 1 ml of each culture was pelleted (5,000$g$, 2 min), resuspended in fresh LB without antibiotics and adjusted to an OD$_{600}$ of 0.1. Then, 20 μl of each culture was added to 180 μl fresh medium to obtain a final concentration of 0.2% L-arabinose and 0.1 mM IPTG in a 96-well plate (final OD$_{600}$ = 0.01). Samples were incubated at 37 °C with vigorous shaking in a BioTek Synergy H1 plate reader, with OD$_{600}$ and GFP fluorescence (excitation of 485/20 nm, emission of 528/20 nm) recorded. Each condition was tested in four biological replicates.

### Protection against phage T4 infection
Fresh *E. coli* BL21 (AI) cells at OD$_{600}$ of 0.3 containing the respective nuclease (CBS-7169 and CBS-7171) and guide (CBS-7181-7183) plasmids (Supplementary Table 2) were grown in LB medium supplemented with 50 μg ml$^{-1}$ kanamycin, 25 μg ml$^{-1}$ chloramphenicol, 0.2% L-arabinose and 0.1 mM IPTG to induce nuclease expression. These cultures were challenged with different multiplicities of infection of T4 phage and incubated at 37 °C with vigorous shaking overnight. The next day, the cultures were serially diluted in tenfold increments in SM buffer (50 mM Tris-HCl, 100 mM NaCl and 8 mM MgSO$_4$, pH 7.5). For plaque assays, control *E. coli* BL21 (AI) cells with an empty carbenicillin-resistant plasmid were mixed with 0.75% top agar and poured onto LB agar plates containing 100 μg ml$^{-1}$ carbenicillin. Serial dilutions of the overnight cultures were then spotted onto the top agar, and the plates were incubated overnight at 37 °C to allow plaque formation. Each experimental condition was performed in three biological replicates. GFP-targeting crRNA-expression plasmid (CBS-7184) was used as a non-target control (Supplementary Table 2).

### In vitro determination of activating targets and cleavage substrates
RNAs were in vitro-transcribed from linear dsDNA templates containing a T7 promoter (Supplementary Table 2). Templates for the GAPDH T17 and T19 crRNAs were generated by PCR amplification of ssDNA oligonucleotides using the primers ODpr1220 and ODpr1221, and ODpr1220 and ODpr1223, respectively. The T17 target RNA template was amplified from plasmid CBS-7159 using the primers ODpr1212 and ODpr1213. In vitro transcription was performed using a HiScribe T7 High Yield RNA Synthesis kit (NEB, E2040) following the manufacturer's instructions. The generated target RNA along with the complementary crRNA were also used for cryo-EM.

For the in vitro experiment, recombinant nucleases were purified primarily as previously described[27,28]. In brief, the proteins were expressed in *E. coli* BL21(DE3) star grown in LB medium and induced at an OD$_{600}$ of about 0.6 with 0.1 mM IPTG, followed by an overnight incubation at 18 °C. Cells were then collected, lysed by sonication in lysis buffer and clarified by centrifugation at 30,000$g$ for 30 min at 4 °C. The soluble fraction was then incubated with Ni-NTA resin (pre-equilibrated in lysis buffer) for 30 min at 6–8 °C, and the column was washed with an IMAC washing buffer supplemented with 2 M NaCl before bound proteins were eluted with imidazole-containing buffer. The eluate was diluted with low-salt buffer and further purified by ion-exchange chromatography on a HiTrap Heparin column using a NaCl gradient. Pooled fractions were concentrated and polished by size-exclusion chromatography on a HiLoad Superdex 200 column. The final purified fractions were pooled, concentrated, flash-frozen in liquid nitrogen and stored at −80 °C.

Preference for targets and cleavage substrates was evaluated in vitro using nucleases purified as described above. Reactions contained 500 nM of the respective nuclease, an equimolar amount of T19 GAPDH non-target crRNA and T17 GAPDH target crRNA, and T17 GAPDH target

at 50 nM provided either as RNA, ssDNA or dsDNA, as well as respective non-target substrates at 1 μM (Supplementary Table 2). For assays with *Ba1*Cas12a3 Y922A and ΔtRLD (Supplementary Fig. 13), reactions contained 100 nM nuclease, crRNA and target RNA together with 5 μM RNA substrate library. Reactions were performed in a buffer comprising 40 mM Tris-HCl (pH 7.5), 50 mM NaCl and 1 mM DTT, with MgCl$_2$ at 2 mM for Cas12a2, Cas12a3 and Cas12a4 nucleases and 10 mM for Cas12a. Fluorescence, indicative of reporter cleavage via separation of the fluorophore and quencher, was monitored at 29 °C on a BioTek Synergy H1 plate reader (excitation of 492 nm, emission of 520 nm). For the heatmap in Fig. 1f, initial reaction rates in the linear range were normalized for each nuclease by setting the highest rate across reporters to one. All experiments were performed in at least three biological replicates.

### Nuclease activity assay in TXTL reactions
Activity of the purified *Ba1*Cas12a3 and *Mp*Cas12a2 nucleases was assessed using the TXTL system (Arbor Biosciences, 540300). *Ba1*Cas12a3 (25 nM) and *Mp*Cas12a2 (50 nM) were individually pre-incubated with 37.5 nM and 75 nM crRNA, respectively, for 15 min at 29 °C in 40 mM Tris-HCl (pH 7.5), 50 mM NaCl and 1 mM DTT, with 2 mM MgCl$_2$. After this pre-incubation step, 50 nM GAPDH T17 RNA target was added and the mixtures were further incubated for 15 min at 29 °C to allow the formation of ribonucleoprotein particle (RNP) complexes. The assembled RNPs were then added to the TXTL reaction and incubated for 4 h at 29 °C. Thereafter, the deGFP plasmid supplied with the TXTL kit was added at 0.5 nM. In parallel, the RNP and the deGFP plasmid were simultaneously introduced into the TXTL mix that had already been pre-incubated for 4 h at 29 °C. GFP signal production was subsequently monitored on a BioTek Synergy H1 plate reader set to an excitation of 485 nm and an emission of 528 nm. All experiments were performed in at least three biological replicates.

### RT–qPCR analysis
TXTL reactions were carried out using myTXTL (Arbor Biosciences, 540300) supplemented with the pCBS420 plasmid constitutively expressing deGFP and the pCBS11 p70-T7RNAP plasmid (Supplementary Table 2). For *Lsh*Cas13a, the non-targeting plasmid CBS-3620 and the *gfp* targeting plasmid CBS-3622 expressing both the nuclease and crRNA were used; for *Ba1*Cas12a3, the non-targeting plasmid CBS-7245 and the targeting plasmid CBS-7246 were used (Supplementary Table 2). Final concentrations were 1 nM nuclease and crRNA co-encoded plasmids, 0.5 nM pCBS420, 0.33 nM p70-T7RNAP and 0.66 mM IPTG. Fluorescence was monitored over 16 h on a Synergy H1 plate reader (BioTek) at 485 nm excitation and 528 nm emission. For the monitoring of fluorescence, 3 μl volume of TXTL was used, whereas for RNA extraction, the reactions were scaled to 30 μl and sampled after 4 h, when strong GFP reduction under targeting conditions was evident. All experiments were performed in three biological replicates.

Following TXTL, reactions were treated with 10 μl proteinase K for 15 min at room temperature and RNA was purified using a RNA Clean & Concentrator-25 kit (Zymo Research, R1017) with on-column DNase I digestion. RT–qPCR was performed using an iTaq Universal SYBR Green One-Step kit (Bio-Rad, 172-5150) on a CFX Opus 384 Real-Time PCR system (Bio-Rad) with CFX Maestro (v.5.3.022.1030). Primer pair 1 (ODpr1662 and ODpr1663) and primer pair 2 (ODpr1670 and ODpr1671) were validated using a dilution series of RNA from TXTL expressing *gfp* from plasmid pCBS420 (Supplementary Table 2), which produced efficiencies of 94.7% and 97.4%, respectively. Each primer was used at 300 nM. Cycling conditions included a 63 °C annealing–extension step, with other parameters following the manufacturer's protocol. Melting curve analysis confirmed single amplicons for both primer pairs. No amplification was detected in no-RT controls after 30 cycles. Each condition was tested in three biological replicates, with three technical replicates per sample.

## Nanopore direct RNA sequencing and analysis

*Ba1*Cas12a3 and *Sm3*Cas12a3 nucleases (250 nM each) were preincubated in 40 mM Tris-HCl (pH 7.5), 50 mM NaCl, 1 mM DTT and 2 mM MgCl$_2$ for 15 min at 29 °C with equimolar amounts of GAPDH T19 non-target crRNA and GAPDH T17 target crRNA. Next, GAPDH T17 target RNA was added to a final concentration of 250 nM and the mixture was incubated for at least 15 min to form the RNP complex. The assembled RNP was then added to 30 µl TXTL reactions. After 4 h of incubation at 29 °C, 6 µl of proteinase K (NEB, P8107) was added and the reactions were incubated for 15 min. RNA was purified from the reactions using a RNA Clean & Concentrator-25 kit (R1017, Zymo Research). To remove full-length rRNAs, RNAs with lengths of ≤200 nucleotides were isolated with a miRNeasy Tissue/Cells Advanced Micro kit (217684, Qiagen) according to the manufacturer's instructions. Subsequently, tRNAs in the purified RNA were deacylated by incubation in 90 mM Tris-HCl (pH 9.0) at 37 °C for 30 min. The RNA was next recovered using a RNA Clean & Concentrator-25 kit with 1.3× ethanol concentration after the RNA prep buffer step. All reactions were performed in triplicate.

For the in vitro total *E. coli* tRNA cleavage reactions, *Ba1*Cas12a3, S*m3*Cas12a3 and *ca23*Cas12a3 nucleases (750 nM each) were combined in 40 mM Tris-HCl (pH 7.5), 50 mM NaCl, 1 mM DTT and 2 mM MgCl$_2$ buffer with an equimolar amount of crRNAs and incubated for 15 min at 37 °C. Next, an equimolar amount of target RNA was added and the mixture was incubated for an additional 15 min to allow RNP complex formation. Total tRNAs purified from *E. coli* MRE600 (10109541001, Roche) were then added to the final concentration of 1.5 µM. The reactions were subsequently incubated at 37 °C for 2 h followed by proteinase K treatment and purification using a RNA Clean & Concentrator-5 kit (R1013, Zymo Research). Finally, the recovered RNAs were deacylated as described for the TXTL samples. All reactions were performed in triplicate.

For each sample, 10 µg of total RNA was polyadenylated using *E. coli* poly(A) polymerase (NEB) in a 20 µl reaction containing 50 mM Tris-HCl pH 8, 250 mM NaCl, 10 mM MgCl$_2$, 1 U µl$^{-1}$ RNasin, 1 mM ATP and 0.25 U µl$^{-1}$ poly(A) polymerase for 30 min at 37 °C. Polyadenylated RNAs were purified using a RNA Clean & Concentrator-5 kit (Zymo Research) following the manufacturer's instructions. Multiplexed Nanopore direct RNA sequencing was then performed following a previously described strategy[67]. In brief, custom barcode containing RTA adapters (bc03, bc04 and bc011; Supplementary Table 2) were annealed at a final concentration of 10 µM in 30 mM HEPES-KOH (pH 7.5) and 100 mM potassium acetate by incubation for 1 min at 95 °C followed by a slow cooling to 25 °C at a rate of 0.5 °C min$^{-1}$. The annealed RTA adapters were then diluted to 0.7 µM and stored at −20 °C until use. Next, 100 ng of polyadenylated RNA was ligated to 1 µl of 0.7 µM annealed RTA in the presence of 2 µl of 5× NEBNext Quick ligation buffer (NEB), 0.5 µl of high-concentration T4 DNA Ligase (M0202M, NEB) and 0.2 µl of RNAsin (Promega) in a final volume of 10 µl by incubation at 22 °C for 15 min. To stop the ligation process, 2 µl of 0.5 M EDTA was added and samples were pooled and purified with 0.5 volumes of SPRI beads (Mag-Bind TotalPure NGS, Omega Bio-tek), washed twice with 200 µl fresh 80% ethanol and eluted in 10 µl RNase-free H$_2$O. RT was performed by adding 8 µl of 5× Induro buffer, 2 µl of 10 mM dNTP, 2 µl of 10 µM random hexamer oligonucleotides, 2 µl Induro RT (NEB), 0.5 µl RNasin in a final volume of 40 µl and incubation 2 min at 22 °C (annealing), 15 min at 60 °C (RT) and 10 min at 70 °C (heat-inactivation). The reaction was purified with 1.4 volumes of SPRI beads, washed twice with 200 µl of fresh 80% ethanol and eluted in 10 µl RNase-free water. Sequencing adapter ligation was performed using a SQK-RNA004 library kit (ONT) according to the manufacturer's instructions. In brief, 10 µl of purified library was mixed with 6 µl 5× NEBNext Quick ligation buffer, 5 µl of RLA sequencing adapter (ONT SQK-RNA004) and 3 µl of T4 DNA Ligase high concentration in a final volume of 30 µl and

incubated 15 min at 22 °C. The reaction was purified with 0.5 volumes of SPRI beads and washed twice with 100 µl RNA wash buffer (WSB, ONT SQK-RNA004). The library was then eluted in 32 µl RNA elution buffer (REB, ONT SQK-RNA004). For sequencing, the libraries were loaded onto a PromethION (FLO-PRO004RA) flow cell for RNA samples originating from the TXTL reactions or onto a MinION (FLO-MIN004RA) flow cell for RNA samples originating from the in vitro reactions. Data acquisition was performed using MinKNOW (v.24.11). Basecalling was performed with dorado (v.0.8.3) using the sup model. Barcode classification was performed using the demux command of warpdemux (v.0.4.4) with the WDX12_rna004_v0_4_4 model[67]. Unaligned bam files were then demultiplexed based on barcode labels using calibrated target performance scores.

Non-coding RNA sequence references for mapping the Nanopore reader were identified in the *E. coli* BL21(DE3) genome (CP010816) for TXTL analysis and the *E. coli* MRE600 genome (CP014197) for the analysis of the in vitro tRNA cleavage experiments. In silico identification were performed using covariance models from Rfam[68] for tRNA (RF00005), tmRNA (RF00023), 5S rRNA (RF00001), 16S rRNA (RF00177) and 23S rRNA (RF02541) with searches executed using cmsearch[69]. The tRNA hits were subsequently annotated using tRNAscan-SE (v.2.0)[70] to generate the final set of tRNA sequences for analysis. The resulting reference fasta files are provided in Supplementary Data 2 and 3.

The poly(A) tail sequences were trimmed and the reads subsequently mapped to the reference using BWA-MEM (v.0.7.17)[71] with the minimum seed length of 19 (-k 19) and the minimum alignments score of 30 (-T 30). The -M option was enabled to mark shorter split hits as secondary alignments. The resulting SAM files were converted to BAM files, sorted and indexed using SAMtools (v.1.9)[72]. Reads with a mapping quality (MAPQ) below 20 were filtered out before downstream analyses.

Depletion scores were computed for each nucleotide position according to the formula: depletion = (sum(non-target)/sum(target)) × (count(target)/count(non-target)). In this equation, 'sum' denotes the total number of reads mapped to a reference sequence (that is, the overall read count) under either non-target or target condition, whereas 'count' refers to the per–nucleotide coverage (the number of reads overlapping that specific position). This normalization procedure produces a relative measure of depletion at each nucleotide position.

The *z* scores for each nucleotide position were calculated using the per-nucleotide depletion score, the overall mean and standard deviation for each sequence reference according to the formula: *z* = (nucleotide depletion score − reference mean)/reference standard deviation. For visualization purposes, the *z* scores were then constrained to a range of 0 to 2.

## Fluorescence in vitro tRNA cleavage

Total purified *E. coli* tRNAs (Roche, 10109541001) were labelled at the 5′ end with Cy5 using a Vector Labs kit (MB-9001) according to the manufacturer's protocol, with Cy5 maleimide monoreactive dye (GEPA15131, Sigma-Aldrich). For 3′ end labelling, total *E. coli* tRNAs and *Saccharomyces cerevisiae* tRNA (Life Technologies, 0000010468) were first deacylated in 90 mM Tris-HCl (pH 9.0) at 37 °C for 30 min. Up to 400 pmol of RNA was then combined with 1 mM ATP, 100 µM pCp-Cy5 (NU-1706-CY5, Jena Bioscience), 1 µl T4 RNA ligase (M0202, NEB) and 1 µl of 100% DMSO in the reaction buffer supplied with the ligase. The reaction was incubated overnight at 16 °C. In vitro-transcribed tRNAs were labelled at the 3′ end using the same protocol.

To generate in vitro-transcribed tRNAs, DNA templates were first prepared by PCR using the following primer pairs: ODpr1343 and ODpr1344 for tRNA[Lys(UUU)]; ODpr1345 and ODpr1346 for tRNA[Ala(UGC)]; ODpr1347 and ODpr1348 for tRNA[Ser(GGA)]; and ODpr1349 and ODpr1350 for tRNA[Tyr(GUA)]. In vitro transcription was then carried out using a HiScribe T7 High Yield RNA Synthesis kit (NEB, E2040) according to the manufacturer's instructions.

To generate truncating tRNA$^{Ala(UGC)}$, self-annealing PCR reactions were performed using the following primer pairs: ODpr1711 and ODpr1712 for ΔA; ODpr1715 and ODpr1716 for ΔT; ODpr1713 and ODpr17134 for ΔD; ODpr1717 and ODpr1718 for ΔTD; ODpr1719 and ODpr1720 for ΔAD; and ODpr1739 and ODpr1740 for h1 (Supplementary Table 2). In vitro transcription was performed using a HiScribe T7 High Yield RNA Synthesis kit (NEB, E2040) according to the manufacturer's instructions, using up to 1 µg of PCR template.

For the in vitro digestion assay, the nucleases MpCas12a, Ba1Cas12a3, Sm3Cas12a3, ca23Cas12a3 and ApCas12a4 were each used at 250 nM, combined with either equimolar GAPDH T19 non-target or T17 target crRNAs. To activate the proteins, 10 nM target RNA was added, followed by the addition of Cy5-labelled tRNA substrates at 50 nM. The reactions were performed in 40 mM Tris-HCl (pH 7.5), 50 mM NaCl, 1 mM DTT and 2 mM MgCl$_2$ and incubated for 2 h at 37 °C. Reaction products were subsequently analysed by electrophoresis on a 10% polyacrylamide gel containing 7 M urea. All reactions were performed in triplicate.

### Northern blotting

For northern blot analysis, TXTL and the in vitro tRNA cleavage reactions were performed as for Nanopore sequencing. After purification, 10 µg of each RNA sample from the TXTL reactions and 1 µg from the in vitro total purified E. coli tRNA reactions were resolved on an 8% polyacrylamide gel containing 7 M urea at 300 V for 2 h and 25 min using a gel transfer system (Doppel-Gelsystem Twin L, PerfectBlue). Using an electroblotter with an applied voltage of 50 V for 1 h at 4 °C (Tank-Elektroblotter Web M, PerfectBlue), the RNA was transferred onto Hybond-XL membranes (Sigma-Aldrich, 15356-1EA), crosslinked with UV-light for a total of 0.12 Joules (UV-lamp T8C; 254 nm, 8 W) and hybridized overnight in 15 ml of Roti-Hybri-Quick buffer at 42 °C with 5 µl γ-[$^{32}$P]-ATP end-labelled oligodeoxyribonucleotides (5 pmol µl$^{-1}$). Probe ODp7 was used for tRNA$^{Ala(UGC)}$, probe ODp17 for tRNA$^{Leu(UAG)}$ and probe ODp6 for 5S rRNA (Supplementary Table 2). The labelled RNA was visualized with a Phosphorimager (Typhoon FLA 7000, GE Healthcare).

### Fluorophore–quencher in vitro cleavage

LwaCas13a protein was obtained from GenCRISPR (Z03486), whereas purified PsmCas13b was provided by M. Kaminski's Laboratory. Target RNA and crRNAs for Cas12a3 nucleases were prepared as described above. For PsmCas13b, PCR amplification was performed using the complementary primers ODpr1448–1449 to generate the GAPDH T17 target crRNA template, and the primers ODpr1450–1451 for the non-target (T19) crRNA template. Similarly, for LwaCas13a, the corresponding crRNA templates were generated by PCR using the primers ODpr1424–1425 for the target and the primers ODpr1426–1427 for the non-target crRNAs. The respective RNAs were in vitro-transcribed using a HiScribe T7 High Yield RNA Synthesis kit (NEB, E2040).

Reactions were performed in 40 mM Tris-HCl (pH 7.5), 50 mM NaCl, 1 mM DTT and 2 mM MgCl$_2$ using nuclease-specific reporters as listed in Supplementary Table 2. For tests with tRNA-mimicking substrates using Cas12a3 nucleases, Ba1Cas12a3 (125 nM), Sm3Cas12a3 (500 nM) and ca23Cas12a3 (500 nM) were each combined with 500 nM of either target or non-target crRNA, 50 nM of GAPDH T17 RNA target and 1 mM of fluorescent RNA reporters. In experiments assessing orthogonality, Ba1Cas12a3 (100 nM), LwaCas13a (10 nM) and PsmCas13b (250 nM) were used with an equimolar amount of target or non-target crRNA, with target RNA added at 50 nM and fluorescent RNA reporters at 1 mM. Reactions were carried out at 29 °C in BioTek Synergy H1 plate reader with fluorescence monitored at an excitation of 492 nm and emission of 520 nm. Fluorescence production rates were determined in the initial linear range, normalized for each nuclease, and analysed as a function of nuclease concentration. All experiments were performed at least in triplicate.

For catalytic efficiency calculations with substrates S13, S14 and h25, reactions contained 50 nM Ba1Cas12a3, crRNA and target RNA, together with fluorophore–quencher reporters at 5,000, 2,500, 1,250, 625 or 312.5 nM. Reactions were performed in a buffer containing 40 mM Tris-HCl (pH 7.5), 50 mM NaCl, 1 mM DTT and 2 mM MgCl$_2$. Fluorescence was monitored over time, and rates of fluorescence increase were background-corrected using the corresponding non-target controls. Rates were fitted to the Michaelis–Menten equation to extract kinetic parameters, and catalytic efficiency was determined as $k_{cat} K_m^{-1}$. Each condition was tested in at least three independent replicates.

To compare truncated tRNA substrates with mutations in the 3′ CCA tail, reactions contained 100 nM Ba1Cas12a3, crRNA and target RNA together with 1 µM reporters h25–h29 in buffer (40 mM Tris-HCl pH 7.5, 50 mM NaCl, 1 mM DTT and 2 mM MgCl$_2$). Fluorescence was monitored over time, with each condition tested in at least three independent replicates.

DNA templates for viral RNA targets were generated by PCR using the primers ODpr1684 and ODpr1685 for SARS-CoV-2 (N gene), ODpr1686 and ODpr1687 for influenza A (NP gene), and ODpr1690 and ODpr1691 for RSV (N gene) (Supplementary Table 2). crRNA templates were generated by PCR as follows: Ba1Cas12a3 non-targeting control (ODpr1692 and ODpr1693); Ba1Cas12a3 RSV N target (ODpr1692 and ODpr1738); LwaCas13a non-target (ODpr1424 and ODpr1696); LwaCas13a influenza A NP target (ODpr1424 and ODpr1697); PsmCas13b non-target (ODpr1701 and ODpr1702); and PsmCas13b SARS-CoV-2 N target (ODpr1709 and ODpr1710) (Supplementary Table 2). All RNAs were transcribed in vitro using a HiScribe T7 High Yield RNA Synthesis kit (NEB, E2040).

Purified Ba1Cas12a3, LwaCas13a and PsmCas13b were each assembled with their cognate crRNAs at 100 nM and tested with target RNA ranging from 50 nM to 0.5 nM. Universal Human Reference RNA (Life Technologies, QS0639) was added at 10 or 50 nM, calculated assuming an average RNA length of 2 kb. Reporters were S11 (TEX fluorophore) for LwaCas13a, S12 (HEX) for PsmCas13b and h25 for Ba1Cas12a3. For one-pot reactions, nucleases, crRNAs and fluorescent reporters were first combined in a single mixture. Target RNAs were then added to initiate the reactions. Fluorescence was monitored over time. Initial reaction velocities were calculated in the linear range. All reactions were performed in at least four independent replicates.

### Microscale thermophoresis measurements

Microscale thermophoresis measurements were performed as previously described[73] in buffer containing 40 mM Tris-HCl (pH 7.5), 50 mM NaCl, 2 mM MgCl$_2$, 1 mM DTT and 1.66% glycerol. To pre-assemble the activated ternary complex, the purified components were mixed at equimolar ratios before preparing the dilution series. For all measurements, a 16-point 2-fold serial dilution series was prepared, starting at an initial concentration of 2.5 µM of unlabelled components. Each dilution was subsequently mixed in equal volumes with a constant concentration of Cy5-labelled RNA. Samples were transferred into premium-coated capillaries (MO-K025, NanoTemper Technologies), and microscale thermophoresis measurements were conducted using a Monolith Pico instrument (NanoTemper Technologies). Data analysis, including determination of $K_d$ values, was performed using NanoTemper Analysis software and plotted using Origin software (OriginLabs).

### Sample preparation for cryo-EM

Carboxy-terminal His-tagged Ba1Cas12a3 (CBS-7171) was overexpressed in E. coli (DE3) star in 4 l of LB medium. The cell culture was induced with 0.1 mM IPTG at an OD$_{600}$ of 0.6 and further grown at 18 °C for 18 h. Cell pellets were collected by centrifugation and resuspended in 100 ml lysis buffer (50 mM Tris pH 7.5, 300 mM NaCl, 2 mM MgCl$_2$, 10 mM imidazole and 10% glycerol) treated with 1 tablet of protease inhibitor cocktail (11873580001, Roche). Cells were disrupted by sonication (Amplitude 58%; 1 s on and 8 s off for 30 min total). The soluble fraction was obtained by centrifugation and incubated with 5 ml (bed volume)

nickel resin rotating for 1 h at 4 °C. Subsequently, the purification procedure was conducted in a 20-ml gravity column. Nickel resin was washed with 50 ml wash buffer A (50 mM Tris pH 7.5, 2 M NaCl, 2 mM $MgCl_2$, 10 mM imidazole and 10% glycerol) and 50 ml of wash buffer B (50 mM Tris pH 7.5, 50 mM NaCl, 2 mM $MgCl_2$, 10 mM imidazole and 10% glycerol) sequentially and eluted with nickel-elution buffer (50 mM Tris pH 7.5, 50 mM NaCl, 2 mM $MgCl_2$, 300 mM imidazole and 10% glycerol). Nickel peak elution fractions were collected and diluted 10 times with low-salt buffer (50 mM Tris pH 7.5, 20 mM NaCl, 2 mM $MgCl_2$ and 10% glycerol). Protein samples were then loaded onto a HiTrap SP HP cation-exchange chromatography column using low-salt buffer and eluted with a gradient of high-salt buffer (50 mM Tris pH 7.5, 1.0 M NaCl, 2 mM $MgCl_2$ and 10% glycerol). Peak elution fractions were pooled and loaded onto Superdex 200 increase 10/300 GL (GE Healthcare) using SEC buffer (50 mM Tris pH 7.5, 150 mM NaCl, 2 mM $MgCl_2$ and 5% glycerol). Peak fractions were then pooled and concentrated, and flash-frozen in aliquots in liquid nitrogen.

The tRNA for cryo-EM was produced using the T7 RNA polymerase-mediated run-off method[74]. In detail, the in vitro transcription reaction was performed in a 500 µl volume containing DNA template, T7 RNA polymerase and reaction buffer (20 mM Tris, pH 8.0, 5 mM DTT, 150 mM NaCl, 8 mM $MgCl_2$, 2 mM spermidine, 20 mM NTPs, RNasin and pyrophosphatase). The reaction was performed at 37 °C overnight and followed by DNase I treatment to remove DNA templates. The product was then purified using a DEAE column and heat treatment at 80 °C for 2 min and followed by the slow cooling process to room temperature for the re-annealing process. To obtain a homogenous tRNA population, the samples were subjected to a Superdex 75 Increase gel filtration column, and the tRNA-containing fractions were pooled and stored at −80 °C. For microscale thermophoresis measurements, the internally Cy5-labelled in vitro-transcribed *E. coli* tRNAs were produced as described above, with an additional 5% of Cy5-CTP introduced in the reaction.

To obtain the *Ba1*Cas12a3 quaternary complex, crRNA was pre-treated by heating at 65 °C for 3 min and slowly cooled down to room temperature. RNA was then added to *Ba1*Cas12a3 in a 1:1.2 molar ratio. The complex was then incubated at room temperature for 10 min. The pure *Ba1*Cas12a3 binary complex was purified via Superdex 200 Increase 10/300 GL in buffer without $MgCl_2$ to inhibit nuclease activity (50 mM Tris pH 7.5, 50 mM NaCl, 1 mM DTT and 5% glycerol). Target RNA and tRNA^Ala were sequentially added to 50 µl of the $Mg^{2+}$-free peak fraction of the binary complex (about 20 µM) at 1:1.2 and 1:2 molar ratios, respectively. The target RNA was incubated at room temperature for 10 min, followed by the addition of tRNA, and the mixture was kept on ice for 30 min before vitrification.

For the binary complex dataset, the *Ba1*Cas12a3 binary complex was prepared as above but with 2 mM $MgCl_2$ present (50 mM Tris pH 7.5, 150 mM NaCl, 1 mM DTT, 2 mM $MgCl_2$ and 5% glycerol). The binary complex sample was vitrified in the presence and absence of 0.02% (w/v) fluorinated octyl maltoside to obtain more particles with different orientations.

For the ternary complex dataset, target RNA was added into the *Ba1*Cas12a3 binary complex (50 mM Tris-HCl pH 7.5, 150 mM NaCl, 1 mM DTT, 2 mM $MgCl_2$ and 5% glycerol) in a 1:1.2 molar ratio. The ternary complex sample was vitrified in the presence of 1 mM ATPγS to obtain more views of particle distribution.

### Cryo-EM data acquisition and processing

Samples were vitrified on cryo-EM grids (Quantifoil R 2/1 300 mesh, Au). In brief, 3.5 µl of the sample was applied onto glow-discharged grids. The grids were blotted for 4 s (blot force: −9; 100% humidity; 4 °C) and plunge-frozen in liquid ethane using a Vitrobot Mark IV (FEI). The vitrified specimens were imaged on a Thermo Fisher Glacios Cryo-TEM operating at 200 kV and equipped with the Falcon 4i Direct Electron Detector camera, an energy filter slit width of 10 eV and a C2 aperture

of 50 µm. The videos were recorded in counting mode using Thermo Fisher Scientific EPU software (v.3.7.1) with the total dose of 40 $e^-$ $A^{-2}$ with a nominal magnification of 130,000-fold, corresponding to a pixel size of 0.91 Å.

Motion correction, contrast transfer function estimation, particle picking and 2D classification were carried out on-the-fly using CryoSPARC live (v.4.4.0 and v.4.6.0)[28,75]. The particles selected from the 2D classification of the live session were divided into small splits and the junk particles were removed again using 2D classification. The remaining particles were further classified by using heterogeneous refinement, and the best 3D classes were subjected to non-uniform refinement. Overall, gold-standard resolution Fourier shell correlation using the threshold of 0.143 and local resolution was calculated by local resolution estimation. The statistics for data collection and processing are provided in Extended Data Table 1.

### Model building, refinement and figure preparation

All models corresponding to the *Ba1*Cas12a3 binary, ternary and quaternary complexes were originally generated by AlphaFold3 (ref. 76) and used as initial models for model building. Models were built in ChimeraX (v.1.7)[77], and interactive refinements against cryo-EM maps were performed using ISOLDE with restraints for secondary-structure elements of the AlphaFold3-predicted structures[77,78]. The resulting models were further refined with real-space refinement and validated in Phenix (v.1.20.1)[79]. Statistics for the final models are described in Extended Data Table 1.

All structural figures were generated using Chimerax (v.1.7). For structural comparison, the structures from binary to quaternary were aligned against the *Ba1*Cas12a3 WED domain. The domain motions from other individual domains were shown with arrows in different domain colours, and the arrows from each two compared models were created by using the PDB-arrows script with slight modification[80].

### Generation and purification of *Ba1*Cas12a3 mutants

The *Ba1*Cas12a3 mutant constructs were generated either by Q5 PCR site-directed mutagenesis followed by KLD or by Gibson assembly, using the C-terminal 6×His-tagged WT *Ba1*Cas12a3 plasmid as a template (CBS-7171; Supplementary Table 2). Each plasmid was transformed into *E. coli* BL21 (DE3) cells, grown on an agar plate and selected by antibiotic resistance selection. A single colony was used to inoculate 60 ml of LB medium, grown at 37 °C at 200 rpm overnight (16–18 h). Next, 20 ml of the overnight growth was used to inoculate 1 litre of LB medium containing 100 µg ml$^{-1}$ kanamycin. Cells were grown at 37 °C at 200 rpm until an $OD_{600}$ of 0.5–0.6 was reached. The cells were then cold shocked on ice for 20 min before inducing with 0.1 mM IPTG, followed by 16–18 h of growth at 18 °C. Cell pellets were collected by centrifugation and stored at −80 °C. Other nucleases were expressed using the same procedure.

For in vitro cleavage experiments shown in Fig. 3f,h,i, Extended Data Fig. 2 and Supplementary Figs. 11 and 13, *Ba1*Cas12a3, mutants thereof, as well as *Ap*Cas12a4, *Mp*Cas12a2 and *Sm3*Cas12a3, were purified similarly to the method used for *Su*Cas12a2 (refs. 27,28). In brief, the cell pellets were thawed on ice before lysis by sonication in a lysis buffer (25 mM Tris pH 8.5, 500 mM NaCl, 10 mM imidazole, 2 mM $MgCl_2$ and 10% glycerol) containing protease inhibitors (2 µg ml$^{-1}$ aprotinin, 10 µM leupeptin and 1.0 µg ml$^{-1}$ pepstatin) and 1 mg ml$^{-1}$ lysozyme. The lysate was clarified by centrifugation and added to 5 ml of Ni-NTA resin and batch bound at 4 °C for 30 min, and then washed with 300 ml wash buffer (25 mM Tris pH 8.5, 2 M NaCl, 10 mM imidazole, 2 mM $MgCl_2$ and 10% glycerol). The protein was eluted with 25 ml Ni-elution buffer (25 mM Tris pH 8.5, 500 mM NaCl, 250 mM imidazole, 2 mM $MgCl_2$ and 10% glycerol). Fractions containing *Ba1*Cas12a3, as determined by SDS–PAGE, were desalted using a Hiprep 26/10 desalting column into low-salt buffer (25 mM Tris pH 8.5, 50 mM NaCl, 2 mM $MgCl_2$ and 10% glycerol). *Ba1*Cas12a3 was

applied to a Hitrap Heparin HP cation-exchange column and eluted using a gradient of high-salt buffer (25 mM Tris pH 8.5, 1 M NaCl, 2 mM $MgCl_2$ and 10% glycerol). The fractions containing $Ba1$Cas12a3 were concentrated using a 50 kDa MWKO concentrator to about 1 ml and loaded onto a Hiload 16/600 Superdex 200 pg size-exclusion column using SEC buffer (25 mM HEPES pH 8.5, 150 mM KCl, 2 mM $MgCl_2$ and 5% glycerol). Fractions containing the desired proteins were concentrated and stored at −80 °C.

### In vitro cleavage of truncated tRNAs

Reactions were made by combining 300 nM crRNA with 250 nM CRISPR-associated enzyme in DTT containing low-salt buffer (40 mM Tris-HCl pH 7.5, 50 mM NaCl, 2 mM $MgCl_2$ and 1 mM DTT) and incubated at room temperature for 15 min. Next, 100 nM of FAM-labelled tRNA substrate was then added followed by 250 nM of target RNA to initiate the reaction. The reaction was performed at 37 °C for 1 h. Reactions were quenched with phenol and nucleic acid was purified by acid phenol–chloroform extraction. FAM-labelled nucleic acid was analysed using 12% urea–PAGE and visualized for fluorescein fluorescence.

For determining the cleavage efficiency of the $Ba1$Cas12a3 TRL mutants, 140 µl reactions were performed with 150 nM $Ba1$Cas12a3, 100 nM guide RNA, 100 nM FAM 26-nucleotide structured tRNA substrate (Supplementary Table 2) and 25 nM target RNA. Next, 20-µl aliquots were extracted and quenched at 10 s, 30 s, 1 min, 2 min, 5 min and 10 min at 37 °C. Nucleic acid was quenched and visualized as described above.

### Orthologue target and substrate preference assay

For determining the nucleic acid targets for $Ap$Cas12a4, $Mp$Cas12a2, $Sm3$Cas12a3 and $Ba1$Cas12a3 (Extended Data Fig. 2), 250 nM crRNA and a given nuclease (250 nM) were incubated in a DTT containing low-salt buffer (40 mM Tris-HCl pH 7.5, 50 mM NaCl, 2 mM $MgCl_2$ and 1 mM DTT) at room temperature for 15 min. Then 100 nM of FAM-labelled target (RNA, ssDNA or dsDNA) was added to initiate the reaction. The reaction was performed at 37 °C for 1 h before quenching the reaction with phenol and extracting the nucleic acid using phenol–chloroform. The samples were analysed using a previously described denaturing FDF–PAGE[81] and visualized for fluorescein fluorescence.

### $Ba1$Cas12a3 target electromobility shift assay

To determine whether the $Ba1$Cas12a3 ΔtRLD mutant could still bind target RNA, an electromobility shift assay was performed. To control the amount of binary (protein–guide) complex in the reaction mixture, 5:1 $Ba1$Cas12a3 to guide RNA concentration ratio was prepared. An initial protein–guide solution was prepared and serial diluted into four reaction concentrations in an EDTA-containing buffer (25 mM HEPES (pH 7.2), 150 mM KCL and 100 mM EDTA). The reaction tubes ranged from 10 nM to 10 µM for $Ba1$Cas12a3 and 2 nM to 2 µM for the crRNA. The FAM-labelled target RNA concentration was held constant at 100 nM. The samples were run on a 6% TBE polyacrylamide gel and visualized for fluorescein fluorescence.

### Protein-folding analysis using circular dichroism spectroscopy

To determine the overall structural components of $Ba1$Cas12a3 ΔtRLD compared with WT $Ba1$Cas12a3, far-UV circular dichroism (CD) was performed using a Jasco-J1500 spectropolarimeter. In brief, 0.36 mg ml$^{-1}$ of the respective protein was prepared in CD buffer (10 mM $K_2HOP_4$ and 50 mM $Na_2SO_4$ pH 8.74) and CD spectra were obtained from 260 to 190 nm using a scanning speed of 50 nm min$^{-1}$ (with a 2-s response time and accumulation of three scans). The CD signal was converted to molar ellipticity using Jasco Spectra Manager software.

### Reporting summary

Further information on research design is available in the Nature Portfolio Reporting Summary linked to this article.

## Data availability

The Illumina-based PFS screen data and the direct RNA Nanopore sequencing reads have been deposited into the European Nucleotide Archive under accession code PRJEB88250 (https://www.ebi.ac.uk/ena/browser/view/PRJEB88250). Models and associated cryo-EM maps have been deposited into the Electron Microscopy Data Bank (EMD) and PDB databases with the following accession codes: $Ba1$Cas12a3 binary complex (EMD-52275; PDB: 9HLX); $Ba1$Cas12a3 ternary complex (EMD-52287; PDB: 9HM6); $Ba1$Cas12a3 quaternary complex at pre-cleavage state (EMD-52285; PDB: 9HM4); and $Ba1$Cas12a3 quaternary complex at post-cleavage state (EMD-52286; PDB: 9HM5). Raw gel images are included as Supplementary Fig. 1. Source data are provided with this paper.

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

**Acknowledgements** We thank Ł. Koziej for processing of the initial cryo-EM datasets, S. Schmelz for support in cryo-EM, A. Gatzemeier for assistance in the purification of d$Ba1$Cas12a3, R. Rarose for support with the in vitro RNA experiments, M. Kaminski for providing purified $Psm$Cas13b protein, L. Schönemann for protein purification, and C. Krempl and S. Backesfor providing the RSV and influenza A transcript-encoding plasmids. This work was supported through funding by the European Research Council (101001394 to S.G.; 865973 and 101158249 to C.L.B.), the R. Gaurth Hansen Family (to R.N.J.), the National Institutes of Health (R35GM138080 to R.N.J.), the PostDoc Plus Program from the Graduate School of Life Sciences at Julius-Maximilians-Universität Würzburg (to O.D.), and the Deutsche Forschungsgemeinschaft (DFG, German Research Foundation) under Germany's Excellence

Strategy–The Berlin Mathematics Research Center MATH+ (EXC–2046/1, project ID: 390685689 to M.v.K.).

**Author contributions** Conceptualization: O.D., B.Y., J.P.K.B., K.T.C., D.W.H., R.N.J. and C.L.B. Discovery of tRNA degradation: O.D. Bioinformatic and phylogenetic analyses: O.D. Experimentation in vitro, in *E. coli* and in TXTL: O.D., K.T.C., M.K. and X.C. Nanopore sequencing and analyses: O.D., A.-S.G.-B., W.v.d.T. and R.P.S. Biochemistry and biophysics: K.T.C., O.D., J.S.N., A.C.-G., S.G. and B.F. Northern blot analyses: T.A. and O.D. Structure solution, model building and structural analyses: B.Y., D.W.H., K.T.C., J.P.K.B. and O.D. Supervision: S.G., M.v.K., D.W.H., R.N.J., and C.L.B. Writing original draft: O.D., B.Y., D.W.H., R.N.J. and C.L.B. Visualization: O.D., B.Y., K.T.C., R.N.J. and C.L.B. Writing, review and editing: all authors. Funding acquisition: S.G., R.N.J., M.v.K., O.D. and C.L.B.

**Funding** Open access funding provided by Helmholtz-Zentrum für Infektionsforschung GmbH (HZI).

**Competing interests** O.D., K.T.C., J.P.K.B., R.N.J. and C.L.B. have filed patent applications on related technologies. W.v.d.T., M.v.K. and R.S. have filed patent applications on the warpdemux technology used as part of direct RNA sequencing. C.L.B. is a co-founder and scientific advisor of Locus Biosciences, a co-founder and officer of Leopard Biosciences, and scientific advisor to Benson Hill. The other authors declare no conflicts of interest.

**Additional information**
**Correspondence and requests for materials** should be addressed to Dirk W. Heinz, Ryan N. Jackson or Chase L. Beisel.

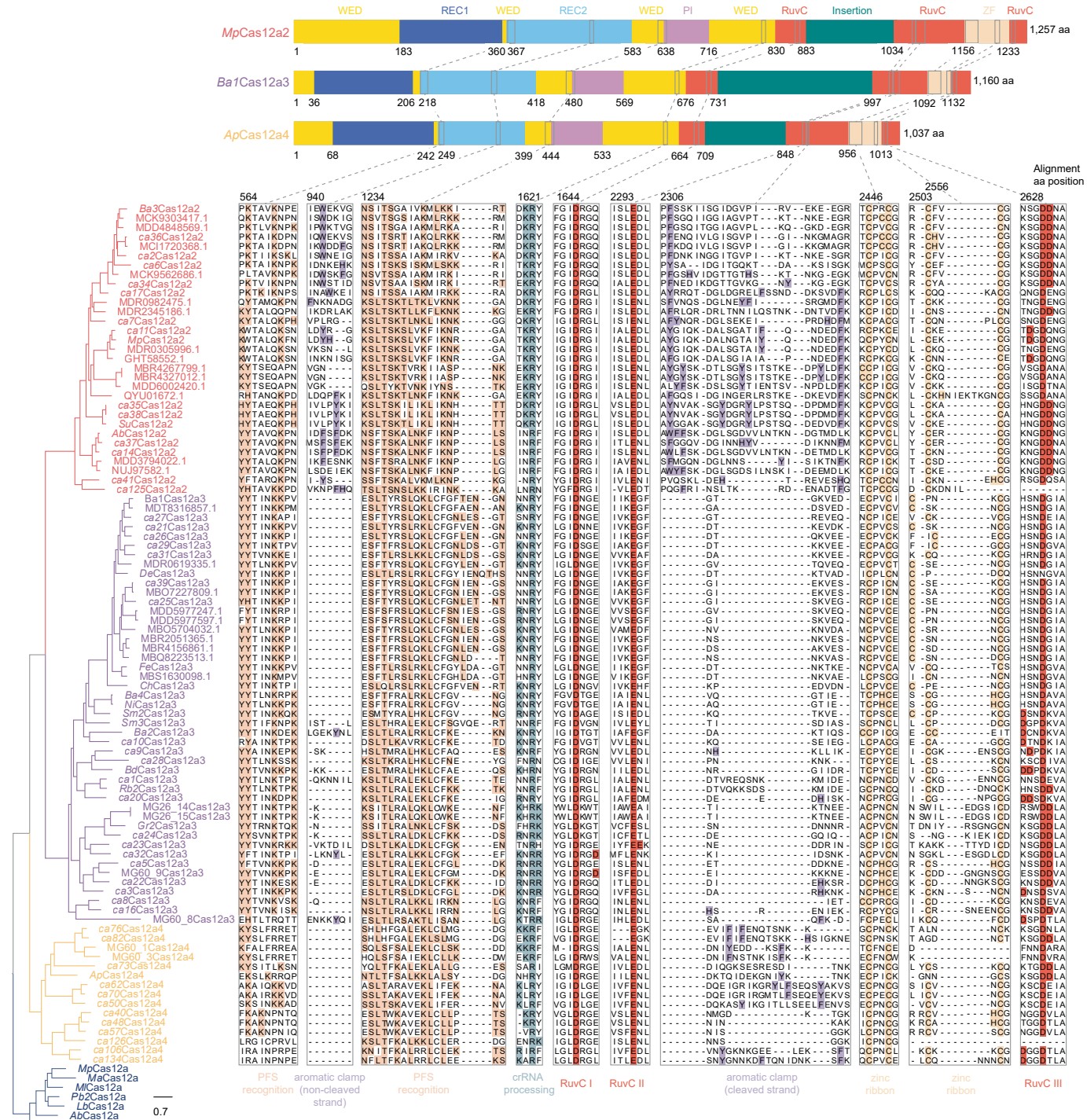

**Extended Data Fig. 1 | Predicted domain organization of *Mp*Cas12a2, *Ba1*Cas12a3, and *Ap*Cas12a4, together with an approximate maximum-likelihood phylogeny of Cas12a (dark blue), Cas12a2 (red), Cas12a3 (purple), and Cas12a4 (yellow) amino acid sequences as well as amino acid sequence alignment of Cas12a2, Cas12a3, and Cas12a4.** The phylogenetic tree was constructed using IQ-TREE with ultrafast bootstrapping (1,000 replicates). The scale bar at the bottom indicates the expected number of substitutions per site. The corresponding simplified phylogeny is shown in Fig. 1a. The alignment highlights key motifs and residues, including those involved in protospacer-flanking sequence (PFS) recognition, the aromatic clamp regions interacting with cleaved and non-cleaved strands, CRISPR RNA (crRNA) recognition, RuvC domain motifs, and zinc-ribbon sites. A subset of the amino acid alignment is shown in Fig. 1b.

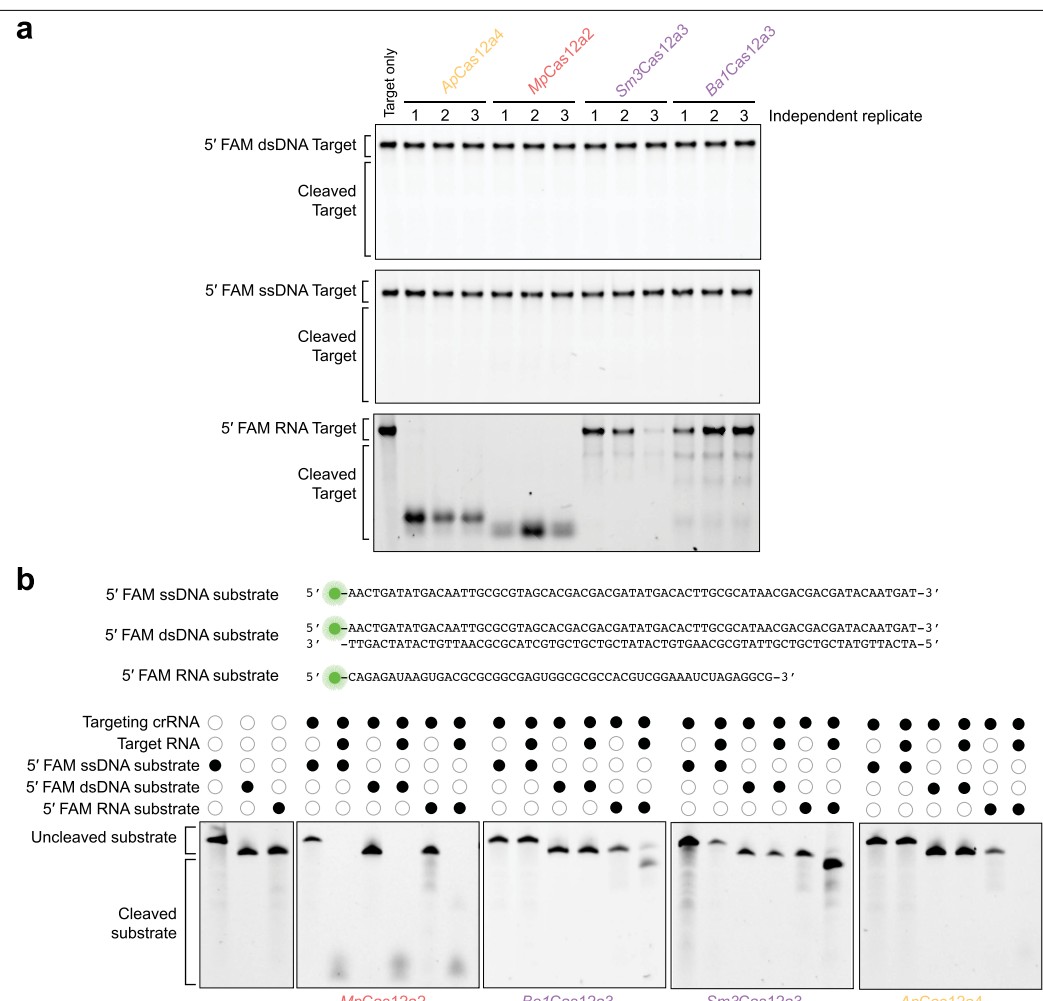

**Extended Data Fig. 2 | Cas12a2, Cas12a3, and Cas12a4 are RNA-targeting enzymes with distinct substrate cleavage profiles.** (**a**) In vitro target cleavage using FAM-labeled dsDNA, ssDNA, or RNA resolved on a fully-denaturing formaldehyde (FDF) polyacrylamide gel. (**b**) In vitro cleavage of FAM-labeled non-target substrates. Gel images are representative of independent cleavage assays (n = 3). For gel source data, see Supplementary Fig. 1.

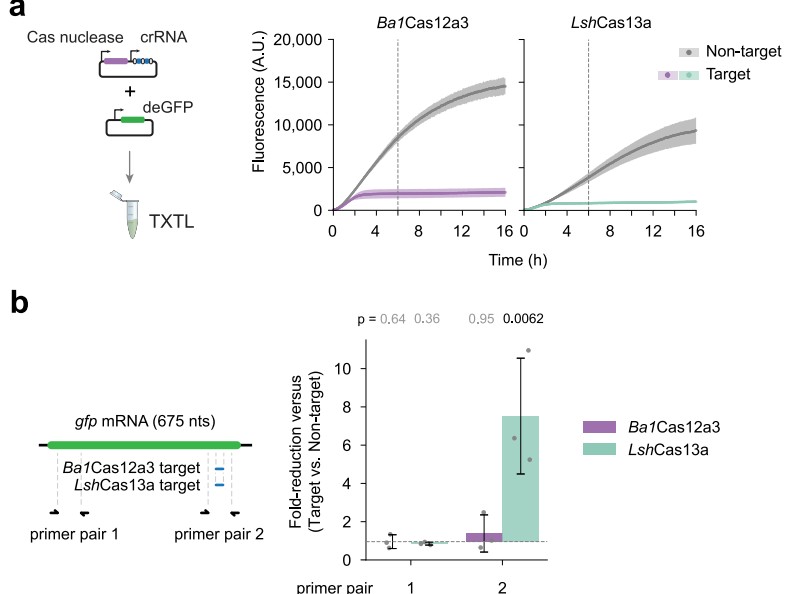

**a**

**b**

**Extended Data Fig. 3 | Comparison of RNA targeting by *Ba1*Cas12a3 and *Lsh*Cas13a in TXTL. (a)** GFP silencing in TXTL. Plasmids encoding the Cas nuclease, GFP-targeting or non-targeting crRNA, and deGFP are combined in a TXTL reaction, and GFP fluorescence is measured over time. Right: fluorescence time course for *Ba1*Cas12a3 and *Lsh*Cas13a. The vertical dashed line represents the time-point when a separate TXTL reaction was stopped for RNA extraction and RT-qPCR. Curves and shaded regions represent the mean ± standard deviation of separately prepared reactions (n = 3 or 4). **(b)** RT-qPCR analysis of the deGFP mRNA to determine the extent of mRNA cleavage. Left: location of the two primer pairs used for qPCR relative to the target sites for *Ba1*Cas12a3 and *Lsh*Cas13a. Reverse transcription was conducted with the respective reverse primer. Fold-reduction was calculated using the non-target condition as the reference. Each dot represents an independent measurement from one TXTL reaction, while bars and error bars represent the mean ± standard deviation from separate reactions (n = 3), with each the average of triplicate technical measurements. One-tailed paired Student's t-test. Non-significant p-values (p ≥ 0.05) are in gray.

**a**

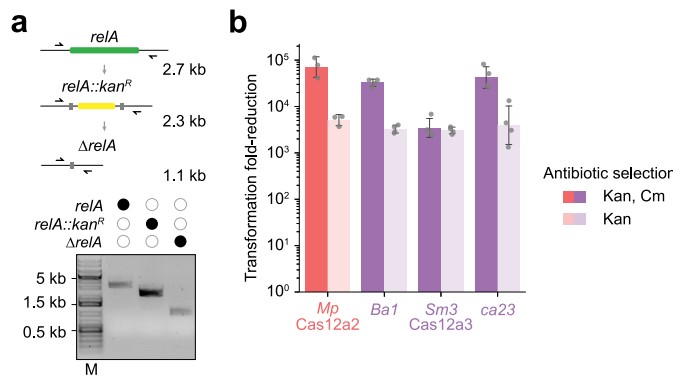

**b**

Extended Data Fig. 4 | Assessing plasmid interference in *E. coli* in the absence of the stringent response regulator *relA*. (**a**) Confirmation of deletion of *relA* in *E. coli* BL21(AI). Top: two-step approach to generate the complete deletion of *relA* using lambda-RED recombineering. Primer pairs to verify the deletion are shown at each step. Bottom: Resolved PCR products at each step of generating the Δ*relA* strain. For gel source data, see Supplementary Fig. 1. (**b**) Transformation fold-reduction as part of the plasmid interference assay using the Δ*relA* strain. See Fig. 1d for an overview of the assay. Each dot represents a biological replicate starting from a separate colony. Bars and error bars represent the mean ± standard deviation of biological replicates (n = 3 or 4). Kan, kanamycin; Cm, chloramphenicol.

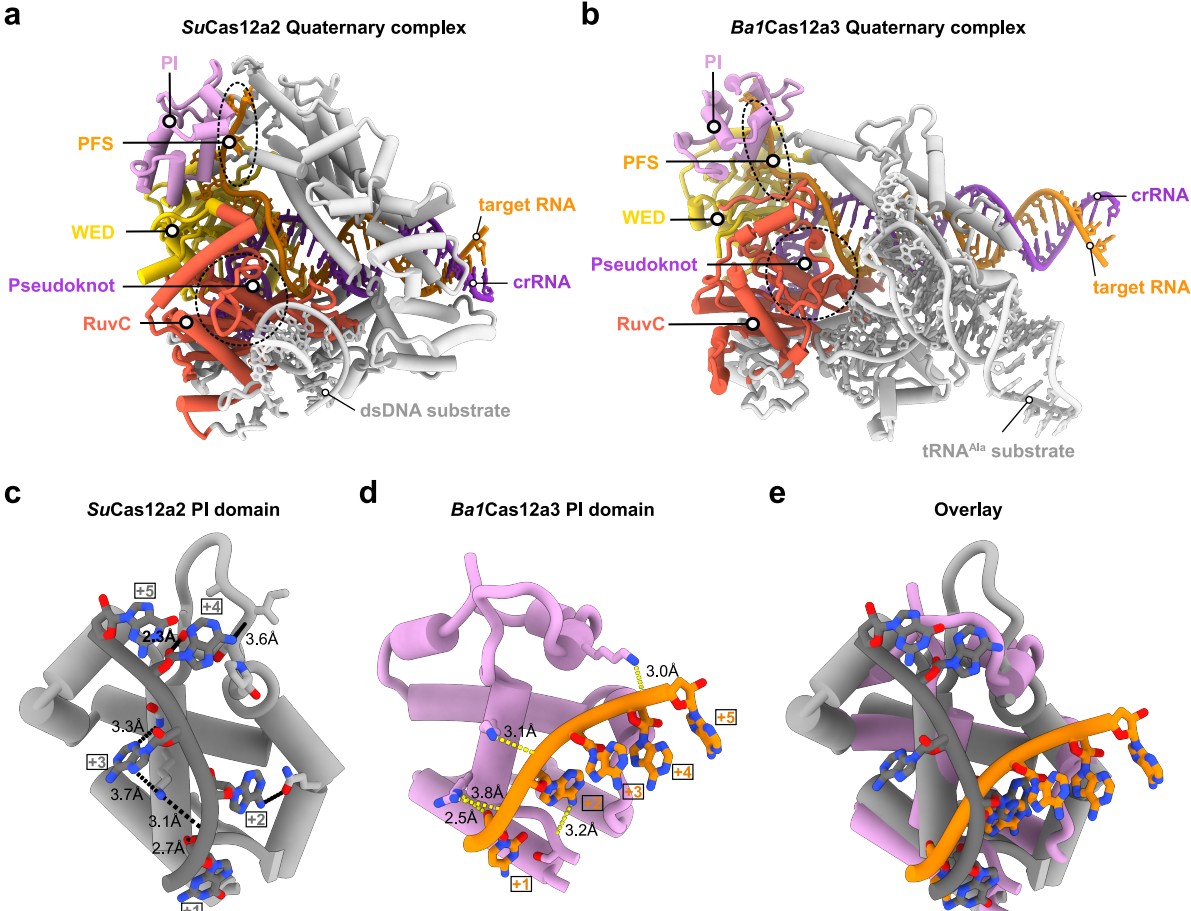

**Extended Data Fig. 5 | Structural comparison of the quaternary structures of *Su*Cas12a2 and *Ba1*Cas12a3.** (**a**) Model of the *Su*Cas12a2-crRNA-target RNA-dsDNA quaternary complex. (**b**) Model of the *Ba1*Cas12a3-crRNA-target RNA-tRNA^Ala(UGC) quaternary complex. (**c**) The interaction between the PI domain and the PFS nucleotides in the *Su*Cas12a2 quaternary complex. The bases of the PFS at position +2 (A22-Q681), +3 (A23-K621, A23-N625), and +4 (A24-I633) interact with the *Su*Cas12a2 PI domain. (**d**) The interaction between the PI domain and

the PFS nucleotides in the *Ba1*Cas12a3 quaternary complex. The base of adenine at position +2 (A37-S469) interacts with the *Ba1*Cas12a3 PI domain. (**e**) *Ba1*Cas12a3 PI domain (in plum) superimposed with *Su*Cas12a2 PI domain (in dark grey). The PFS for *Ba1*Cas12a3 is colored in dark orange. The PFS for *Su*Cas12a2 is colored in dim grey. The PFS sequences bind to each PI domain in a distinct manner. The differences between *Su*Cas12a2 and *Ba1*Cas12a3 in PFS recognition pattern are well correlated with the PFS library screening in Supplementary Fig. 2a.

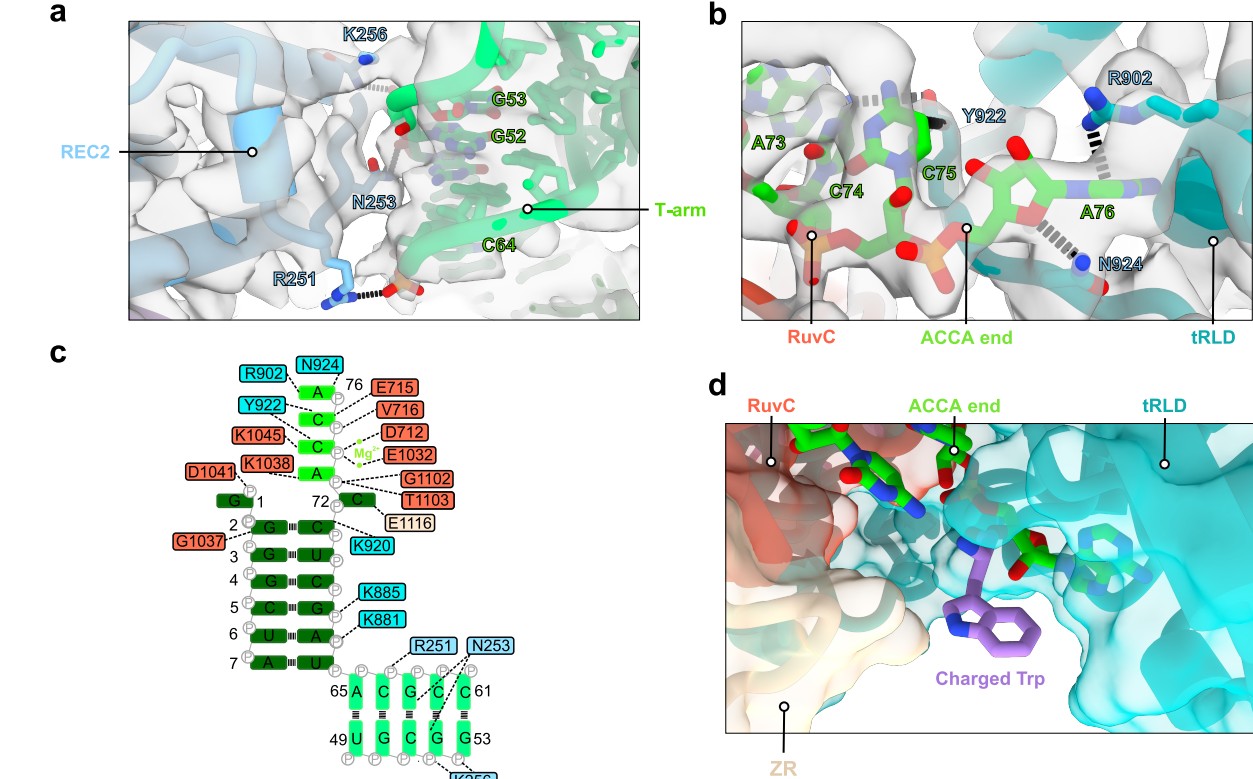

**Extended Data Fig. 6 | Detailed interactions between activated *Ba1*Cas12a3 and tRNA^Ala(UGC).** (**a**) Interactions between the REC2 loop of *Ba1*Cas12a3 and the T-arm of tRNA are shown with underlying EM density map. (**b**) Interactions between the tRLD of *Ba1*Cas12a3 and the 3′ CCA tail of the tRNA with underlying EM density map. (**c**) 2D interaction plot displaying the binding interface between *Ba1*Cas12a3 and tRNA. Electrostatic contacts are primarily between positively charged side chains or peptide backbone amines located at the ends of helices. Several residues (e.g., N253 and K1038) make contacts within the minor groove of RNA-duplexes, likely sensing the specific shape of the tRNA. (**d**) The bound tRNA is modeled with a tryptophan residue, demonstrating no steric clashes in the structure to impair binding to aminoacylated tRNAs.

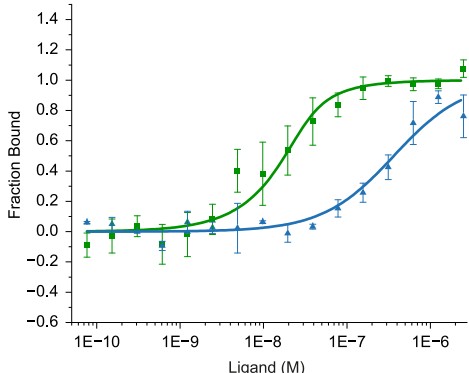

**Extended Data Fig. 7 | Measured binding affinities of a full-length and truncated tRNA$^{Ala}$ to d$Ba1$Cas12a3.** Binding of the activated d$Ba1$Cas12a3 ternary complex to the indicated fluorescent RNA substrate was quantified by MST. d$Ba1$Cas12a3 contains the E1065A mutation in the RuvC endonuclease domain to render it catalytically inactive. Symbols and error bars represent the mean ± standard deviation of independent experiments (n = 3).

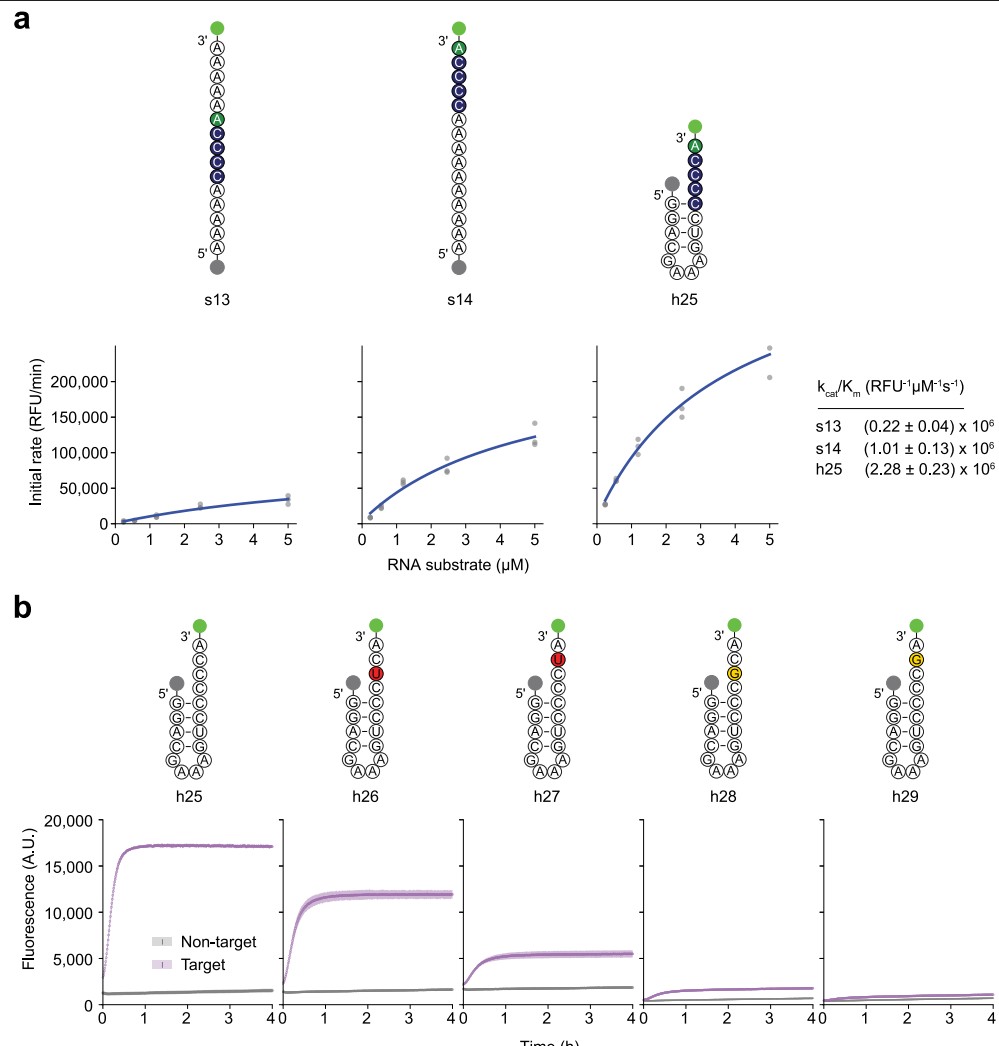

**Extended Data Fig. 8 | Assessing *trans* cleavage of other RNA substrates by *Ba1*Cas12a3.** (**a**) Comparing cleavage of different RNA substrates containing CCCCA. An activated *Ba1*Cas12a3 RNP was indicated with the indicated substrate, and fluorescence through the separation of the conjugated fluorophore (green circle) and quencher (gray circle) is measured over time for a range of substrate concentrations. Circles with nucleotides are colored to represent a mutation from C to U, a pyrimidine (red), or to G, a purine (yellow). The data from independent measurements (n = 3) were used to calculate catalytic efficiency ($k_{cat}/K_m$), with means ± standard error shown. (**b**) Comparing different truncated tRNA substrates with mutations to the cytosines in the 3′ CCA tail. Dotted curves and shaded regions represent the mean ± standard deviation of independently prepared and monitored reactions (n = 4).

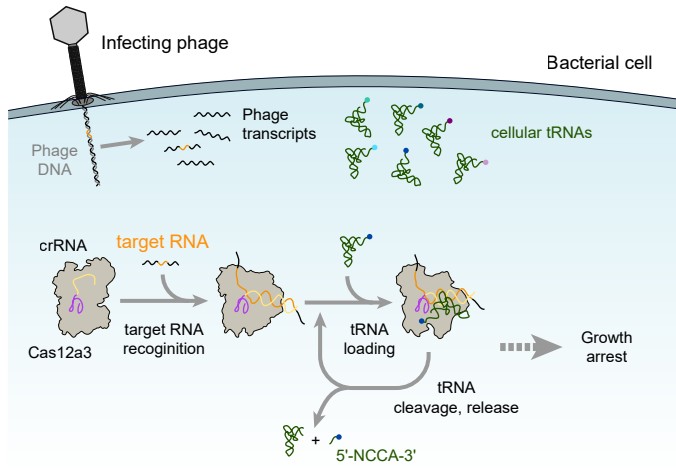

**Extended Data Fig. 9 | Model of immune defense via tRNA trimming by Cas12a3.** From left to right: Stepwise activation of *Ba1*Cas12a3 and tRNA cleavage leading to translational inhibition and growth arrest in response to a recognized target RNA expressed from an invading bacteriophage. The N in the cleaved tRNA tail is the discriminator base.

**Extended Data Table 1 | Cryo-EM data collection, refinement and validation statistics**

| | *Bal*Cas12a3 binary complex (EMDB-52275) (PDB 9HLX) | *Bal*Cas12a3 ternary complex (EMDB-52287) (PDB 9HM6) | *Bal*Cas12a3 quaternary complex (EMDB-52285) (PDB 9HM4) | *Bal*Cas12a3 post-cleavage complex (EMDB-52286) (PDB 9HM5) |
|---|---|---|---|---|
| **Data collection and processing** | | | | |
| Magnification | 130K | 130K | 130K | 130K |
| Voltage (kV) | 200 | 200 | 200 | 200 |
| Electron exposure (e–/Å²) | 40 | 40 | 40 | 40 |
| Defocus range (µm) | −0.8 to −2.0 | −0.8 to −2.0 | −0.8 to −2.0 | −0.8 to −2.0 |
| Pixel size (Å) | 0.91 | 0.91 | 0.91 | 0.91 |
| Symmetry imposed | C1 | C1 | C1 | C1 |
| Initial particle images (no.) | 981,900 | 437,410 | 1,070,711 | 1,070,711 |
| Final particle images (no.) | 99,354 | 119,647 | 116,277 | 45,337 |
| Map resolution (Å) | 3.8 | 4.0 | 3.1 | 3.3 |
| FSC threshold | 0.143 | 0.143 | 0.143 | 0.143 |
| Map resolution range (Å) | 3.3-16.7 | 3.4-51.6 | 2.6-46.4 | 2.8-51.6 |
| **Refinement** | | | | |
| Initial model used | AlphaFold3 model | AlphaFold3 model | AlphaFold3 model | AlphaFold3 model |
| Model resolution (Å) | 4.0 | 4.4 | 3.5 | 3.5 |
| FSC threshold | 0.5 | 0.5 | 0.5 | 0.5 |
| Map sharpening $B$ factor (Å²) | -189.3 | -50 | -50 | -50 |
| Model composition | | | | |
| Non-hydrogen atoms | 10,287 | 9556 | 11,182 | 9645 |
| Protein residues | 1156 | 960 | 958 | 958 |
| Nucleotide base | 41 | 80 | 157 | 85 |
| Ligands | 0 | 0 | 2Mg²⁺ | 2Mg²⁺ |
| $B$ factors (Å²) | | | | |
| Protein | 73.69 | 213.18 | 100.97 | 87.74 |
| Nucleotide | 99.72 | 233.63 | 178.84 | 101.22 |
| Ligand | - | - | 73.47 | 51.45 |
| R.m.s. deviations | | | | |
| Bond lengths (Å) | 0.004 | 0.003 | 0.003 | 0.006 |
| Bond angles (°) | 0.555 | 0.739 | 0.589 | 0.615 |
| Validation | | | | |
| MolProbity score | 1.82 | 2.04 | 1.55 | 1.65 |
| Clash score | 7.06 | 13.08 | 4.67 | 5.03 |
| Poor rotamers (%) | 0.10 | 0 | 0.35 | 0.12 |
| Ramachandran plot | | | | |
| Favored (%) | 93.40 | 93.70 | 95.58 | 94.42 |
| Allowed (%) | 6.51 | 5.99 | 4.32 | 5.37 |
| Disallowed (%) | 0.09 | 0.32 | 0.11 | 0.21 |

# Reporting Summary

## Statistics

For all statistical analyses, confirm that the following items are present in the figure legend, table legend, main text, or Methods section.

| n/a | Confirmed | |
|---|---|---|
| ☐ | ☒ | The exact sample size (*n*) for each experimental group/condition, given as a discrete number and unit of measurement |
| ☐ | ☒ | A statement on whether measurements were taken from distinct samples or whether the same sample was measured repeatedly |
| ☐ | ☒ | The statistical test(s) used AND whether they are one- or two-sided<br>*Only common tests should be described solely by name; describe more complex techniques in the Methods section.* |
| ☒ | ☐ | A description of all covariates tested |
| ☒ | ☐ | A description of any assumptions or corrections, such as tests of normality and adjustment for multiple comparisons |
| ☐ | ☒ | A full description of the statistical parameters including central tendency (e.g. means) or other basic estimates (e.g. regression coefficient) AND variation (e.g. standard deviation) or associated estimates of uncertainty (e.g. confidence intervals) |
| ☐ | ☒ | For null hypothesis testing, the test statistic (e.g. *F*, *t*, *r*) with confidence intervals, effect sizes, degrees of freedom and *P* value noted<br>*Give P values as exact values whenever suitable.* |
| ☒ | ☐ | For Bayesian analysis, information on the choice of priors and Markov chain Monte Carlo settings |
| ☒ | ☐ | For hierarchical and complex designs, identification of the appropriate level for tests and full reporting of outcomes |
| ☒ | ☐ | Estimates of effect sizes (e.g. Cohen's *d*, Pearson's *r*), indicating how they were calculated |

*Our web collection on statistics for biologists contains articles on many of the points above.*

## Software and code

Policy information about availability of computer code

| Data collection | CFX Maestro v5.3.022.1030, Typhoon FLA 7000, MinKNOW 24.11, Thermo Scientific Smart EPU |
|---|---|
| Data analysis | Python 3.9, pandas 1.2.0, matplotlib 3.5.2, SAMtools 1.9, ClipKIT, IQ-TREE 2.3.6, warpdemux 0.4.4,   dorado 0.8.3, cmsearch, tRNAscan-SE 2.0, BWA-MEM 0.7.17, AlphaFold3, ChimeraX 1.7, ISOLDE,  Phenix 1.20.1, DefenseFinder 2.0.1, CRISPRCasFinder 4.2.21, cryoSPARC 4.6, Trimmomatic v0.39 |

For manuscripts utilizing custom algorithms or software that are central to the research but not yet described in published literature, software must be made available to editors and reviewers. We strongly encourage code deposition in a community repository (e.g. GitHub). See the Nature Portfolio guidelines for submitting code & software for further information.

## Data

Policy information about availability of data

All manuscripts must include a data availability statement. This statement should provide the following information, where applicable:

- Accession codes, unique identifiers, or web links for publicly available datasets
- A description of any restrictions on data availability
- For clinical datasets or third party data, please ensure that the statement adheres to our policy

The Illumina-based PFS screen data and the direct RNA Nanopore sequencing reads were deposited at the European Nucleotide Archive (ENA) under accession code PRJEB88250 (https://www.ebi.ac.uk/ena/browser/view/PRJEB88250).

Models and associated cryo-EM maps have been deposited into the PDB database with the following accession codes: Ba1Cas12a3 binary complex (EMD-52275, PDB: 9HLX), Ba1Cas12a3 ternary complex (EMD-52287, PDB: 9HM6), Ba1Cas12a3 quaternary complex at pre-cleavage state (EMD-52285, PDB: 9HM4), and Ba1Cas12a3 quaternary complex at post-cleavage state (EMD-52286, PDB: 9HM5).
Source data are included for graphical results. Raw gel images are included as Supplementary Figure 1.

## Research involving human participants, their data, or biological material

Policy information about studies with [human participants or human data](). See also policy information about [sex, gender (identity/presentation), and sexual orientation]() and [race, ethnicity and racism]().

| Reporting on sex and gender | Not applicable. No human participants, their data, or biological material used in this work. |
|---|---|
| Reporting on race, ethnicity, or other socially relevant groupings | Not applicable. No human participants, their data, or biological material used in this work. |
| Population characteristics | Not applicable. No human participants, their data, or biological material used in this work. |
| Recruitment | Not applicable. No human participants, their data, or biological material used in this work. |
| Ethics oversight | Not applicable. No human participants, their data, or biological material used in this work. |

Note that full information on the approval of the study protocol must also be provided in the manuscript.

# Field-specific reporting

Please select the one below that is the best fit for your research. If you are not sure, read the appropriate sections before making your selection.

☒ Life sciences          ☐ Behavioural & social sciences          ☐ Ecological, evolutionary & environmental sciences

For a reference copy of the document with all sections, see [nature.com/documents/nr-reporting-summary-flat.pdf](nature.com/documents/nr-reporting-summary-flat.pdf)

# Life sciences study design

All studies must disclose on these points even when the disclosure is negative.

| Sample size | Quantitative experiments involved a sample size of at least three, the minimal number needed to perform statistical analyses. The exact sample size is indicated in each figure legend. |
|---|---|
| Data exclusions | No data were excluded from the analyses. |
| Replication | All measurements were performed in at least three biological replicates, unless stated otherwise. All replicates were successful. |
| Randomization | No randomization was performed, as is common in molecular biology and microbiology studies. In some cases, values were normalized to a control condition in the same experiment, such as the PFS library under non-targeting conditions (e.g., Fig. 1c), CFUs under non-targeting conditions (e.g., Fig. 1d) or the maximum enzymatic activity in an in vitro cleavage assay (e.g., Fig. 1f), to reduce between -experiment variability independent of the specific sample. |
| Blinding | No blinding was performed, as is common in molecular biology and microbiology studies. A major rationale is that the process of experimental design, conducting an experiment, and analyzing the results is conducted by a single researcher, and involving others to blind the results creates an additional burden on personnel and creates the potential of labeling errors. Instead, results are confirmed through biological replicates and parallel assays to ensure consistency, such as testing Cas12a3 nucleases in E. coli and in vitro (e.g., Figs. 1d-f). |

# Reporting for specific materials, systems and methods

We require information from authors about some types of materials, experimental systems and methods used in many studies. Here, indicate whether each material, system or method listed is relevant to your study. If you are not sure if a list item applies to your research, read the appropriate section before selecting a response.

## Materials & experimental systems

| n/a | Involved in the study |
|---|---|
| ☒ ☐ | Antibodies |
| ☒ ☐ | Eukaryotic cell lines |
| ☒ ☐ | Palaeontology and archaeology |
| ☒ ☐ | Animals and other organisms |
| ☒ ☐ | Clinical data |
| ☒ ☐ | Dual use research of concern |
| ☒ ☐ | Plants |

## Methods

| n/a | Involved in the study |
|---|---|
| ☒ ☐ | ChIP-seq |
| ☒ ☐ | Flow cytometry |
| ☒ ☐ | MRI-based neuroimaging |

# Plants

| Seed stocks | No plants used in this study. |
|---|---|
| Novel plant genotypes | No plants used in this study. |
| Authentication | No plants used in this study. |

