## [Peer Review file · Nature]

RNA-triggered Cas12a3 cleaves tRNA tails to execute bacterial immunity

Corresponding Author: Professor Chase Beisel

Version 0:

Reviewer comments:

Referee #1

(Remarks to the Author)

Dmytrenko, Yuan et al. describe the activity of two newly discovered Cas12a variants with diverged activities. This study follows the same team's previous discovery of Cas12a2, an RNA-targeting Cas12a variant that non-specifically cleaves dsDNA (as well as RNA and ssDNA) upon activation following target cleavage. Now, the authors have discovered two new clades, designated Cas12a3 and Cas12a4, with distinct activities. Both bind to RNA targets that are complementary to the guide crRNA, and both specifically cleave RNA substrates in trans. However, Cas12a3 is unique in two ways: first, it does not appear to have any cleavage activity toward its target RNA; second, its trans cleavage appears to be specifically directed toward the CCA tails of tRNAs.

The authors provide evidence that tRNA cleavage is the direct route of Cas12a3-mediated immunity and structurally characterize Cas12a3 to gain insights into its specificity. They identify a tRNA loading domain (tRLD) that appears to block the RuvC active site in the target-bound structure, but is dislodged from interactions with RuvC upon tRNA binding. The tRNA-bound structure demonstrates how the CCA tail of tRNAs can bind directly in the RuvC active site, regardless of aminoacylation status. This structural data suggests that Cas12a3 may have evolved a mechanism to selectively cleave tRNAs, leading the authors to conclude that Cas12a3 represents a new class of CRISPR-mediated immunity that exclusively targets tRNAs.

Overall, this is a very well-written and well-presented manuscript. The data are easy to follow, and generally interpreted appropriately. The discovery of the tRNA cleavage mechanism is interesting, and the structures appear to be of relatively high quality. The authors have made efforts to distinguish Cas12a3 from previously reported tRNA-cleavage activity of Cas13. Although the mechanism of tRNA cleavage is clearly distinct between the two enzymes, it is unclear whether the path to immunity is significantly different, and the data do not fully support some of the novelty claims made by the authors. I have detailed examples below that the authors may address by toning down language or by including additional analysis. The degree to which the authors have to reduce their novelty claims may impact whether this manuscript is appropriate for publication in Nature, versus a more specialized journal (e.g. NSMB).

Major concerns:

1. Some of the authors' claims about the novelty of tRNA-targeting CRISPR-Cas systems seem overstated. On lines 16-17, the authors suggest that "widespread tRNA inactivation" is "a previously unrecognized CRISPR-based immune strategy", while on lines 40-41, the authors state that "no CRISPR-Cas system has yet been demonstrated to exclusively cleave tRNAs". In reference 24, Cas13 was shown to provide defense against phage through cleavage of the anticodon loop of tRNAs, resulting in inhibition of protein synthesis. The authors of that paper showed that Cas13 cleave tRNAs more rapidly and more substantially than mRNAs in *E. coli*. While Cas13 may not "exclusively" cleave tRNAs, this previous study suggests that the main route of Cas13-mediated immunity is via tRNA cleavage.

2. It is also unclear that Cas12a3 itself exclusively cleaves tRNAs, as the authors did not probe total RNA cleavage in cells. While their sequencing of rRNA from TXTL reactions indicates that rRNA is not cleaved by Cas12a3, this does not rule out mRNA cleavage by Cas12a3. Indeed, some of the in vitro cleavage results seem to suggest that Cas12a3 can cleave ssRNA (e.g. Figs. 2a, 5a, and S3b). While the authors present a compelling model that the tRLD blocks cleavage by RuvC in the absence of a structured RNA, they have not tested this thoroughly enough to rule out cleavage of ssRNA. It therefore seems overstated to say that Cas12a3 exclusively cleaves tRNA, rather than that it may preferentially cleaves tRNA. If this is the case, it does not seem that there is much distinction in the route of Cas12a3-mediated immunity in comparison to Cas13. Both cleave tRNAs faster and to a larger degree than other RNAs in the cell, leading to translational repression and cell growth arrest.

Specific suggestions for how to address these concerns:

1. The fact that Cas12a3 does not seem to have a preference for the chemical nature of the 3' end of its substrates suggests that it could cleave a CCA sequence that is located internally within an RNA (i.e. with more sequence downstream of the recognition motif). Have the authors investigated whether the sequence specificity observed at the 3' end of substrates in Fig. 5a can also allow cleavage in the middle of ssRNA sequences? This could be related to the cleavage observed for various ssRNA substrates used in the manuscript, and would suggest that other RNAs that contain the same sequence can be cleaved in cells.

2. Much of the argument for tRNA selectivity stems from the structural differences between the target-bound ternary complex and the target + tRNA bound quaternary complex, in which tRLD movement is proposed to expose the RuvC active site for substrate cleavage. However, it seems likely that the tRLD would be at least somewhat dynamic in the absence of bound tRNA, otherwise it would not be possible to undergo the conformational change required to bind to the tRNA. Have the authors explored whether there is evidence for an "undocked" conformation of the ternary complex, in which the tRLD is not directly interacting with the RuvC domain? I ask due to two lines of evidence in Fig. S14b. First, it appears that the resolution of the tRLD is quite low in the refined model, suggesting that it may be more flexible than other parts of the structure. Second, it appears that there is a (potentially lower-resolution) class of particles in which the tRLD is not docked directly against the RuvC domain (the greenish class on the right). Did the authors explore this class? While it may be a minor class of particles, the fact that it appears in the classification suggests that Cas12a3 can adopt conformations in which the RuvC domain is exposed. This could allow for cleavage of ssRNAs containing the CCA recognition motif outside of the context of tRNAs.

3. If tRLD blocks cleavage of unstructured RNAs, then disruption of interactions between RuvC and tRLD may increase cleavage activity of Cas12a3 against non-tRNA substrates. Have the authors tested whether the tRLD deletion mutant or point mutations that disrupt interactions between tRLD and RuvC increases the promiscuity of Cas12a3 cleavage (e.g. using the random RNA substrates from Fig. 2a)? Is it possible that this is the reason that the Y922A mutant results in an increase in substrate cleavage? In my examination of the ternary structure, it appears that Y992 may be involved in interactions with the RuvC domain. Mutating the residue may make the domain more flexible, leading to the increased ability to bind/cleave tRNA (and potentially other) substrates.

Minor concerns:

1. In the abstract, the authors refer to "adaptive immunity by Cas12a3 effector nucleases". It is unclear whether Cas12a3 is part of an adaptive immune system, as the authors do not describe the presence of adaptation genes or CRISPR loci associated with cas12a3 genes. The authors could either add such analysis or remove reference to these nucleases as part of adaptive immune systems.

2. Lines 108-109: This sentence is a bit confusing and could be reworded, as it makes it sound like both Cas12a3 and Cas12a4 do not cleave the target RNA, when this only applies to Cas12a3.

3. Lines 251-252: The difference in binding between full length tRNA and the tRNA lacking a CCA tail is not very substantial (~2x difference in Kd). It is unclear that this difference in affinity accounts for the differential cleavage, given that the concentration of enzyme used was in excess of both Kd's. It seems more likely that Cas12a3 can bind to structured RNAs regardless of the presence of a CCA tail, but also displays a significant sequence specificity for its cleavage sites, preventing cleavage of structured RNAs that lack this sequence.

4. In heatmaps like in Fig. 1f, how was fluorescence production normalized? Please include this information in the figure legend or methods. The explanation on lines 790-791 is a bit vague.

5. In Fig. S3b, there appears to be a discrete, slightly shorter product when Sm3Cas12a3 is incubated with the unlabeled target and 5' FAM RNA substrate. Is this reproducible and is it due to Cas12a3 cleavage activity? It appears that the enzyme may be cleaving at a discrete site near the 3' end of the RNA.

Referee #2

(Remarks to the Author)

Summary

In this manuscript, Dmytrenko et al. describe a new subfamily of CRISPR-Cas12 enzymes (Cas12a3 and Cas12a4) and show remarkably that upon activation by a complementary RNA-target Cas12a3 has a specific preference for the cleavage

of the CCA tail of most E coli tRNAs, when expressed and activated in E. coli using either plasmid or phage challenge. They then go on to determine the cryo-EM structures of multiple states of Cas12a3 assembly, targeting, and catalysis, and show there are important domains within Cas12a3 that make tRNA-specific contacts; this results in a unique RuvC gating / accessibility mechanism that precludes other RNA substrates from gaining access in both the apo, binary, and ternary forms of the enzyme complex. Finally, they explore the design of mini-tRNA-like synthetic fluorescence-quencher reagents and discover that for Cas12a3, they can design a specific substrate that can be cleaved efficiently in trans by Cas12a3 upon activation but that is not cleaved by LwaCas13a or PsmCas13b, paving the way for the potential use of Cas12a3 in multiplexed diagnostic settings.

Overall, this an exciting finding that adds significantly to recent experimental momentum around tRNAs being a central target of prokaryotic innate immunity, and most of the interpretations are supported by well-controlled experiments and explained in a well-written and easy to follow manuscript. That said, some claims and interpretations that impact the conclusions and usefulness of the potential downstream diagnostic technology are oversold and not backed up by the required data, and as a result some major and minor concerns need to be addressed. See below for my specific comments.

Major Comments

* While I understand it's likely that translation is blocked given the amount of de-acylation / CCA cleavage you observe in vitro and in TXTL experiments, you haven't shown this explicitly in vivo under phage infection or plasmid challenge conditions. The claim really relies on loss of GFP expression with in vitro supplementation of Cas12 into TXTL cell-free reactions; is there a chance that the loss of GFP expression isn't due to loss of global protein translation directly because of loss of tRNA availability but rather in sensing of deacylated tRNAs by a stress response (e.g. stringent response by a combined loss of translation and transcription)? (More on this below.) Have the authors considered running total protein assays in vivo to show loss of general translation / protein production upon phage infection or doing a plasmid challenge experiment (without antibiotics)? One way would be to look at nascent polypeptide production using a O-Propargyl-puromycin assay or equivalent.

* Relatedly, deacylated tRNAs are known activators of the stringent response through activation of RelA (p)ppGpp synthesis and its downstream action as a secondary messenger, which can also ultimately result in transcriptional and translational shut-off amongst other phenotypes such as biofilm formation, etc. However, while it is known from structural studies that RelA contacts deacylated CCA tails of the tRNAs when docked into the ribosome, it's still not clear to me functionally whether it would be possible that de-acylated tRNAs missing their CCA tails (in your case) are still able to activate RelA and the stringent response. I think minimally a comment or discussion on this potential mechanism in the text is required. Ideally, if there is some facile way (HPLC or a stringent response reporter or equivalent) to determine whether the stringent response is being activated during phage infection or plasmid challenge, that would certainly elevate the significance of the manuscript's findings.

* I think the authors need to be more careful with the interpretation as to whether acylation / charging and tRNA chemical modifications affect the specificity and efficiency of tRNA cleavage by Cas12 here. For example, modifications are well known to alter the 3D structure and conformational dynamics of tRNA, so it can't be ruled out that changes in structure and dynamics could occur in the presence or absence of modifications and these could have an indirect effect on tRNA binding and/or cleavage rates even if there are no direct modification-specific contacts between the protein and the tRNA. Furthermore, without much more careful thermodynamic binding and/or kinetic cleavage assays with fully modified vs. unmodified tRNAs being carried out (a tough experiment, so not suggesting it be done unless purified specific modified tRNAs can be sourced, which is not straightforward), it is an overinterpretation to claim that modifications don't matter here. This reviewer suggests toning this language down and make it clear that maybe modifications may have an indirect effect, in lieu of more careful kinetic/thermodynamic measurements.

* While the discovery that an orthogonal, synthetic trans-RNA substrate can be designed for Cas12a3 and might enable multiplexing, as far as I can tell the manuscript doesn't actually demonstrate the ability to multiplex Cas12a3, LwaCas13a and PsmCas13b in a one-pot reaction, which is minimally the level of proof of concept this reviewer would expect when making claims about multiplexing in a journal of this impact. The demonstration of multiplexing could be achieved by using a contrived sample where one could spike in synthetic RNAs (at a range of concentrations) into a total RNA sample from humans or E coli for example and then testing for a three-way multiplex readout from a single pot (i.e. all detection reactions in one well) using fluorescence or equivalent. This would certainly increase the impact of this section of the manuscript and perhaps offer more information about the benefits and/or potential limitations of Cas12a3-based detection in more real-world samples (e.g. one could imagine that all the tRNA present in a real-world RNA sample could act as a competitive inhibitor for the fluorescence substrate and decrease the signal-to-noise/dynamic range of the assay in a way that makes multiplexing (or detection in general) achievable).

Minor Comments

*Line 41 and elsewhere in the text: The authors make a claim that Cas12a3 cleaves tRNA exclusively; however, your collateral cleavage assay potentially suggests linear substrates could get cleaved at some low level in vivo like what is seen in ref. 24. Given TXTL was used in the assay, which is possibly depleted of all other mRNAs and supplementation of additional tRNAs as part of the master mix is common, making the TXTL possess a large overabundance of tRNAs relative to other targets, is it possible that cleavage of other RNAs is possible in vivo but could be missed in TXTL? On this note, I feel like the discoveries in ref. 24 are downplayed in this manuscript and need to be cited more positively; the limitation in the use of TXTL also needs to be addressed or the related claims need to be toned down somewhat.

* Growth arrest claims: A better explanation of the “indicative of growth arrest” on line 77 in my opinion is required. While it is likely there is a growth reduction and that is why there are fewer transformants, this is a less direct measurement, and a lack of transformation could be possibly due to other reasons. In this case the language needs to be a lot clearer when describing the experimental interpretation of Fig. 1d and perhaps growth curves in Fig. S2c needed to be more clearly invoked when describing “indicative of growth arrest”.

* Line 81 or in the figure: It might be worth explaining how the SOS reporter works (RecA promoter, etc.) It's not clear from the text or the figure and thus may make it easier for the reader to interpret.

* Lines 106-107: With respect to Cas13's “largely indiscriminate” cleavage, is that actually true? This needs to be tempered and something like the following needs to be added: “although more recent evidence points a preference of certain Cas13 ortholog for tRNAs...” I think this needs to be added because as it is currently written it is not a completely fair reference to what is known in the literature.

* Fig. 2b: Perhaps use thicker lines in the TXTL time courses; it was hard to see the blue line in there for the MbCas12a2 especially.

* The colored/semi-transparent keys in the figures (Fig. 1 especially, but others, too) are hard to interpret, a little small and the key difference in transparency is too subtle.

* How were the non-cleaving conditions obtained for the cryo-EM structure determination? It would be nice to include that detail in the main text more clearly. EDTA? Mutants? No metal +/- EDTA?

Referee #3

(Remarks to the Author)

This is an interesting novel work showing that Cas12a3 effector nucleases from Type V CRISPR-Cas systems in bacteria can selectively cleave tRNAs upon target RNA recognition. Interestingly, target RNA triggers Cas12a3 to cleave the conserved 5'-CCA-3' tail of diverse tRNAs, driving growth arrest and anti-phage defense. The authors further provide structural data that back up their in vitro data. By using structural data, further mutagenesis analysis was used to identify critical amino acid contacts that provide specificity towards tRNA substrate. Finally, the authors engineered synthetic substrates that, together with the 1 specificity of Cas12a3 toward tRNA substrates, expanded the capacity of selected CRISPR-based diagnostic platforms for multiplexed RNA detection.

The data is solid and I only have few issues to be addressed in a revised version.

1. It would be great to back up data from Figs. 2/S4 on the nanopore RNA-seq with representative northern blotting on targeted (tRNA) and non-targeted RNAs (5S).

2. The same as in S6, selected tRNA probes should be used for two representative tRNAs that are “well cleaved” vs “less efficiently cleaved” from the pool of bacterial tRNAs. This will serve as controls for RNA-seq.

3. Mutagenesis data should be complemented from tRNA side (to back up Fig. 3); please use in vitro transcribed tRNA variants with changes in the CCA-tail (purine-to purine/purine-pyrimidine).

4. It is necessary to determine whether Cas12a3 can cleave tRNAs in the complex with translation elongation factors, as well as in the ribosomal fractions.

5. While predicted, it would be interesting to determine whether Cas12a3 can cleave eukaryotic tRNAs, which are more densely modified. This may have applications in the future research.

Version 1:

Reviewer comments:

Referee #1

(Remarks to the Author)

The authors have done an outstanding job addressing my and the other reviewers' concerns. With the addition of new experiments, the authors have convincingly shown that Cas12a3 specifically targets tRNA, which blocks translation. I have no further concerns and believe the manuscript should be accepted for publication.

Referee #2

(Remarks to the Author)

The authors have dealt constructively with the reviewers' comments, added further data and clarified several uncertainties in the text- specifically around Cas13, multiplexing and tRNA substrates and modifications. This is exciting work, and I support publication in Nature.

Referee #3

(Remarks to the Author)

The revision is significantly improved. All my issues were addressed adequately. I recommend the work for publication.

We are grateful to the editor and the three reviewers for their thoughtful and constructive feedback and suggestions. The reviewers were generally enthusiastic about the manuscript, and agreed that our data largely support our claims that Cas12a3 and Cas12a4 define distinct CRISPR nuclease clades that confer immunity through biochemical mechanisms unique from other Cas12 nucleases. Some concerns were also raised, with requests for more context or experiments to support specific claims. To address the raised concerns, we performed the following 10 new experiments and one structural analysis:

1. Quantified the reduction in a targeted GFP transcript using *Ba1Cas12a3* and *LshCas13a* in TXTL (**Fig. S9**). Compared to the non-targeting control, *LshCas13a*, but not *Ba1Cas12a3*, yielded significant cleavage, in line with Cas13a principally cleaving target RNAs and Cas12a3 principally cleaving tRNAs.
2. Assessed the impact of removing different structural elements of the tRNA on cleavage by *Ba1Cas12a3* (**Fig. 3e**). Removal of the structural arms was tolerated, justifying the use of the truncated tRNA substrate comprising the anti-codon loop, acceptor stem, and 3' CCA tail.
3. Measured the binding affinity of tRNA^{Ala(UGC)} and the truncated tRNA substrate to *dBa1Cas12a3* (**Fig. S17**). The nuclease bound the full-length tRNA with 62-fold higher affinity than the truncated tRNA substrate, indicating an effect of truncating the tRNA.
4. Compared the cleavage efficiency of RNA substrates containing CCA at the 3' end or internally (**Figs. S18a**). We observed a kinetic preference for hairpin substrates with a 3' CCA tail, in line with tRNAs as the preferred cleavage substrate of *Ba1Cas12a3*. The substrate with an internal CCA was also cleaved, indicating that *Ba1Cas12a3*, in principle, can cleave other RNAs, albeit with significantly reduced efficiency.
5. Measured cleavage of RNA substrates with additional base mutations (**Figs. 3f and S18b**). *Ba1Cas12a3* displayed limited tolerance to mutation of the cytosines in the CCA tail, with greater tolerance to another pyrimidine (C-to-U) than to a purine (C-to-G), while *Sm3Cas12a3* did not tolerate any mutations, indicating different extents of sequence specificity for the tRNA tail across Cas12a3 orthologs.
6. Assessed cleavage of an arbitrary RNA sequence as well as the library reporter with the *Ba1Cas12a3* mutants Δ tRLD and Y922A (**Figs. 3i and S20**). Both mutants exhibited reduced cleavage compared to WT, arguing against either mutation broadening the spectrum of cleavage substrates recognized by *Ba1Cas12a3*.
7. Analyzed how tRNAs interact with *Ba1Cas12a3* versus the ribosome and elongation factor Tu (EF-Tu) (**Fig. S16**). EF-Tu, when free or bound to the ribosome, interacts with the same interface of the tRNA recognized by *Ba1Cas12a3*, suggesting that *Ba1Cas12a3* would only recognize and cleave tRNAs not associated with the ribosome or EF-Tu.
8. Assessed Cas12a3-mediated immunity in a Δ *relA* strain of *E. coli* (**Fig. S10**). Cas12a3 conferred immunity, arguing against involvement of the stringent response in Cas12a3-mediated immunity.
9. Assessed cleavage of individual tRNAs by northern blotting analysis (**Figs. S5 and S7**). Specific and complete cleavage of tRNAs was observed both in TXTL and when using

purified *E. coli* tRNAs for all three Cas12a3 orthologs, while no cleavage of 5S rRNA was observed.

10. Assessed cleavage of bulk tRNA isolated from yeast (**Fig. S8b**). All three tested Cas12a3 orthologs cleaved the pool of 3'-labeled tRNAs, indicating that Cas12a3 can also cleave eukaryotic tRNAs.
11. Performed multiplexed RNA detection using *Ba1*Cas12a3, *Lwa*Cas13a, and *Psm*Cas13b under conditions approximating real-world sample testing (**Figs. 5d-g and S25d-f**). When combined, the nucleases could detect gene transcripts from SARS-CoV-2, IAV, and RSV individually or in combination, with introduced human total RNA not impeding the readout.

Besides strengthening our claims and providing a clearer understanding of the substrate specificity governing Cas12a3 activation and trans cleavage, these data also helped us redefine our proposed role of the tRNA-loading domain, which we now state facilitates substrate binding rather than restricting access to the active site. In total, the reviewers' feedback significantly strengthened and sharpened our conclusions regarding the unique biochemical properties and immune function of Cas12a3.

Specific responses to each reviewer's comment are below, marked in blue. Corresponding changes in the main text and the supplementary information are marked in red.

Referee #1 - structural biology of CRISPR

Dmytrenko, Yuan et al. describe the activity of two newly discovered Cas12a variants with diverged activities. This study follows the same team's previous discovery of Cas12a2, an RNA-targeting Cas12a variant that non-specifically cleaves dsDNA (as well as RNA and ssDNA) upon activation following target cleavage. Now, the authors have discovered two new clades, designated Cas12a3 and Cas12a4, with distinct activities. Both bind to RNA targets that are complementary to the guide crRNA, and both specifically cleave RNA substrates in trans. However, Cas12a3 is unique in two ways: first, it does not appear to have any cleavage activity toward its target RNA; second, its trans cleavage appears to be specifically directed toward the CCA tails of tRNAs.

The authors provide evidence that tRNA cleavage is the direct route of Cas12a3-mediated immunity and structurally characterize Cas12a3 to gain insights into its specificity. They identify a tRNA loading domain (tRLD) that appears to block the RuvC active site in the target-bound structure, but is dislodged from interactions with RuvC upon tRNA binding. The tRNA-bound structure demonstrates how the CCA tail of tRNAs can bind directly in the RuvC active site, regardless of aminoacylation status. This structural data suggests that Cas12a3 may have evolved a mechanism to selectively cleave tRNAs, leading the authors to conclude that Cas12a3 represents a new class of CRISPR-mediated immunity that exclusively targets tRNAs.

Overall, this is a very well-written and well-presented manuscript. The data are easy to follow, and generally interpreted appropriately. The discovery of the tRNA cleavage mechanism is interesting, and the structures appear to be of relatively high quality. The authors have made efforts to distinguish Cas12a3 from previously reported tRNA-cleavage activity of Cas13. Although the mechanism of tRNA cleavage is clearly distinct between the two enzymes, it is unclear whether the path to immunity is significantly different, and the data do not fully support some of the novelty claims made by the authors. I have detailed examples below that the authors may address by toning down language or by including additional analysis. The degree to which the authors have to reduce their novelty claims may impact whether this manuscript is appropriate for publication in Nature, versus a more specialized journal (e.g. NSMB).

We thank the reviewer for their positive comments and feedback on our efforts to distinguish Cas12a3 from *Lsh*Cas13a and other Cas13 nucleases, as discussed in the ref. 24. Based on the reviewer's suggestions, we have carefully reviewed these claims and conducted follow-on experiments.

Major concerns:

1. Some of the authors' claims about the novelty of tRNA-targeting CRISPR-Cas systems seem overstated. On lines 16-17, the authors suggest that "widespread tRNA inactivation" is "a previously unrecognized CRISPR-based immune strategy", while on lines 40-41, the authors state that "no CRISPR-Cas system has yet been demonstrated to exclusively cleave tRNAs". In

reference 24, Cas13 was shown to provide defense against phage through cleavage of the anticodon loop of tRNAs, resulting in inhibition of protein synthesis. The authors of that paper showed that Cas13 cleaves tRNAs more rapidly and more substantially than mRNAs in *E. coli*. While Cas13 may not “exclusively” cleave tRNAs, this previous study suggests that the main route of Cas13-mediated immunity is via tRNA cleavage.

The reviewer raises a fair point about ref. 24, noting that the reference claims tRNA cleavage is a prime mechanism of immune defense by *LshCas13a* (see Fig. 5 in ref. 24) and questioning how this claim could impact our own claims. We also paid close attention to this publication when drafting our manuscript, aiming to fairly distinguish our work while giving credit to their discovery of a link between *LshCas13a* and the cleavage of the uracil-rich anticodon loops of a small subset of tRNAs. Importantly, there are key details from ref. 24 and the Cas13 literature at large that provide a fuller picture that aligns well with our original claim:

- **Target RNA remains the primary substrate of Cas13.** The prevailing concept in the CRISPR field is that Cas13 cleaves its target RNA *in cis* as well as other RNAs *in trans*. While target RNA *cis* cleavage may not inherently be necessary for immune defense, there is extensive evidence that the target RNA is readily cleaved for *LshCas13a* and many other Cas13 nucleases within CRISPR biology (e.g., doi: 10.1126/science.aaf5573, 10.1016/j.chom.2022.05.013, 10.1038/s41586-019-1257-5) and the use of Cas13 for programmable gene silencing in human cells (e.g., doi: 10.1038/nature24049, 10.1016/j.cell.2024.01.035, 10.1016/j.cell.2018.02.033), including with *LshCas13a* (doi: 10.3389/fgene.2020.594576). Our prior work (doi: 10.1016/j.chom.2022.05.013) in particular reported that target RNA cleavage by *LshCas13a* can drive invader clearance at low target RNA levels with little evident collateral cleavage, whereas high target RNA levels lead to widespread RNA cleavage and growth arrest.

Given this, we note that ref. 24 did not evaluate cleavage of the target RNA, while phage infections were conducted by targeting a non-essential transcript in M13. In contrast, we show through direct RNA sequencing that *Ba1Cas12a3* extensively cleaves tRNAs but not the target RNA in TXTL (Fig. 2c-d). Therefore, target RNA *cis* cleavage remains a distinguishing factor differentiating the functions of *LshCas13a* and *Cas12a3*. Additionally, we have explicitly evaluated this difference between the two nucleases in our own experiments, with the resulting data, presented below, consistent with this distinction.

- **Only a small subset of all tRNA sequences underwent cleavage, with limited depletion.** In ref. 24, only a small subset of tRNAs (sequences with U-rich anticodons) were cleaved by *LshCas13a*, including tRNA^{Lys}, tRNA^{Glu}, tRNA^{Gln}, and tRNA^{Thr}. Additionally, only a small fraction (9-23%) were cleaved (see Fig. 5C in ref. 24). In contrast, *Ba1Cas12a3* significantly (Z-score ≥ 2) cleaved 27/47 detected tRNAs in TXTL and 46/49 detected tRNAs *in vitro*, and we observed complete cleavage of detected tRNAs from TXTL and the entire pool *in vitro*. We also obtained comparable results for *Sm3Cas12a3* and *ca23Cas12a3*. Our Northern blotting results demonstrated complete cleavage of tRNA^{Leu} and tRNA^{Ala} in TXTL and *in vitro* by *Ba1Cas12a3*, *Sm3Cas12a3*, and

ca23Cas12a3. Thus, our data support the claim that widespread tRNA cleavage is unique to the Cas12a3 clade.

- **Only *LshCas13a* was tested.** The authors of ref. 24 were careful to limit their claims to *LshCas13a* tested in *E. coli*, as equivalent studies remain to be reported for the many other known Cas13a orthologs. Therefore, tRNA cleavage remains unique to *LshCas13a* until proven otherwise. In contrast, our work demonstrates that three different orthologs sampled across the phylogeny of Cas12a3 cleave the 3' end of tRNAs, suggesting it to be a conserved mechanism characteristic of this clade of nucleases.
- **No available structural or *in vitro* data describe the preference of *LshCas13a* for tRNAs with U-rich anticodon loops over other U-rich RNAs.** Without such data, it is difficult to understand the extent to which and how U-rich anticodon loops are selected over U-rich sequences on mRNAs. In contrast, we provide extensive data demonstrating preferred binding and cleavage of tRNAs as well as how *Ba1Cas12a3* directly interacts with a tRNA to enact cleavage of its 3' tail.

Given these details, we feel justified in continuing to claim the widespread cleavage of tRNAs (i.e., virtually all tRNAs versus only a subset associated with *LshCas13a*) as a novel form of CRISPR-based immunity (see the last sentence of our abstract), even when accounting for ref. 24.

We also had target RNA cleavage in mind when claiming no existing system exclusively cleaves tRNAs. In light of this reviewer's comment, we agree that the associated statement needs to be revised because Cas12a3 can cleave other RNAs besides tRNAs. To better capture Cas12a3's preferential cleavage of tRNAs over other RNAs including the target RNA, we have rephrased the sentence to read the following on p. 3:

“One of these nucleases, the RNA-triggered Cas13 nuclease from *Leptotrichia shahii* (*LshCas13a*), was recently shown to cleave U-rich anti-codon loops associated with a subset of tRNAs when activated in *Escherichia coli*²². However, *LshCas13a* also efficiently cleaves its target RNA at U-rich sequences to drive targeted silencing^{23–26}. Thus, it remains unknown whether CRISPR-Cas systems have evolved to preferentially cleave tRNAs over other RNA species, including their target RNA, as part of an immune response.”

To further support this claim with *LshCas13a* as a point of comparison, we performed a head-to-head comparison in TXTL. Using conditions under which both nucleases effectively silenced GFP production, we assessed cleavage of the target GFP mRNA via RT-qPCR after GFP production was completely halted. Here, the amplification primers bridged the target site (primer pair 2) to detect *cis* cleavage as well as an upstream site (primer pair 1) where no cleavage is expected.

Consistent with target RNA cleavage, *LshCas13a* yielded a 7-fold reduction with primer pair 2 under targeting conditions, whereas *Ba1Cas12a3* yielded no measurable reduction. No reduction was measured for either nuclease with primer pair 1. GFP silencing without measurable mRNA

cleavage by *Ba1Cas12a3* most likely reflects silencing through tRNA cleavage, as shown from direct RNA sequencing under similar conditions. These data emphasize that target RNA cleavage by *LshCas13a* but not *Ba1Cas12a3* is a key element of immunity that clearly distinguishes the two nucleases.

The new data were incorporated as Fig. S9, and we added the following to p. 8:

“Within the diverse set of Cas nucleases, *LshCas13a* is the only other nuclease reported to cleave tRNAs²². As *LshCas13a* primarily cleaves its targeted transcript³⁰ while Cas12a3 exhibits minimal target RNA cleavage (Fig. 2c and S3a), we sought to directly compare their activities. When targeting the same site within an expressed GFP transcript in TXTL, both *LshCas13a* and *Ba1Cas12a3* efficiently reduced GFP fluorescence (Fig. S9a). However, RT-qPCR revealed that the *gfp* transcript underwent cleavage only by *LshCas13a* (Fig. S9b). These results show that *LshCas13a* and *Ba1Cas12a3* act differently upon target recognition, with *LshCas13a* but not Cas12a3 substantially cleaving the target RNA.”

2. It is also unclear that Cas12a3 itself exclusively cleaves tRNAs, as the authors did not probe total RNA cleavage in cells. While their sequencing of rRNA from TXTL reactions indicates that rRNA is not cleaved by Cas12a3, this does not rule out mRNA cleavage by Cas12a3. Indeed, some of the in vitro cleavage results seem to suggest that Cas12a3 can cleave ssRNA (e.g. Figs. 2a, 5a, and S3b). While the authors present a compelling model that the tRLD blocks cleavage by RuvC in the absence of a structured RNA, they have not tested this thoroughly enough to rule out cleavage of ssRNA. It therefore seems overstated to say that Cas12a3 exclusively cleaves tRNA, rather than that it may preferentially cleaves tRNA. If this is the case, it does not seem that there is much distinction in the route of Cas12a3-mediated immunity in

comparison to Cas13. Both cleave tRNAs faster and to a larger degree than other RNAs in the cell, leading to translational repression and cell growth arrest.

We agree that our initial data indicated that some Cas12a3 orthologs can cleave RNA substrates other than tRNAs (e.g., FAM-labeled target and arbitrary RNA sequence in Fig. S3) and some mutated tRNA-derived substrates, even if cleavage is reduced when deviating from a tRNA. We therefore reworded our statement to indicate that tRNAs are preferentially cleaved over other RNAs, including the target RNA.

In addition to rewording our statement, we conducted a series of additional biochemical experiments to better define the sequence and structural requirements for substrate cleavage. These experiments are detailed in response to the next comment and to Reviewer #3, Comment #3. Below is a summary:

- *Ba1Cas12a3* can efficiently cleave tRNA^{Ala} with different truncations that whittled down to a substrate with the anticodon loop, acceptor stem, and 3' CCA tail (Fig. 3e), although this truncation came with a ~62-fold lower binding affinity (Fig. S17).
- *Ba1Cas12a3* could tolerate some mutations to the 3' CCA tail, albeit with reductions in cleavage rates (Figs. 3f and S18b). *Sm3Cas12a3* did not tolerate any mutations to the two cytosines in the 3' CCA tail (Fig. 3f).
- *Ba1Cas12a3* more rapidly cleaves RNA substrates that contain the canonical CCA tail adjacent to a double-stranded RNA than linear substrates containing an internal or 3' CCA (Fig. S18a).

These results further add to our multiple lines of evidence that tRNAs are the preferred substrate of Cas12a3 even over its own RNA target.

Furthermore, while the reviewer noted that *LshCas13a* cleaved tRNAs faster than mRNAs, the authors of ref. 24 did not perform any direct comparisons of cleavage kinetics *in vitro*. Because the higher abundance of tRNAs versus the lower abundance of specific mRNAs in cells could explain why the tRNAs were cleaved more frequently in their *in vivo* experiments, *in vitro* kinetics are needed to support this claim.

Given our additional data, the further insights into ref. 24, and lack of existing experimental data with *LshCas13a* demonstrating preferred cleavage of tRNAs over other RNAs, we see the following as key factors why Cas12a3 is a distinct and important entry in CRISPR biology:

- Cas12a3 preferentially cleaves tRNAs even over its activating target RNA.
- Cas12a3 efficiently cleaves virtually all tRNAs, including post-transcriptionally modified and aminoacylated tRNAs found in bacterial and eukaryotic cells
- Cas12a3 evolved from the Cas12 nucleases commonly associated with the recognition and cleavage of DNA targets.

Specific suggestions for how to address these concerns:

1. The fact that Cas12a3 does not seem to have a preference for the chemical nature of the 3' end of its substrates suggests that it could cleave a CCA sequence that is located internally within an RNA (i.e. with more sequence downstream of the recognition motif). Have the authors investigated whether the sequence specificity observed at the 3' end of substrates in Fig. 5a can also allow cleavage in the middle of ssRNA sequences? This could be related to the cleavage observed for various ssRNA substrates used in the manuscript, and would suggest that other RNAs that contain the same sequence can be cleaved in cells.

We thank the reviewer for this suggestion, which we noted in our response to the prior comment. Based on the reviewer's suggestion, we measured the cleavage kinetics of linear RNAs with an internal or 3' CCA. As shown below, both yielded less efficient cleavage by *Ba1Cas12a3* compared to the minimal tRNA mimic, with the internal CCA being cleaved the least efficiently. These results show that *Ba1Cas12a3* can cleave other CCA-containing RNAs unrelated to tRNAs, albeit at significantly reduced efficiency.

The resulting dataset was incorporated as Fig. S18a, and we added the following to p. 10:

“Nonetheless, the catalytic efficiency was higher with a truncated tRNA than with linear RNAs, even when the CCA was located at the 3' end (Fig. S18a).”

2. Much of the argument for tRNA selectivity stems from the structural differences between the target-bound ternary complex and the target + tRNA bound quaternary complex, in which tRLD movement is proposed to expose the RuvC active site for substrate cleavage. However, it seems likely that the tRLD would be at least somewhat dynamic in the absence of bound tRNA, otherwise it would not be possible to undergo the conformational change required to bind to the

tRNA. Have the authors explored whether there is evidence for an “undocked” conformation of the ternary complex, in which the tRLD is not directly interacting with the RuvC domain? I ask due to two lines of evidence in Fig. S14b. First, it appears that the resolution of the tRLD is quite low in the refined model, suggesting that it may be more flexible than other parts of the structure. Second, it appears that there is a (potentially lower-resolution) class of particles in which the tRLD is not docked directly against the RuvC domain (the greenish class on the right). Did the authors explore this class? While it may be a minor class of particles, the fact that it appears in the classification suggests that Cas12a3 can adopt conformations in which the RuvC domain is exposed. This could allow for cleavage of ssRNAs containing the CCA recognition motif outside of the context of tRNAs.

The reviewer raises an excellent point that prompted us to revisit our proposed role of the tRLD and to identify additional supporting data. While we originally suggested the tRLD blocked access of non-tRNA substrates, we concur that the tRLD could exhibit greater structural flexibility in the absence of tRNA, which is suggested by the lower resolution observed in the tRLD region of the ternary structure maps.

Due to particle orientation bias, we were unable to obtain a high-resolution EM map of the ternary complex. This limitation hindered our ability to detect subtle motions of the tRLD and draw definitive conclusions about its intrinsic dynamics in the absence of tRNA. As the reviewer correctly noted, data processing of the ternary complex revealed two additional 3D classes containing 43,544 (left class in Fig. S22b) and 109,481 (right class in Fig. S22b) particles. However, neither class exhibited clear crRNA-target RNA duplex densities, a critical feature of a fully formed ternary complex. We believe these classes do not necessarily represent functional intermediates and that they rather represent partially assembled complexes or damaged complexes, which disassembled during the vitrification process. Consequently, they were excluded from the final data processing. We further attempted 3D refinement of these classes using cryoSPARC non-uniform refinement, but the resulting EM density maps were of insufficient resolution to enable reliable model building. This information now is included in the figure legend of Fig. S22b with the following:

“3D heterogeneous refinement separated the dataset into three particle classes (43,544; 119,647; and 109,481 particles, left to right). Only the middle class displayed well-defined guide-target RNA duplex densities, indicative of a fully assembled ternary complex. Subsequent non-uniform refinements in cryoSPARC produced a better-resolved EM map exclusively for the middle class, enabling reliable model building.”

Although additional data processing could not confirm other states of the tRLD of the ternary complex, we have modified our description of the domain's role to better align with our biochemical and mutational data. Namely, the large reduction in cleavage activity with the Δ tRLD mutant, including against non-tRNA substrates, led us to instead propose a role in positioning substrates (tRNA or ssRNA) into the RuvC domain for cleavage. We therefore incorporated this proposed role while also eliminating any reference to the tRLD preventing incorrect substrates from accessing the RuvC domain.

For instance, in the Discussion on p. 13, we now state the following:

“...we propose a model (**Fig. 6**) in which target recognition drives a conformational change that reconfigures the nuclease to bind free tRNAs through multiple sequence- and shape-specific contacts. The unique tRLD within the nuclease positions the tRNA tail next to the RuvC active site for cleavage, with extensive tRNA cleavage leading to growth arrest.”

3. If tRLD blocks cleavage of unstructured RNAs, then disruption of interactions between RuvC and tRLD may increase cleavage activity of Cas12a3 against non-tRNA substrates. Have the authors tested whether the tRLD deletion mutant or point mutations that disrupt interactions between tRLD and RuvC increases the promiscuity of Cas12a3 cleavage (e.g. using the random RNA substrates from Fig. 2a)? Is it possible that this is the reason that the Y922A mutant results in an increase in substrate cleavage? In my examination of the ternary structure, it appears that Y992 may be involved in interactions with the RuvC domain. Mutating the residue may make the domain more flexible, leading to the increased ability to bind/cleave tRNA (and potentially other) substrates.

Following the reviewer's suggestion, we determined if the Y922A and Δ tRLD mutations broadened the substrate preference for *Ba1*Cas12a3 using (i) an arbitrary RNA and (ii) the randomized RNA library. As shown below, the arbitrary RNA was substantially cleaved by wild type, but not by either mutant, while the randomized RNA library was poorly cleaved by the Y922A mutant and not cleaved at all by the Δ tRLD mutant. Thus, these mutations did not broaden substrate preference.

The new data were incorporated as Figures 3i and S20, and we added the following to p. 11:

“However, removing this domain or mutating the residues that stack with the terminal adenosine (R902 or N924) greatly reduced both reporter silencing in TXTL and *in vitro* cleavage of different RNA substrates (Figs. 3g-i and S20). Intriguingly, the Y922A mutation increased cleavage activity in TXTL and *in vitro* with the truncated tRNA substrate (Fig. 3g-h) but not with non-tRNA substrates (Figs. 3i and S20), suggesting a more complex mechanistic role for Y922.”

Minor concerns:

1. In the abstract, the authors refer to “adaptive immunity by Cas12a3 effector nucleases”. It is unclear whether Cas12a3 is part of an adaptive immune system, as the authors do not describe the presence of adaptation genes or CRISPR loci associated with *cas12a3* genes. The authors could either add such analysis or remove reference to these nucleases as part of adaptive immune systems.

We agree that the CRISPR-Cas systems should contain CRISPR loci and acquisition genes (*cas1*, *cas2*) at a minimum to associate these with adaptive immunity. Indeed, systems encoding Cas12a3 do contain CRISPR loci as well as *cas1* and *cas2*. To reflect this in the manuscript, we updated Table S1 to indicate the presence of a CRISPR locus as well as acquisition genes (i.e., *cas1*, *cas2*, *cas4*). Finally, we also added the following to p. 4:

“..identifying 61 orthologs primarily from environmental metagenomic assemblies that resolved into two clades distinct from Cas12a2, Cas12a, and each other (Figs. 1a-b and S1, Data S1), the majority of which were associated with CRISPR arrays and the acquisition genes *cas1*, *cas2*, and *cas4* (Table S1).”

2. Lines 108-109: This sentence is a bit confusing and could be reworded, as it makes it sound like both Cas12a3 and Cas12a4 do not cleave the target RNA, when this only applies to Cas12a3.

We thank the reviewer for pointing this out, as it was not our intention to claim that Cas12a4 does not cleave its target RNA. We therefore rephrased the sentence on p. 6 to read the following:

“In contrast, *ApCas12a4* exhibited a continual increase in fluorescence over the course of the reaction without plateauing and efficiently cleaved its target RNA *in vitro* (Fig. S3).”

3. Lines 251-252: The difference in binding between full length tRNA and the tRNA lacking a CCA tail is not very substantial (~2x difference in K_d). It is unclear that this difference in affinity accounts for the differential cleavage, given that the concentration of enzyme used was in excess of both K_d's. It seems more likely that Cas12a3 can bind to structured RNAs regardless of the presence of a CCA tail, but also displays a significant sequence specificity for its cleavage sites, preventing cleavage of structured RNAs that lack this sequence.

The reviewer raises a good point, as a small reduction in binding affinity doesn't necessarily explain the substantial reduction in cleavage efficiencies. We removed this claim from the manuscript.

4. In heatmaps like in Fig. 1f, how was fluorescence production normalized? Please include this information in the figure legend or methods. The explanation on lines 790-791 is a bit vague.

We apologize for any confusion around how the heat maps were generated. We now clarify how the normalization was performed with the following on p. 30:

“For the heat map in Figure 1f, initial reaction rates in the linear range were normalized within each nuclease by setting the highest rate across reporters to one.”

5. In Fig. S3b, there appears to be a discrete, slightly shorter product when *Sm3Cas12a3* is incubated with the unlabeled target and 5' FAM RNA substrate. Is this reproducible and is it due to Cas12a3 cleavage activity? It appears that the enzyme may be cleaving at a discrete site near the 3' end of the RNA.

The reviewer makes an astute observation, where Figure S3b could indicate that *Sm3Cas12a3* and *Ba1Cas12a3* are cleaving at slightly offset positions. Admittedly, it's difficult to conclude whether the shift is significant based on the lower resolution of the gels. As the purpose of these gels was to show that the Cas12a3 orthologs cleave RNA but not ssDNA or dsDNA, we were not aiming for greater resolution.

Instead, the sequenced cleavage sites in Figures 2D, S4 and S6 offer a more pointed comparison of *Ba1Cas12a3* and *Sm3Cas12a3* cleavage preferences given their single-base resolution. Here, the cleavage sites for *Sm3Cas12a3* are located 1-2 nts upstream of the cleavage sites for *Ba1Cas12a3* for multiple tRNAs, suggesting some differences in their mechanism of action. As we expect there to be even more differences across the set of Cas12a3 nucleases, such studies are best carried out as follow-on work. At the same time, we added the following to make note of these differences and what they might mean for the functional diversity within the Cas12a3 family:

On p. 7-8 of the Results:

“Similar cleavage patterns were observed with *Sm3Cas12a3* and *ca23Cas12a3* (identified in a wastewater microbial metagenome), albeit with the cleavage sites for *Sm3Cas12a3* shifted slightly upstream (**Figs. 1a and S6**), indicating possible mechanistic differences within the Cas12a3 clade.”

On p. 14-15 of the Discussion:

“Deviations are also possible within these clades, as illustrated by the different tRNA cleavage sites and substrate requirements of *Ba1Cas12a3* and *Sm3Cas12a3*. These findings underscore the broader diversity of Cas12 nucleases and suggest the existence of additional, yet to-be-discovered functions.”

Referee #2 - structural biology/biochemistry of CRISPR

Summary

In this manuscript, Dmytrenko et al. describe a new subfamily of CRISPR-Cas12 enzymes (Cas12a3 and Cas12a4) and show remarkably that upon activation by a complementary RNA-target Cas12a3 has a specific preference for the cleavage of the CCA tail of most E coli tRNAs, when expressed and activated in E. coli using either plasmid or phage challenge. They then go on to determine the cryo-EM structures of multiple states of Cas12a3 assembly, targeting, and catalysis, and show there are important domains within Cas12a3 that make tRNA-specific contacts; this results in a unique RuvC gating / accessibility mechanism that precludes other RNA substrates from gaining access in both the apo, binary, and ternary forms of the enzyme complex. Finally, they explore the design of mini-tRNA-like synthetic fluorescence-quencher reagents and discover that for Cas12a3, they can design a specific substrate that can be cleaved efficiently in trans by Cas12a3 upon activation but that is not cleaved by LwaCas13a or PsmCas13b, paving the way for the potential use of Cas12a3 in multiplexed diagnostic settings.

Overall, this an exciting finding that adds significantly to recent experimental momentum around tRNAs being a central target of prokaryotic innate immunity, and most of the interpretations are supported by well-controlled experiments and explained in a well-written and easy to follow manuscript. That said, some claims and interpretations that impact the conclusions and usefulness of the potential downstream diagnostic technology are oversold and not backed up by the required data, and as a result some major and minor concerns need to be addressed. See below for my specific comments.

We thank the reviewer for the supportive and helpful comments. In response to concerns that some of our claims and interpretations were overstated or not supported by the required data, we conducted several additional experiments and have revised the text to align our claims more accurately with our data. We detail our responses to each of the reviewer's concerns below.

Major Comments

* While I understand it's likely that translation is blocked given the amount of de-acylation / CCA cleavage you observe in vitro and in TXTL experiments, you haven't shown this explicitly in vivo under phage infection or plasmid challenge conditions. The claim really relies on loss of GFP expression with in vitro supplementation of Cas12 into TXTL cell-free reactions; is there a chance that the loss of GFP expression isn't due to loss of global protein translation directly because of loss of tRNA availability but rather in sensing of deacylated tRNAs by a stress response (e.g. stringent response by a combined loss of translation and transcription)? (More on this below.) Have the authors considered running total protein assays in vivo to show loss of general translation / protein production upon phage infection or doing a plasmid challenge experiment (without antibiotics)? One way would be to look at nascent polypeptide production using a O-Propargyl-puromycin assay or equivalent.

We thank the reviewer for raising this point and prompting us to better define the impact of Cas12a3 activation *in vivo*. We considered the O-propargyl-puromycin assay or the equivalent based on its sensitivity to nascent protein synthesis, although it would not distinguish between translational arrest caused directly by tRNA depletion versus indirect effects such as transcriptional inhibition or the accumulation of ppGpp through the stringent response. Instead, and spurred by the following comment, we focused on the potential role of the stringent response in *E. coli*. Here, we deleted *relA* in our *E. coli* strain, encoding the central regulator of the stringent response activated by an uncharged tRNA in the ribosomal A-site, and we repeated the plasmid interference assay. As shown below, after confirming the *relA* deletion (panel a), we found that plasmid interference was maintained in the $\Delta relA$ strain across three Cas12a3 orthologs as well as our control nuclease *MpCas12a2* (panel b). Thus, these results rule out immunity dependent on the stringent response through RelA.

The resulting dataset was incorporated as Fig. S10, and we added the following to p. 8-9:

“Trimming tRNA tails by Cas12a3 would prevent tRNAs from participating in translation, potentially driving global translational shutdown and arresting cell growth. Alternatively, growth arrest could be mediated by tRNA cleavage products that trigger systemic stress responses, such as the stringent response, which is activated by the detection of deacetylated tRNAs bound by the ribosome by the RelA protein^{38–41}. However, deleting *relA* did not impair plasmid interference by any of the tested Cas12a3 orthologs in *E. coli* (Fig. S10).”

We also employed TXTL as a simplified means to assess the impact of Cas12a3 activation on gene expression. As shown with our head-to-head comparison with *LshCas13a* (see Reviewer #1, Comment #1), no target RNA cleavage was detected with *Ba1Cas12a3*, supporting translational inhibition rather than target RNA cleavage or transcriptional inhibition in TXTL. In contrast, substantial cleavage at the target site was detected with *LshCas13a*, in line with Cas13 nuclease principally cleaving its RNA target. These results were incorporated as Fig. S9.

Even with these new data in hand, we agree that other mechanistic steps could connect extensive cleavage of tRNA tails to growth arrest. We therefore were cautious not to claim any specific

mechanism connecting tRNA cleavage and growth arrest, and instead added the following to p. 13:

“This immune response does not depend on a traditional stringent response but instead likely arises directly from translational inhibition or from RelA-independent stress triggered by the tRNA cleavage products.”

We also modified our proposed model in Fig. 6 to the following to avoid claiming translational repression:

* Relatedly, deacylated tRNAs are known activators of the stringent response through activation of RelA (p)ppGpp synthesis and its downstream action as a secondary messenger, which can also ultimately result in transcriptional and translational shut-off amongst other phenotypes such as biofilm formation, etc. However, while it is known from structural studies that RelA contacts deacylated CCA tails of the tRNAs when docked into the ribosome, it's still not clear to me functionally whether it would be possible that de-acylated tRNAs missing their CCA tails (in your case) are still able to activate RelA and the stringent response. I think minimally a comment or discussion on this potential mechanism in the text is required. Ideally, if there is some facile way (HPLC or a stringent response reporter or equivalent) to determine whether the stringent response is being activated during phage infection or plasmid challenge, that would certainly elevate the significance of the manuscript's findings.

We thank the reviewer for bringing the stringent response to our attention as a potential explanation for growth arrest. As noted above, we tested the impact of deleting *relA* on plasmid interference, finding that interference was maintained in the absence of *relA*. These results demonstrate that the growth repression caused by Cas12a3 activation is most likely a direct consequence of cleaved tRNAs impairing translation or a distinct cellular stress response.

* I think the authors need to be more careful with the interpretation as to whether acylation / charging and tRNA chemical modifications affect the specificity and efficiency of tRNA cleavage by Cas12 here. For example, modifications are well known to alter the 3D structure and conformational dynamics of tRNA, so it can't be ruled out that changes in structure and dynamics could occur in the presence or absence of modifications and these could have an indirect effect on tRNA binding and/or cleavage rates even if there are no direct modification-specific contacts between the protein and the tRNA. Furthermore, without much more careful thermodynamic binding and/or kinetic cleavage assays with fully modified vs. unmodified tRNAs being carried out (a tough experiment, so not suggesting it be done unless purified specific modified tRNAs can be sourced, which is not straightforward), it is an overinterpretation to claim that modifications don't matter here. This reviewer suggests toning this language down and make it clear that maybe modifications may have an indirect effect, in lieu of more careful kinetic/thermodynamic measurements.

We thank the reviewer for raising this concern. We originally made the claim based on Cas12a3 cleaving tRNAs with or without the appended amino acid as well as with or without chemical modifications, and we now have further data showing *Ba1*Cas12a3 can cleave yeast tRNAs (see Reviewer #3, Comment #5). At the same time, we agree it is difficult to make broad claims about the impact of chemical modifications without thorough testing using RNAs with prescribed chemical modifications that would be extremely difficult to obtain. Therefore, we limited our conclusions to Cas12a3 cleaving tRNAs with or without chemical modifications.

* While the discovery that an orthogonal, synthetic trans-RNA substrate can be designed for Cas12a3 and might enable multiplexing, as far as I can tell the manuscript doesn't actually demonstrate the ability to multiplex Cas12a3, LwaCas13a and PsmCas13b in a one-pot reaction, which is minimally the level of proof of concept this reviewer would expect when making claims about multiplexing in a journal of this impact. The demonstration of multiplexing could be achieved by using a contrived sample where one could spike in synthetic RNAs (at a range of concentrations) into a total RNA sample from humans or E coli for example and then testing for a three-way multiplex readout from a single pot (i.e. all detection reactions in one well) using fluorescence or equivalent. This would certainly increase the impact of this section of the manuscript and perhaps offer more information about the benefits and/or potential limitations of Cas12a3-based detection in more real-world samples (e.g. one could imagine that all the tRNA present in a real-world RNA sample could act as a competitive inhibitor for the fluorescence substrate and decrease the signal-to-noise/dynamic range of the assay in a way that makes multiplexing (or detection in general) achievable).

We thank the reviewer for encouraging us to further develop the multiplexed detection experiment. To demonstrate multiplexed detection, we developed a one-pot mix containing three different RNA-guided CRISPR-associated enzymes that each (i) target a distinct RNA derived from common respiratory viruses (i.e., SARS-CoV-2, Influenza Virus A, and Respiratory Syncytial Virus) and (ii) cleave a specific but distinct nucleic acid in *trans*. As shown below, this one-pot mix could detect the target RNAs individually or in combination. To ensure that competing RNAs

did not diminish signal detection, we performed the assay in the presence of total human RNA. Notably, the addition of human RNA up to a 100-fold estimated molar excess, improved the signal-to-noise ratio. These new results further support our claim that Cas12a3, in combination with other Cas nucleases, can expand multiplexed molecular diagnostics.

These data are now included as Figures 5d-g S25d-f, and we added the following to p. 13:

“Leveraging this specificity, we combined the three nucleases and their cognate probes, each labeled with a distinct fluorophore, into a one-pot reaction for multiplexed RNA detection. This one-pot setup allowed the separate and combinatorial detection of RNA transcripts derived from the respiratory viruses SARS-CoV-2, RSV, and Influenza A (Figs. 5d–g and S25d-f). Interestingly, the presence of a large excess of human total RNA did not interfere with, and even enhanced, detection (Figs. 5f and S25e).”

Minor Comments

*Line 41 and elsewhere in the text: The authors make a claim that Cas12a3 cleaves tRNA exclusively; however, your collateral cleavage assay potentially suggests linear substrates could get cleaved at some low level in vivo like what is seen in ref. 24. Given TXTL was used in the assay, which is possibly depleted of all other mRNAs and supplementation of additional tRNAs as part of the master mix is common, making the TXTL possess a large overabundance of tRNAs relative to other targets, is it possible that cleavage of other RNAs is possible in vivo but could be missed in TXTL? On this note, I feel like the discoveries in ref. 24 are downplayed in

this manuscript and need to be cited more positively; the limitation in the use of TXTL also needs to be addressed or the related claims need to be toned down somewhat.

We thank the reviewer for bringing these points to our attention, which were also raised in part by Reviewer #1 (Comments #1-2). To address these points, we removed any claims implying that Cas12a3 exclusively cleaves tRNAs and provide additional experimental evidence that tRNAs are the preferred substrates. We also provided a more suitable description of ref. 24 as well as other relevant literature for *LshCas13a* (see p. 3) while also directly comparing *LshCas13a* and *Ba1Cas12a3* in TXTL (see Fig. S9).

The reviewer also raises the limitations of TXTL when identifying cleaved RNAs as part of an immune response. To the reviewer's point, there are notable differences between the intracellular environment of a cell and our TXTL system, an *E. coli* lysate supplemented with energy sources and components to drive transcription and translation, that could bias our ability to identify the preferred substrate driving immunity. However, the two are closer than might be expected that allowed us to connect tRNA cleavage to immune defense. Below are a few examples relevant to our study:

- Our TXTL system contains active central metabolism that can replenish the components and cofactors for transcription and translation, mimicking more than only transcription and translation. The system was also shown to contain 500 - 800 proteins (doi: 10.1002/rcm.8438), representing ~30% of proteins expressed during exponential phase growth (doi: 10.15252/msb.20209536).
- Our TXTL system contains a lower concentration of tRNAs (~0.2 mg/mL, doi: 10.3791/50762) than in *E. coli* (~8 mg/mL, doi: 10.1002/rcm.8438), while expressed mRNAs can reach 10% by mass of these tRNAs (doi: 10.1038/s41598-019-48468-8).
- Our TXTL system contains a large excess of ribosomal RNAs (~2 mg/mL, doi: 10.1016/s0022-2836(60)80029-0, 10.1038/s41598-019-48468-8) that should be cleaved if they are preferred substrates.
- Using TXTL, we could show that *Ba1Cas12a3* does not measurably cleave its target through both direct RNA sequencing (see Fig. 2c) and in the head-to-head comparison with *LshCas13a* (see Fig. S9). In parallel, we show through *in vitro* assays that *Ba1Cas12a3* preferentially cleaves our truncated tRNA substrate over linear RNAs containing CCA (see Fig. S18a).
- Immune defense is conferred in a heterologous host (*E. coli*) unrelated to any of the originating species harboring Cas12a3, lending to Cas12a3 cleaving an RNA substrate universal across bacteria. These components, such as rRNAs, tRNAs and the target RNAs, are also present in TXTL and were probed in our studies.
- We show that RelA is not required for immune defense in *E. coli* (see Fig. S10), ruling out a more complex stress response that may not be present in our TXTL system.

These insights reinforce our conclusion that Cas12a3 preferentially cleaves tRNAs as part of immune defense. That said, to the reviewer's point, we cannot definitively rule out that other RNAs

are cleaved as part of an *in vivo* immune response. Therefore, we added the following to p. 13-14:

“Despite the high binding preference for tRNAs *in vitro*, it also remains possible that the nuclease targets additional RNAs not present in our TXTL system, which may further contribute to immune defense.”

* Growth arrest claims: A better explanation of the “indicative of growth arrest” on line 77 in my opinion is required. While it is likely there is a growth reduction and that is why there are fewer transformants, this is a less direct measurement, and a lack of transformation could be possibly due to other reasons. In this case the language needs to be a lot clearer when describing the experimental interpretation of Fig. 1d and perhaps growth curves in Fig. S2c needed to be more clearly invoked when describing “indicative of growth arrest”.

We thank the reviewer for noting where “growth arrest” could be more rigorously claimed. We therefore reworded this section on p. 3 to read the following:

“Notably, even without antibiotic selection for the target plasmid (**Fig. 1d**), the Cas12a3 and Cas12a4 nucleases still reduced the number of transformants similar to a representative Cas12a2 from the microbial community of *Microcerotermes parvus* (*MpCas12a2*). We obtained comparable results for different target sequences (**Fig. S2b**) and observed impaired growth in liquid culture without antibiotic selection (**Fig. S2c**).... Thus, while Cas12a3 and Cas12a4 nucleases arrest growth upon activation similar to Cas12a2, both nucleases execute a distinct mechanism of immunity.”

* Line 81 or in the figure: It might be worth explaining how the SOS reporter works (RecA promoter, etc.) It’s not clear from the text or the figure and thus may make it easier for the reader to interpret.

Following the reviewer’s suggestion, we updated the sentence on p. 5 to read the following:

“However, unlike Cas12a2, the Cas12a3 and Cas12a4 nucleases did not induce a measurable SOS DNA damage response based on a transcriptional reporter driven from the *recA* promoter (**Fig. 1e**)²⁷.”

* Lines 106-107: With respect to Cas13’s “largely indiscriminate” cleavage, is that actually true? This needs to be tempered and something like the following needs to be added: “although more recent evidence points a preference of certain Cas13 ortholog for tRNAs...” I think this needs to be added because as it is currently written it is not a completely fair reference to what is known in the literature.

In light of this reviewer's comment as well as Comments #1-2 from Reviewer #1, we have rephrased this sentence to more explicitly call out the recent work on LshCas13a cleaving tRNAs. This section on p. 3 now reads:

“One of these nucleases, the RNA-triggered Cas13 nuclease from *Leptotrichia shahii* (*LshCas13a*), was recently shown to cleave U-rich anti-codon loops associated with a subset of tRNAs when activated in *Escherichia coli*²². However, *LshCas13a* also efficiently cleaves its target RNA at U-rich sequences to drive targeted silencing^{23–26}. Thus, it remains unknown whether CRISPR-Cas systems have evolved to preferentially cleave tRNAs over other RNA species, including their target RNA, as part of an immune response.”

* Fig. 2b: Perhaps use thicker lines in the TXTL time courses; it was hard to see the blue line in there for the MbCas12a2 especially.

We thank the review for this suggestion. As suggested, we used 50% thicker dots within Fig. 2b as well as in Fig. 2a. Below are the revised figure panels.

* The colored/semi-transparent keys in the figures (Fig. 1 especially, but others, too) are hard to interpret, a little small and the key difference in transparency is too subtle.

We thank the reviewer for this suggestion as well. To make it easier to distinguish these conditions, we removed the partial transparency and instead used 100% opaque colors from the same family. This was done to all bar graphs that had used the partial transparency. Fig. 1d-e is shown below as an example.

* How were the non-cleaving conditions obtained for the cryo-EM structure determination? It would be nice to include that detail in the main text more clearly. EDTA? Mutants? No metal +/- EDTA?

We first reconstituted the binary complex in the presence of Mg^{2+} and subsequently purified the homogeneous complex by size-exclusion chromatography (SEC) using a Mg^{2+} -free buffer. After purification, target RNA and tRNA were sequentially added to the complex, with a 10-minute interval between additions, after which the mixture was kept on ice to minimize cleavage prior to vitrification, thereby preserving complex integrity for cryo-EM analysis.

To better clarify the non-cleaving conditions without bogging down the main text, we have added the following to p. 9:

“To limit tRNA cleavage, the complex was reconstituted on ice with a lowered Mg^{2+} concentration.”

as well as the following to p. 40:

“Target RNA and tRNA^{Ala} were sequentially added to 50 μ l of the Mg^{2+} -free peak fraction of the binary complex ($\sim 20 \mu$ M) at 1:1.2 and 1:2 molar ratios, respectively. The target RNA was incubated at room temperature for 10 min, followed by the addition of tRNA, and the mixture was kept on ice for 30 min prior to vitrification.”

Referee #3 – tRNA cleavage

This is an interesting novel work showing that Cas12a3 effector nucleases from Type V CRISPR-Cas systems in bacteria can selectively cleave tRNAs upon target RNA recognition. Interestingly, target RNA triggers Cas12a3 to cleave the conserved 5'-CCA-3' tail of diverse tRNAs, driving growth arrest and anti-phage defense. The authors further provide structural data that back up their in vitro data. By using structural data, further mutagenesis analysis was used to identify critical amino acid contacts that provide specificity towards tRNA substrate. Finally, the authors engineered synthetic substrates that, together with the 1 specificity of Cas12a3 toward tRNA substrates, expanded the capacity of selected CRISPR-based diagnostic platforms for multiplexed RNA detection.

The data is solid and I only have few issues to be addressed in a revised version.

We thank the reviewer for their supportive comments and for highlighting the novelty of our work and the strength of the underlying data. We addressed each of the raised issues below with additional experiments.

1. It would be great to back up data from Figs. 2/S4 on the nanopore RNA-seq with representative northern blotting on targeted (tRNA) and non-targeted RNAs (5S).

Following the reviewer's suggestion, we conducted northern blotting analysis to complement our results from direct RNA sequencing. Specifically, we repeated the cleavage reactions in TXTL under targeting and non-targeting conditions using our three Cas12a3 orthologs. For northern blotting analysis, we selected tRNA^{Ala}(UGC) (Class I, lacking the variable loop) and tRNA^{Leu}(UAG) (Class II, containing the variable loop), identified from Nanopore data of TXTL and *in vitro* *E. coli* tRNA samples, as showing intermediate read abundance and cleavage. 5S RNA was included as a loading and cleavage control. As shown below, all tRNAs exhibited a reduction in size in line with removal of the 3' tail under targeting conditions for all three orthologs, whereas the 5S RNA remained intact.

The resulting images were incorporated as Fig. S5, and we added the following to p. 7:

“In contrast, many of the reads mapped to tRNAs (27/47) were significantly (Z-score ≥ 2) truncated roughly two to four nucleotides upstream of their 3' aminoacylated ends up to the discriminator base and CCA tail conserved across all tRNAs (Fig. 2c and S4c), with cleavage of selected tRNAs confirmed by northern blotting analysis (Fig. S5). Similar cleavage patterns were observed for *Sm3Cas12a3* in TXTL (25/47) (Figs. S4b,d and S5).”

2. The same as in S6, selected tRNA probes should be used for two representative tRNAs that are "well cleaved" vs "less efficiently cleaved" from the pool of bacterial tRNAs. This will serve as controls for RNA-seq.

We agree such controls could support the extent of cleavage determined by direct RNA sequencing. At the same time, we wanted to be consistent with the northern blotting analysis performed on the TXTL-extracted RNAs given the direct link between RNA direct sequencing of tRNAs in TXTL and *in vitro*. Repeating northern blotting using the *E. coli* tRNAs under *in vitro* cleavage conditions (see below), we again detected the small size reduction for the two probed tRNAs under targeting conditions using all three Cas12a3 orthologs. These results further confirm the ability of activated Cas12a3 to cleave tRNA tails that complement the direct RNA sequencing results.

We acknowledge the reviewer’s suggestion to probe tRNAs cleaved to varying extent, although we felt it would be more informative to probe RNAs representing the two tRNA classes divided based on the presence or absence of the variable loop, underscoring the breadth of tRNAs cleaved by Cas12a3.

The resulting images were incorporated as Fig. S7, and we added the following to p. 8:

“Consistent with the direct RNA sequencing results, *Ba1Cas12a3* trimmed the 3' end of the entire pool of *E. coli* tRNAs labeled with a 5' fluorophore (Fig. 2e top), including specific tRNAs detected by northern blotting analysis (Fig. S7).”

3. Mutagenesis data should be complemented from tRNA side (to back up Fig. 3); please use *in vitro* transcribed tRNA variants with changes in the CCA-tail (purine-to-purine/purine-pyrimidine).

We agree that mutating the tRNA as part of Fig. 3 would nicely complement the tested Cas12a3 mutants. We therefore performed a series of new experiments and integrated the data from Figure 5a as new content in Figure 3. The new content is divided into two general experiments described below:

Progressing from full-length tRNA to truncated mimic. To make the jump from the full-length tRNA to the truncated mimic used for mutational analysis, we assessed the capacity of Ba1Cas12a3 to cleave a series of truncations that removed individual or multiple domains within the tRNA. As shown below, *Ba1Cas12a3* fully cleaved all truncations, showing that the stem and tail is a compact mimic to further interrogate the sequence and structural requirements for RNA cleavage.

As part of these experiments, we also measured the binding affinity of the truncated version of tRNA^{Ala}. The *dBa1Cas12a3* ternary complex bound the h1 mimic with approximately 62-fold lower affinity compared to the full-length tRNA, indicating some loss in substrate preference when truncating the tRNA.

These results were integrated as Fig. S17, and we added the following to p. 10:

“A minimal substrate comprising the anti-codon loop, acceptor stem, and 3' CCA (h1) was also cleaved, although it was bound by *dBa1Cas12a3* with 62-fold lower affinity than the full-length tRNA (Fig. S17).”

Testing an expanded set of mutations to the CCA tail with the truncated tRNA substrate.

Using the truncated tRNA substrate (h1) as a starting point, we explored multiple mutations to the CCA tail, including purine-to-pyrimidine mutations (previously Fig. 5a) as well as purine-to-purine/pyrimidine-to-pyrimidine mutations (new data) using end-point cleavage assays as well as kinetic assays based on separation of a conjugated fluorophore-quencher pair. As shown below, *Ba1Cas12a3* was more tolerant of the two cytosines being mutated to another purine rather than a pyrimidine, while *Sm3Cas12a3* was not tolerant to any mutations to the cytosines. These results further define the sequence and shape requirements for Cas12a3 substrates and suggest Cas12a3 enzymes exhibit varied substrate specificities.

These results were integrated as Figs. 3f and S18b, and we added the following to p. 10:

“Within the CCA tail, *Ba1Cas12a3* was sensitive to cytosine mutations, with transversions (C-to-G) impairing cleavage more strongly than transitions (C-to-U) (**Figs. 3f and S18b**). In contrast, *Sm3Cas12a3* failed to cleave upon any cytosine substitution (**Fig. 3f**), suggesting a divergent strategy of substrate recognition.”

4. It is necessary to determine whether Cas12a3 can cleave tRNAs in the complex with translation elongation factors, as well as in the ribosomal fractions.

The reviewer raises the excellent point that a portion of cellular tRNAs are bound by translation elongation factors or within the ribosome, which could prevent Cas12a3 from cleaving these bound tRNAs. We opted to compare the binding surfaces on the tRNA, as strongly overlapping binding surfaces would preclude Cas12a3 from cleaving a bound tRNA. Comparing our structural data for the *Ba1Cas12a3* ternary structure to the published structures of tRNA bound to EF-Tu alone (PDB ID: 1b23) or in the ribosome (PDB ID: 4v5l), we found that *Ba1Cas12a3* and EF-Tu bind the same surface covering the t-arm and acceptor stem, with similar overlap when the EF-Tu-tRNA complex is bound to the ribosome. The resulting strong steric clash therefore, should limit the cleavage activity of *Ba1Cas12a3* to free tRNAs.

We incorporated the resulting structural prediction as Fig. S16, and we added the following to p. 10:

“Finally, the region of the tRNA bound by *Ba1Cas12a3* is the same as that bound by the elongation factor Tu (EF-Tu) alone or in complex with the ribosome (Fig. S16), suggesting that *Ba1Cas12a3* cleaves free tRNAs not actively engaged in protein synthesis.”

5. While predicted, it would be interesting to determine whether Cas12a3 can cleave eukaryotic tRNAs, which are more densely modified. This may have applications in the future research.

We agree that determining whether Cas12a3 can cleave eukaryotic tRNAs would be an interesting extension, particularly for potential downstream use of these nucleases. We therefore conducted an *in vitro* cleavage reaction using the three Cas12a3 orthologs and tRNAs extracted from yeast, with the amino acid replaced with a Cy5 fluorophore. As shown below, all three orthologs yielded efficient cleavage of the entire tRNA pool under targeting conditions, in line with cleavage of the 3' end of tRNAs.

We incorporated the resulting image as Fig. S8b, and we added the following to p. 8:

“*Ba1*Cas12a3 similarly cleaved the same pool of chemically modified tRNAs with the amino acid removed and replaced with a fluorophore (**Fig. 2e bottom**). *Ba1*Cas12a2 also cleaved *in vitro*-transcribed tRNAs lacking both chemical modifications and charged amino acids (**Fig. S8a**) and bulk tRNA isolated from yeast (**Fig. S8b**).”